# Local-Global MCMC kernels: the best of both worlds

**Sergey Samsonov**[1]    **Evgeny Lagutin**[1]    **Marylou Gabrié**[2]    **Alain Durmus**[3]

**Alexey Naumov**[1]    **Eric Moulines**[2]
[1]HSE University    [2]Ecole Polytechnique    [3]ENS Paris-Saclay
{svsamsonov,elagutin,anaumov}@hse.ru
{eric.moulines,marylou.gabrie}@polytechnique.edu
alain.durmus@ens-paris-saclay.fr

## Abstract

Recent works leveraging learning to enhance sampling have shown promising results, in particular by designing effective non-local moves and global proposals. However, learning accuracy is inevitably limited in regions where little data is available such as in the tails of distributions as well as in high-dimensional problems. In the present paper we study an Explore-Exploit Markov chain Monte Carlo strategy (Ex$^2$MCMC) that combines local and global samplers showing that it enjoys the advantages of both approaches. We prove $V$-uniform geometric ergodicity of Ex$^2$MCMC without requiring a uniform adaptation of the global sampler to the target distribution. We also compute explicit bounds on the mixing rate of the Explore-Exploit strategy under realistic conditions. Moreover, we also analyze an adaptive version of the strategy (FlEx$^2$MCMC) where a normalizing flow is trained while sampling to serve as a proposal for global moves. We illustrate the efficiency of Ex$^2$MCMC and its adaptive version on classical sampling benchmarks as well as in sampling high-dimensional distributions defined by Generative Adversarial Networks seen as Energy Based Models.

## 1   Introduction

We consider the setting where a target distribution $\pi$ on a measurable space $(\mathbb{X}, \mathcal{X})$ is known up to a normalizing constant and one tries to estimate the expectations of some function $f : \mathbb{X} \to \mathbb{R}$ with respect to $\pi$. Examples include the extraction of Bayesian statistics from posterior distributions derived from observations as well as the computation of observables of a physical system $x \in \mathbb{X}$ under the Boltzmann distribution with non-normalized density $\pi(x) = \mathrm{e}^{-\beta U(x)}$ for the energy function $U$ at the inverse temperature $\beta$.

A common strategy to tackle this estimation is to resort to Markov chain Monte Carlo algorithms (MCMCs). The MCMC approach aims to simulate a realization of a time-homogeneous Markov chain $\{Y_n, \ n \in \mathbb{N}\}$, such that the distribution of the $n$-th iterate $Y_n$ with $n \to \infty$ is arbitrarily close to $\pi$, regardless of the initial distribution of $Y_0$. In particular, the Metropolis-Hastings kernel (MH) is the cornerstone of MCMC simulations, with a number of successful variants following the process of a *proposal* step followed by an *accept/reject* step (see e.g. [62]). In large dimensions, proposal distributions are typically chosen to generate local moves that depend on the last state of the chain in order to guarantee an admissible acceptance rate. However, local samplers suffer from long mixing times as exploration is inherently slow, and mode switching, when there is more than one, can be extremely infrequent.

On the other hand, independent proposals are able to generate more global updates, but they are difficult to design. Developments in deep generative modelling, in particular versatile autoregressive and normalising flows [39, 37, 20, 55], spurred efforts to use learned probabilistic models to improve

36th Conference on Neural Information Processing Systems (NeurIPS 2022).

the exploration ability of MCMC kernels. Among a rapidly growing body of work, references include [36, 2, 53, 25, 33]. While these works show that global moves in a number of practical problems can be successfully informed by machine learning models, it remains the case that the acceptance rate of independent proposals decreases dramatically with dimensions – except in the unrealistic case that they perfectly reproduce the target. This is a well-known problem in the MCMC literature [12, 71, 1], and it was recently noted that deep learning-based suggestions are no exception in works focusing on physical systems [19, 46].

In this paper we focus on the benefits of combining local and global samplers. Intuitively, local steps interleaved between global updates from an independent proposal (learned or not) increase accuracy by allowing accurate sampling in tails that are not usually well handled by the independent proposal. Also, mixing time is usually improved by the local-global combination, which prevents long chains of consecutive rejections. Here we focus on a global kernel of type iterative-sampling importance resampling (i-SIR) [73, 4, 5]. This kernel uses multiple proposals in each iteration to take full advantage of modern parallel computing architectures. For local samplers, we consider common techniques such as Metropolis Adjusted Langevin (MALA) and Hamiltonian Monte Carlo (HMC). We call this combination strategy Explore-Exploit MCMC (Ex$^2$MCMC) in the following.

**Contributions**   The main contributions of the paper are as follows:

- We provide theoretical bounds on the accuracy and convergence speed of Ex$^2$MCMC strategies. In particular, we prove $V$-uniform geometric convergence of Ex$^2$MCMC under assumptions much milder than those required to prove uniform geometric ergodicity of the global sampler i-SIR alone.
- We provide convergence guarantees for an adaptive version of the strategy, called FlEx$^2$MCMC, which involves learning an efficient proposal while sampling, as in adaptive MCMC.
- We perform a numerical evaluation of Ex$^2$MCMC and FlEx$^2$MCMC for various sampling problems, including sampling GANs as energy-based models. The results clearly show the advantages of the combined approaches compared to purely local or purely global MCMC methods.

**Notations**   Denote $\mathbb{N}^* = \mathbb{N} \setminus \{0\}$. For a measurable function $f : \mathbb{X} \mapsto \mathbb{R}$, we define $|f|_\infty = \sup_{x \in \mathbb{X}} |f(x)|$ and $\pi(f) := \int_{\mathbb{X}} f(x)\pi(\mathrm{d}x)$. For a function $V : \mathbb{X} \mapsto [1, \infty)$ we introduce the $V$-*norm* of two probability measures $\xi$ and $\xi'$ on $(\mathbb{X}, \mathcal{X})$, $\|\xi - \xi'\|_V := \sup_{|f(x)| \leq V(x)} |\xi(f) - \xi'(f)|$. If $V \equiv 1$, $\|\cdot\|_1$ is equal to the total variation distance (denoted $\|\cdot\|_{\mathrm{TV}}$).

## 2   Explore-Exploit Samplers

Suppose we are given a target distribution $\pi$ on a measurable space $(\mathbb{X}, \mathcal{X})$ that is known only up to a normalizing constant. We will often assume that $\mathbb{X} = \mathbb{R}^d$ or a subset thereof. Two related problems are sampling from $\pi$ and estimating integrals of a function $f : \mathbb{X} \mapsto \mathbb{R}$ w.r.t. $\pi$, i.e., $\pi(f)$. Among the many methods devoted to solving these problems, there is a popular family of techniques based on *Importance Sampling* (IS) and relying on independent proposals, see e.g. [1, 74]. We first give a brief overview of IS, to describe the global sampler i-SIR. We recall ergodicity results for the latter before investigating the Explore-Exploit sampling strategy which couples the global sampler with a local kernel. Then we present the main theoretical result of the paper on the ergodicity of the coupled strategy.

### 2.1   From Importance Sampling to i-SIR

The primary purpose of IS is to approximate integrals of the form $\pi(f)$. Its main instrument is a (known) *proposal distribution*, which we denote by $\lambda(\mathrm{d}x)$. To describe the algorithm, we assume that $\pi(\mathrm{d}x) = w(x)\lambda(\mathrm{d}x)/\lambda(w)$. In this formula, $w(x)$ is the *importance weight* function assumed to be known and positive, i.e., $w(x) > 0$ for all $x \in \mathbb{X}$, and $\lambda(w)$ is the *normalizing constant* of the distribution $\pi$. Typically $\lambda(w)$ is unknown. If we assume that $\pi$ and $\lambda$ have positive densities w.r.t. a common dominant measure, denoted also by $\pi$ and $\lambda$ respectively, then the *self-normalized importance sampling* (SNIS, see [61]) estimator of $\pi(f)$ is given by

$$\widehat{\pi}_N(f) = \sum_{i=1}^N \omega_N^i f(X^i) , \tag{1}$$

where $X^{1:N} \overset{\text{i.i.d.}}{\sim} \lambda$, and $\omega_N^i = w(X^i)/\sum_{j=1}^N w(X^j)$ are the self-normalized importance weights. Note that computing $\omega_N^i$ does not require the knowledge of $\lambda(w)$. The main problem in the practical applications of IS is the choice of the proposal distribution $\lambda$. The representation $\pi(\mathrm{d}x) = w(x)\lambda(\mathrm{d}x)/\lambda(w)$ implies that the support of $\lambda$ covers the support of $\pi$. At the same

---
**Algorithm 1:** Single stage of i-SIR algorithm with independent proposals
---

**1 Procedure** i-SIR $(Y_k, \lambda)$**:**

    **Input** : Previous state $Y_k$; proposal distribution $\lambda$;

    **Output** : New state $Y_{k+1}$; pool of proposals $X_{k+1}^{2:N} \sim \lambda$;

**2**    Set $X_{k+1}^1 = Y_k$, draw $X_{k+1}^{2:N} \sim \lambda$; **for** $i \in [N]$ **do**

**3**         compute the normalized weights $\omega_{i,k+1} = w(X_{k+1}^i)/\sum_{\ell=1}^N w(X_{k+1}^\ell)$;

**4**    Draw the proposal index $I_{k+1} \sim \mathrm{Cat}(\omega_{1,k+1}, \ldots, \omega_{N,k+1})$;

**5**    Set $Y_{k+1} := X_{k+1}^{I_{k+1}}$.

---

time, too large variance of $\lambda$ is obviously detrimental to the quality of (1). This suggests *adaptive importance sampling* techniques (discussed in [16]), which involve learning the proposal $\lambda$ to improve the quality of (1). We return to this idea in section 3.

IS -based techniques can also be used to draw an (approximate) sample from $\pi$. For instance, Sampling Importance Resampling (SIR, [68]) follows the steps:

1. Draw $X^{1:N} \overset{\text{i.i.d.}}{\sim} \lambda$;
2. Compute the self-normalized importance weights $\omega_N^i = w(X^i)/\sum_{\ell=1}^N w(X^\ell), i \in \{1, \ldots, N\}$;
3. Select $M$ samples $Y^{1:M}$ from the set $X^{1:N}$ choosing $X^i$ with probability $\omega_N^i$ with replacement.

The drawback of the procedure is that it is only asymptotically valid with $N \to \infty$. Alternatively, SIR can be repeated to define a Markov Chain as in *iterated SIR* (i-SIR), proposed in [73] and also studied in [4, 43, 42, 5]. At each iteration of i-SIR described in Algorithm 1, a candidate pool $X_{k+1}^{2:N}$ is sampled from the proposal and the next state $Y_{k+1}$ is choosen among the candidates and the previous state $X_{k+1}^1 = Y_k$ according to the importance weights. i-SIR shares similarities with the Multiple-try Metropolis (MTM) algorithm [44], but is computationally simpler and exhibits more favorable mixing properties; see Appendix A.1. The Markov chain $\{Y_k, \ k \in \mathbb{N}\}$ generated by i-SIR has the following Markov kernel

$$\mathsf{P}_N(x, \mathsf{A}) = \int \delta_x(\mathrm{d}x^1) \sum_{i=1}^N \frac{w(x^i)}{\sum_{j=1}^N w(x^j)} \mathbf{1}_\mathsf{A}(x^i) \prod_{j=2}^N \lambda(\mathrm{d}x^j).$$

Interpreting i-SIR as a systematic-scan two-stage Gibbs sampler (see Appendix A.2 for more details), it follows easily that the Markov kernel $\mathsf{P}_N$ is reversible w.r.t. the target $\pi$, Harris recurrent and ergodic (see Theorem 5). Provided also that $|w|_\infty < \infty$, it was shown in [5] that the Markov kernel $\mathsf{P}_N$ is uniformly geometrically ergodic. Namely, for any initial distribution $\xi$ on $(\mathbb{X}, \mathcal{X})$ and $k \in \mathbb{N}$,

$$\|\xi\mathsf{P}_N^k - \pi\|_{\mathrm{TV}} \leq \kappa_N^k \quad \text{with} \ \epsilon_N = \frac{N-1}{2\mathrm{L} + N - 2}, \mathrm{L} = |w|_\infty/\lambda(w) \ , \ \text{and} \ \kappa_N = 1 - \epsilon_N. \quad (2)$$

We provide a simple direct proof of (2) in Appendix B.1. Yet, note that the bound (2) relies significantly on the restrictive condition that weights are uniformly bounded $|w|_\infty < \infty$. Moreover, even when this condition is satisfied, the rate $\kappa_N$ can be close to 1 when the dimension $d$ is large.[1] We illustrate this phenomenon on a Gaussian target in Appendix E.2 Figure 7 with an experiment that also contrasts the degradation as dimension grows of the purely global sampler with the robustness of the local-global kernels analyzed in the next section.

## 2.2 Coupling with local kernels: $\mathrm{Ex}^2\mathrm{MCMC}$

After each i-SIR step, we apply a local MCMC kernel R (rejuvenation kernel), with an invariant distribution $\pi$. We call this startegy $\mathrm{Ex}^2\mathrm{MCMC}$ because it combines steps of exploration by i-SIR and steps of exploitation by the local MCMC moves. The resulting algorithm, formulated in Algorithm 2, defines a Markov chain $\{Y_j, \ j \in \mathbb{N}\}$ with Markov kernel $\mathsf{K}_N(x, \cdot) = \mathsf{P}_N\mathsf{R}(x, \cdot) = \int \mathsf{P}_N(x, \mathrm{d}y)\mathsf{R}(y, \cdot)$.

We now present the main theoretical result of this paper on the properties of $\mathrm{Ex}^2\mathrm{MCMC}$. Under rather weak conditions, provided that R is geometrically regular (see [21, Chapter 14]), it is possible

---

[1]Indeed, consider a simple scenario $\pi(x) = \prod_{i=1}^d p(x_i)$ and $\lambda(x) = \prod_{i=1}^d q(x_i)$ for some densities $p(\cdot)$ and $q(\cdot)$ on $\mathbb{R}$. Then it is easy to see that $\mathrm{L} = (\sup_{y \in \mathbb{R}} p(y)/q(y))^d$ grows exponentially with $d$.

---
**Algorithm 2:** Single stage of $\mathrm{Ex^2MCMC}$ algorithm with independent proposals
---
**1** **Procedure** $\mathrm{Ex^2MCMC}$ $(Y_k, \lambda, \mathsf{R})$**:**

    **Input** : Previous state $Y_k$; proposal distribution $\lambda$; rejuvenation kernel $\mathsf{R}$;

    **Output** : New sample $Y_{k+1}$; pool of proposals $X_{k+1}^{2:N} \sim \lambda$;

**2**     $Z_{k+1}, X_{k+1}^{2:N} = \text{i-SIR}(Y_k, \lambda)$;

**3**     Draw $Y_{k+1} \sim \mathsf{R}(Z_{k+1}, \cdot)$.
---

to establish that $\mathrm{Ex^2MCMC}$ remains $V$-uniformly geometrically ergodic even if the weight function $w(x)$ is unbounded.

**Definition 1** ($V$-Geometric Ergodicity). *A Markov kernel* $\mathsf{Q}$ *with invariant probability measure* $\pi$ *is* $V$-*geometrically ergodic if there exist constants* $\rho \in (0,1)$ *and* $M < \infty$ *such that, for all* $x \in \mathbb{X}$ *and* $k \in \mathbb{N}$, $\|\mathsf{Q}^k(x, \cdot) - \pi\|_V \leq M\{V(x) + \pi(V)\}\rho^k$.

In particular, $V$-geometric ergodicity ensures that the distribution of the $k$-th iterate of a Markov chain converges geometrically fast to the invariant probability in $V$-norm, for all starting points $x \in \mathbb{X}$. Here the dependence on the initial state $x$ appears on the right-hand side only in $V(x)$. Denote by $\mathrm{Var}_\lambda[w] = \int \{w(x) - \lambda(w)\}^2 \lambda(\mathrm{d}x)$ the variance of the importance weight functions under the proposal distribution and consider the following assumptions:

**A1.** *(i)* $\mathsf{R}$ *has* $\pi$ *as its unique invariant distribution; (ii) There exists a function* $V: \mathbb{X} \to [1, \infty)$, *such that for all* $r \geq r_\mathsf{R} > 1$ *there exist* $\lambda_{\mathsf{R},r} \in [0,1)$, $\mathsf{b}_{\mathsf{R},r} < \infty$, *such that* $\mathsf{R}V(x) \leq \lambda_{\mathsf{R},r}V(x) + \mathsf{b}_{\mathsf{R},r}\mathbb{1}_{\mathsf{V}_r}$, *where* $\mathsf{V}_r = \{x: V(x) \leq r\}$;

**A2.** *(i) For all* $r \geq r_\mathsf{R}$, $w_{\infty,r} := \sup_{x \in \mathsf{V}_r}\{w(x)/\lambda(w)\} < \infty$ *and (ii)* $\mathrm{Var}_\lambda[w]/\{\lambda(w)\}^2 < \infty$.

A1-(ii) states that $\mathsf{R}$ satisfies a Foster-Lyapunov drift condition for $V$. This condition is fulfilled by most classical MCMC kernels - like Metropolis-Adjusted Langevin (MALA) algorithm or Hamiltonian Monte Carlo (HMC), typically under tail conditions for the target distribution; see [63, 22], and [21, Chapter 2] with the references therein. A2-(i) states that the (normalized) importance weights $w(\cdot)/\lambda(w)$ are upper bounded on level sets of $\mathsf{V}_r$. This is a mild condition: if $\mathbb{X} = \mathbb{R}^d$, and $V$ is norm-like, then the level sets $\mathsf{V}_r$ are compact and $w(\cdot)$ is bounded on $\mathsf{V}_r$ as soon as $\pi$ and $\lambda$ are positive and continuous. A2-(ii) states that the variance of the importance weights is bounded; note that this variance is also equal to the $\chi^2$-distance between the proposal and the target distributions which plays a key role in the non-asymptotic analysis of the performance of IS methods [1, 70].

**Theorem 2.** *Assume A1 and A2. Then, for all* $x \in \mathbb{X}$ *and* $k \in \mathbb{N}$,

$$\|\mathsf{K}_N^k(x, \cdot) - \pi\|_V \leq c_{\mathsf{K}_N}\{\pi(V) + V(x)\}\tilde{\kappa}_{\mathsf{K}_N}^k, \tag{3}$$

*where the constant* $c_{\mathsf{K}_N}$, $\tilde{\kappa}_{\mathsf{K}_N} \in [0,1)$ *are given in the proof. In addition,* $c_{\mathsf{K}_N} = c_{\mathsf{K}_\infty} + O(N^{-1})$ *and* $\tilde{\kappa}_{\mathsf{K}_\infty} = \tilde{\kappa}_{\mathsf{K}_N} + O(N^{-1})$ *with explicit expressions provided in* (13).

The proof of Theorem 2 is provided in Appendix B.2. We stress that in many situations, the mixing rate $\tilde{\kappa}_{\mathsf{K}_N}$ of the $\mathrm{Ex^2MCMC}$ Markov Kernel $\mathsf{K}_N$ is significantly better than the corresponding mixing rate of the local kernel $\mathsf{R}$, provided $N$ is large enough. This is due to the fact that assumptions A1 and A2 do not require to identify the small sets of the rejuvenation kernel $\mathsf{R}$ (see [21, Definition 9.3.5]). At the same time, the quantitative bounds on the mixing rates relies on the constants appearing in the small set condition, see [21, Theorem 19.4.1]. Focusing on MALA (see, e.g. [66]) as the rejuvenation kernel $\mathsf{R}$ we detail bounds in Appendix C and prove in Theorem 20 that the ratio of mixing times of $\mathsf{K}_N$ is typically very favorable compared to MALA provided that $N$ is large enough.

## 3 Adaptive version: $\mathrm{FlEx^2MCMC}$

The performance of proposal-based samplers depends on the distribution of importance weights which is related to the similarity of the proposal and target distributions[2]. Therefore, yet another strategy to improve sampling performance is to select the proposal distribution $\lambda$ from a family of parameterized distributions $\{\lambda_\theta\}$ and fit the parameter $\theta \in \Theta = \mathbb{R}^q$ to the target $\pi$, for example, by minimizing a Kullback-Leibler divergence (KL) [57, 2, 50] or matching moments [59]. In *adaptive*

---
[2]more specifically, it depends on the the quantities appearing in A2, namely, the maximum of the importance weight on a level set of the drift function for the local kernel $\mathsf{R}$ and the variance of the importance weights under the proposal

*MCMCs*, parameter adaptation is performed along the MCMC run [6, 9, 64]. In this section we propose an adaptive version of $\text{Ex}^2\text{MCMC}$, which we call $\text{FlEx}^2\text{MCMC}$.

**Normalizing flow proposal.** A flexible way to parameterize proposal distributions is to combine a tractable distribution $\varphi$ with an invertible parameterized transformation. Let $T : \mathbb{X} \mapsto \mathbb{X}$ be a $C^1$ diffeomorphism. We denote by $T\#\varphi$ the push-forward of $\varphi$ under $T$, that is, the distribution of $X = T(Z)$ with $Z \sim \varphi$. Assuming that $\varphi$ has a p.d.f. (also denoted $\varphi$), the corresponding push-forward density (w.r.t. the Lebesgue measure) is given by $\lambda_T(y) = \varphi\big(T^{-1}(y)\big) \, \text{J}_{T^{-1}}(y)$, where $\text{J}_T$ denotes the Jacobian determinant of $T$. The parameterized family of diffeomorphisms $\{T_\theta\}_{\theta \in \Theta}$ defines a family of distributions $\{\lambda_{T_\theta}\}_{\theta \in \Theta}$, denoted for simplicity as $\{\lambda_\theta\}_{\theta \in \Theta}$. This construction is called a *normalizing flow* (NF) and a great deal of work has been devoted to ways of parameterizing invertible flows $T_\theta$ with neural networks; see [40, 55] for reviews.

**Simultaneous learning and sampling.** As with adaptive MCMC methods, the parameters of a NF proposal are learned for the global proposal during sampling, see also [25]. We work with $M$ copies of the Markov chains $\{(Y_k[j], X_k^{1:N}[j])\}_{k \in \mathbb{N}^*}$ indexed by $j \in \{1, \ldots, M\}$. At each step $k \in \mathbb{N}^*$, each copy is sampled as in $\text{Ex}^2\text{MCMC}$ using the NF proposal, independently from the other copies, but conditionally to the the current value of the parameters $\theta_{k-1}$. We then adapt the parameters by taking steps of gradient descent on a convex combination of the *forward* KL, $\text{KL}(\pi\|\lambda_\theta) = \int \pi(x) \log(\pi(x)/\lambda_\theta(x)) \mathrm{d}x$ and the backward KL $\text{KL}(\lambda_\theta\|\pi) = \int \lambda_\theta(x) \log(\pi(x)/\lambda_\theta(x)) \mathrm{d}x = \int \varphi(z) \log w_\theta \circ T_\theta(z) \mathrm{d}z$. Let $\{\gamma_k, \ k \in \mathbb{N}\}$ be a sequence of nonnegative stepsizes and $\{\alpha_k, \ k \in \mathbb{N}\}$ be a nondecreasing sequence in $[0, 1]$ with $\alpha_\infty = \lim_{k \to \infty} \alpha_k$. The update rule is $\theta_k = \theta_{k-1} + \gamma_k M^{-1} \sum_{j=1}^M H(\theta_{k-1}, X_k^{1:N}[j], Z_k^{2:N}[j])$ where $H(\theta, x^{1:N}, z^{2:N}) = \alpha_k H^f(\theta, x^{1:N}) + (1 - \alpha_k) H^b(\theta, z^{2:N})$ with

$$H^f(\theta, x^{1:N}) = \sum_{\ell=1}^N \frac{w_\theta(x^\ell)}{\sum_{i=1}^N w_\theta(x^i)} \nabla_\theta \log \lambda_\theta(x^\ell) \,, \quad w_\theta(x) = \pi(x)/\lambda_\theta(x) \,, \tag{4}$$

$$H^b(\theta, z^{2:N}) = -\frac{1}{N-1} \sum_{\ell=2}^N \{\nabla_\theta \log \pi \circ T_\theta(z^\ell) + \nabla_\theta \log \text{J}_{T_\theta}(z^\ell)\} \,. \tag{5}$$

Note that we use a Rao-Blackwellized estimator of the gradient of the forward KL (4) where we fully recycle all the $N$ candidates sampled at each iteration of i-SIR. The quality of this estimator is expected to improve along the iterations $k$ of the algorithm as the variance of importance weights decreases as the proposal improves. Note also that using only gradients from the backward KL (5) is prone to mode-collapse [57, 54, 50, 25], hence the need for also using gradients from the forward KL $H^f(\theta, x^{1:N})$, which requires the simultaneous sampling from $\pi$. See also Appendix E.5 for further discussions. The $\text{FlEx}^2\text{MCMC}$ algorithm is summarized in Algorithm 3.

Since the parameters of the Markov kernel $\theta_k$ are updated using samples $X_k^{1:N}$ from the chain, $((Y_k, X_k^{1:N}))_{k \in \mathbb{N}}$ is no longer Markovian. This type of problems has been considered in [48, 13, 30, 7] and to prove convergence of the strategy we need to strengthen assumptions compared to the previous section.

**A3.** *There exists a function $W : \mathbb{X} \to \mathbb{R}_+$ such that $\varphi(W^2) = \int W^2(z)\varphi(\mathrm{d}z) < \infty$, and a constant $L < \infty$ such that, for all $\theta, \theta' \in \Theta$ and $z \in \mathbb{X}$, $\|\nabla_\theta \log \pi \circ T_\theta(z) - \nabla_\theta \log \pi \circ T_{\theta'}(z)\| \leq L\|\theta - \theta'\|W(z)$ and $\|\nabla_\theta \log \text{J}_{T_\theta}(z) - \nabla_\theta \log \text{J}_{T_{\theta'}}(z)\| \leq L\|\theta - \theta'\|W(z)$.*

**A4.** *(i) For all $d \geq d_{\mathsf{R}}$, $w_{\infty,d} = \sup_{\theta \in \Theta} \sup_{x \in \mathsf{V}_d} w_\theta(x)/\lambda_\theta(w_\theta) < \infty$ and (ii) $\sup_{\theta \in \Theta} \text{Var}_\varphi(w_\theta \circ T_\theta)/\{\lambda_\theta(w_\theta)\}^2 < \infty$.*

A3 is a continuity condition on the NF push-forward density w.r.t. its parameters $\theta$. A4 implies that the Markov kernel $\mathsf{K}_{N,\theta} = \mathsf{P}_{N,\theta}\mathsf{R}$ satisfies a drift and minorization condition uniform in $\theta$.

**Theorem 3** (simplified)**.** *Assume A 1-A 3-A 4 and that $\sum_{k=0}^\infty \gamma_k = \infty$, $\sum_{k=0}^\infty \gamma_k^2 < \infty$ and $\lim_{k \to \infty} \alpha_k = \alpha_\infty$. Then, w.p. 1, the sequence $\{\theta_k, \ k \in \mathbb{N}\}$ converges to the set $\{\theta \in \Theta, 0 = \alpha_\infty \nabla \text{KL}(\pi\|\lambda_\theta) + (1 - \alpha_\infty)\nabla \text{KL}(\lambda_\theta\|\pi)\}$.*

Theorem 3 proves the convergence of the learning of parameters $\theta$ to a stationary point of the loss. The proof is postponed to Appendix D. Note that once the proposal learning has converged, $\text{FlEx}^2\text{MCMC}$ boils back to $\text{Ex}^2\text{MCMC}$ with a fixed learned proposal. Our experiments show that adaptivity can significantly speed up mixing for i-SIR, especially for distributions with complex geometries and that the addition of a rejuvenation kernel further improves samples quality.

**Algorithm 3:** Single stage of FlEx$^2$MCMC. Steps of Ex$^2$MCMC use the NF proposal with parameters $\theta_k$. Step 4 updates the parameters using the gradient estimate obtained from all the chains.

**Input** : weights $\theta_k$, batch $Y_k[1:M]$
**Output** : new weights $\theta_{k+1}$, batch $Y_{k+1}[1:M]$

1 **for** $j \in [M]$ **do**
2 $\quad \big\lfloor \; Y_{k+1}[j] = \text{Ex}^2\text{MCMC}\,(Y_k, T_{\theta_k}\#\varphi, \mathsf{R})$
3 Draw $Z[1:M] \sim \varphi$.
4 Update $\theta_k = \theta_{k-1} + \gamma_k M^{-1} \sum_{j=1}^{M} H(\theta_{k-1}, X_k^{1:N}[j], Z_k^{2:N}[j])$

## 4 Related Work

The possibility to parametrize very flexible probabilistic models with neural networks thanks to deep learning has rekindled interest in adapting MCMC kernels; see e.g. [72, 36, 2, 53, 33]. While significant performance gain were found in problems of moderate dimensions, these learning-based methods were found to suffer from increasing dimensions as fitting models accurately becomes more difficult [19, 46]. Similarly to FlEx$^2$MCMC, a few work proposed adaptive algorithms that alternates between global and local MCMC moves to ensure ergodicity without requiring a perfect learning of the proposal[59, 25]. More precisely, [59] focused on multimodal distributions and analysed a mode jumping algorithm using proposals parametrized as mixture of simple distributions. While [25], closer to this work, introduced a combination of a local and a global sampler leveraging normalizing flows with a more classical choice for the global sampler: independent Metropolis-Hasting (IMH) instead of i-SIR. The present work builds on these previous propositions of combinations of local and global sampler by clarifying the reasons of their effectiveness through entirely novel detailed mathematical and empirical analyses. We chose to focus on i-SIR with an adaptive proposal as the global sampler since (i) the learning component allows to tackle high-dimensional targets, (ii) theoretical guarantees can be obtained for i-SIR whereas IMH is more difficult to analyze, (iii) IMH and i-SIR (as a multiple-try MCMC) are expected to have similar performances for comparable computational budget [11] but IMH is sequential where i-SIR can be parallelized by increasing the number $N$ of proposals per iteration.

Another line of work exploits both normalizing flows and common local MCMC kernels for sampling [57, 36, 54, 77], yet following the different paradigm of using the flow as a reparametrization map, a method sometimes referred to as neural transport: the flow $T$ is trained to transport a simple distribution $\varphi$ near $\pi$, which is equivalent to bringing $T^{-1}\#\pi$ (the pushforward of the original target distribution $\pi$ by the inverse flow $T^{-1}$) close to $\varphi$. If $\varphi$ is simple enough to be efficiently sampled by local samplers, the hope is that local samplers can also obtain high-quality samples of $T^{-1}\#\pi$ – samples which can be transported back through $T$ to obtain samples of $\pi$. This method attempts to reparametrize the space to disentangle problematic geometries for local kernels. Yet, it is unclear what will happen in the tails of the distribution for which the flow is likely poorly learned. Furthermore, in order to derive an ergodicity theory for these transported samplers, [57] necessitated substantial constraints on maps (see section 2.2.2.).

## 5 Numerical experiments

We provide the code to reproduce the experiments below at https://github.com/svsamsonov/ex2mcmc_new.

### 5.1 Synthetic examples

**Multimodal distributions.** Let us start with a toy example highlighting differences between purely global i-SIR, purely local MALA and Ex$^2$MCMC combining both. We consider sampling from a mixture of 3 equally weighted Gaussians in dimension $d = 2$. In Figure 1a, we compare single chains produced by each algorithms. The global proposal is a wide Gaussian, with pools of $N = 3$ candidate. The MALA stepsize is chosen to reach a target acceptance rate of $\sim 0.67$. This simple experiment illustrates the drawbacks of both approaches: i-SIR samples reach all the modes of the target, but the chains often get stuck for several steps hindering variability. MALA allows for better local exploration of each particular mode, yet it fails to cover all the target support. Meanwhile, Ex$^2$MCMC retains the benefits of both methods, combining the i-SIR-based global exploration with MALA-based local exploration.

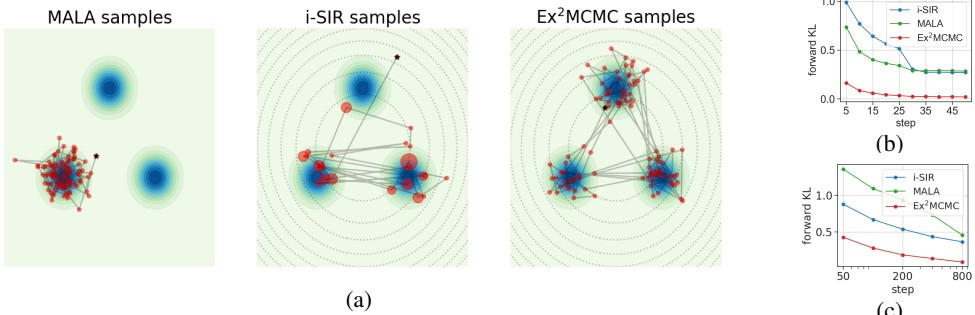

Figure 1: (a) – Single chain mixing visualization. – Blue color levels represent the target 2d density. Random chain initialization is noted in black, 100 steps are plotted per sampler: the size of each red dot corresponds to the number of consecutive steps the walkers remains at a given location. Note that the variance of the global proposal (dotted countour lines) should be relatively large to cover well all the modes. (b - c) – Inhomogeneous 2d Gaussian mixture. – Quantitative analysis during burn-in of parallel chains (b, $M = 500$ chains KDE) and for after burn-in for single chains statistics (c, $M = 100$ average).

In larger dimensions, an adaptive proposal is necessary. In Appendix E.5 we show that FlEx$^2$MCMC can mix between modes of a $50d$ Gaussian mixture, provided that the rough location of all the modes is known and used to initialize walkers. We also stress the robustness of the on-the-fly training exploiting running MCMC chains to evaluate the forward KL term of the loss.

To illustrate further the performance of the combined kernel, we keep the $2d$ target mixture model yet assigning the uneven weights $(2/3, 1/6, 1/6)$ to the 3 modes. We start $M$ chains drawing from the initial distribution $\xi \sim \mathcal{N}(0, 4\,\mathrm{I}_d)$ and use the same hyper-parameters as above. In Figure 1b we provide a simple illustration to the statement (2) and Theorem 2, namely we compare the target density to the instantaneous distributions for each sampler propagating $\xi$ during burn-in steps. As MALA does not mix easily between modes, the different statistical weights of the different modes can hardly be rendered in few iterations and KL and TV distances stalls after a few iterations. i-SIR can visit the different modes, yet it does not necessarily move at each step which slows down its covering of the modes full support, which again shows in the speed of decrease of the TV and KL. Overcoming both of these shortcomings, Ex$^2$MCMC instantaneous density comes much closer to the target. Finally, Figure 1c evaluates the same metrics yet for the density estimate obtained with single chain samples after burn-in. Results demonstrate once again the superiority of Ex$^2$MCMC. Further details on these experiments can be found in Appendix E.3.

**Distributions with complex geometry.** Next, we turn to highly anisotropic distributions in high dimensions. Following [52] and [32], we consider the *funnel* and the *banana-shape* distributions. We remind densities in Appendix E.6 along with providing experiments details. For $d \in [10; 200]$, we run i-SIR, MALA, Ex$^2$MCMC, FlEx$^2$MCMC, adaptive i-SIR (using the same proposal as FlEx$^2$MCMC, but without interleaved local steps) and the versatile sampler NUTS [35] as a baseline. Here the parameter adaptation for FlEx$^2$MCMC is performed in a pre-run and parameters are frozen before sampling. For the adaptive samplers, a simple RealNVP-based normalizing flow [20] is used such that total running times, including training, are comparable with NUTS. For Ex$^2$MCMC and i-SIR the global proposal is a wide Gaussian with a pool of $N = 2000$ candidates drawn at each iteration. For MALA we tune the step size in order to keep acceptance rate approximately at 0.5. We report the average sliced TV distance and ESS in Figure 2 (see Appendix E.1 for metrics definition). In most cases, FlEx$^2$MCMC is the most reliable algorithm. The only exception is at very high dimension for the banana where NUTS performs the best: in this case, tuning the flow to learn tails in high-dimension faithfully was costly such that we proceeded to an early stopping to maintain comparability with the baseline. Remarkably, FlEx$^2$MCMC compensates significantly for the imperfect flow training, improving over adaptive-i-SIR, but NUTS eventually performs better. Conversely, for the funnel, most of the improvement comes from well-trained proposal flow, leading to similar behaviors of adaptive i-SIR and FlEx$^2$MCMC, while both algorithms clearly outperforms NUTS in terms of metrics.

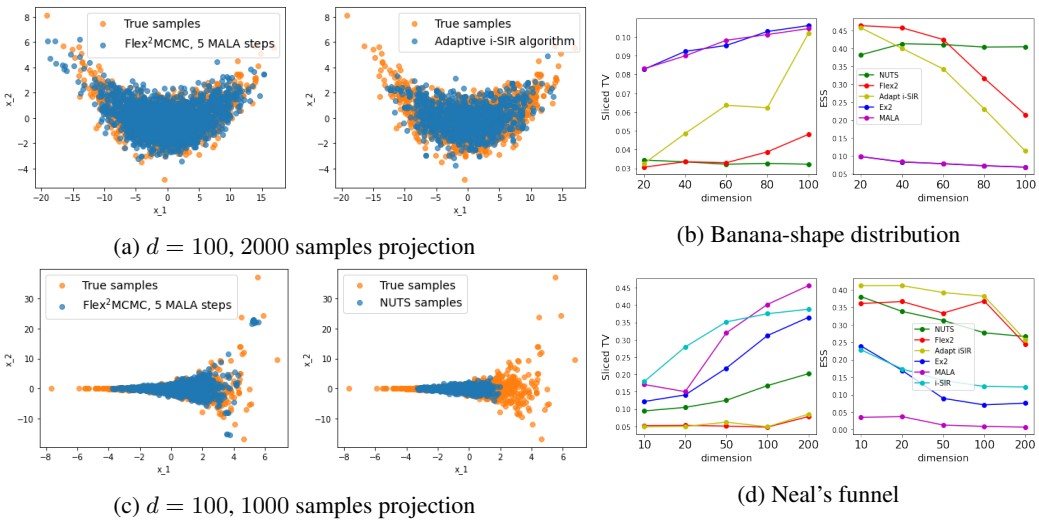

(a) $d = 100$, 2000 samples projection

(b) Banana-shape distribution

(c) $d = 100$, 1000 samples projection

(d) Neal's funnel

Figure 2: Anisotropic Funnel and Banana-shape distributions – (a) and (b) visualize samples projected onto the first 2 coordinates of tested algorithms (blue) versus true samples obtained by reparametrization (orange). (c) and (d) compare Sliced Total Variation and Effective Sample Size as a function of dimension. i-SIR is removed from (b) as corresponding metrics for $d > 20$ are significantly worse.

## 5.2 Sampling from GANs as Energy-based models (EBMs)

Generative adversarial networks (GANs [27]) are a class of generative models defined by a pair of a generator network $G$ and a discriminator network $D$. The generator $G$ takes a latent variable $z$ from a prior density $p_0(z)$, $z \in \mathbb{R}^d$, and generates an observation $G(z) \in \mathbb{R}^D$ in the observation space. The discriminator takes a sample in the observation space and aims to discriminate between true examples and false examples produced by the generator. Recently, it has been advocated to consider GANs as Energy-Based Models (EBMs) [75, 17]. Following [17], we consider the EBM model induced by the GAN in latent space. Recall that an EBM is defined by a Boltzmann distribution $p(z) = \mathrm{e}^{-E(z)}/Z$, $z \in \mathbb{R}^d$, where $E(z)$ is the energy function and Z is the normalizing constant. Note that Wasserstein GANs also allow for an energy-based interpretation (see [17]), although the interpretation of the discriminator in this case is different. The energy function is given by

$$E_{JS}(z) = -\log p_0(z) - \mathrm{logit}\big(D(G(z))\big), \quad E_W(z) = -\log p_0(z) - D(G(z)), \quad z \in \mathbb{R}^d, \quad (6)$$

for the vanilla Jensen-Shannon and Wasserstein GANs, respectively. Here $\mathrm{logit}(y)$, $y \in (0, 1)$ is the inverse of the sigmoid function and $p_0(z) = \mathcal{N}(0, \mathrm{I}_d)$.

**MNIST results.** We consider a simple Jensen-Shannon GAN model trained on the MNIST dataset with latent space dimension $d = 2$. We compare samples obtained by i-SIR, MALA, and $\mathrm{Ex}^2\mathrm{MCMC}$ from the energy-based model associated with $E_{JS}(z)$, see (6). We use a wide normal distribution as the global proposal for i-SIR and $\mathrm{Ex}^2\mathrm{MCMC}$, and pools of candidates at each iteration $N = 10$. The step-size of MALA is tuned to keep an acceptance rate $\sim 0.5$. We visualize chains of 100 steps in the latent space obtained with each method in Figure 3. Note that the poor agreement between the proposal and the landscape makes it difficult for i-SIR to accept from the proposal and for MALA to explore many modes of the latent distribution, as shown in Figure 3. $\mathrm{Ex}^2\mathrm{MCMC}$ combines effectively global and local moves, encouraging better diversity associated with a better mixing time. The images corresponding to the sampled latent space locations are displayed in Figure 4 and reflect the diversity issue of MALA and i-SIR. Further details and experiments are provided in Appendix E.7.1, including similar results for WGAN-GP [31] and the associated EBM $E_W(z)$.

**Cifar-10 results.** We consider two popular architectures trained on Cifar-10, DC-GAN [60] and SN-GAN [49]. In both cases the dimension of the latent space equals $d = 128$. Together with the non-trivial geometry of the corresponding energy landscapes, the large dimension makes sampling with NUTS unfeasible in terms of computational time. We perform sampling from mentioned GANs as energy-based models using i-SIR, MALA, $\mathrm{Ex}^2\mathrm{MCMC}$, and $\mathrm{FlEx}^2\mathrm{MCMC}$. In i-SIR and $\mathrm{Ex}^2\mathrm{MCMC}$ we use the prior $p_0(z)$ as a global proposal with a pool of $N = 10$ candidates. For $\mathrm{FlEx}^2\mathrm{MCMC}$ we perform training and sampling simultaneously. Implementation details are

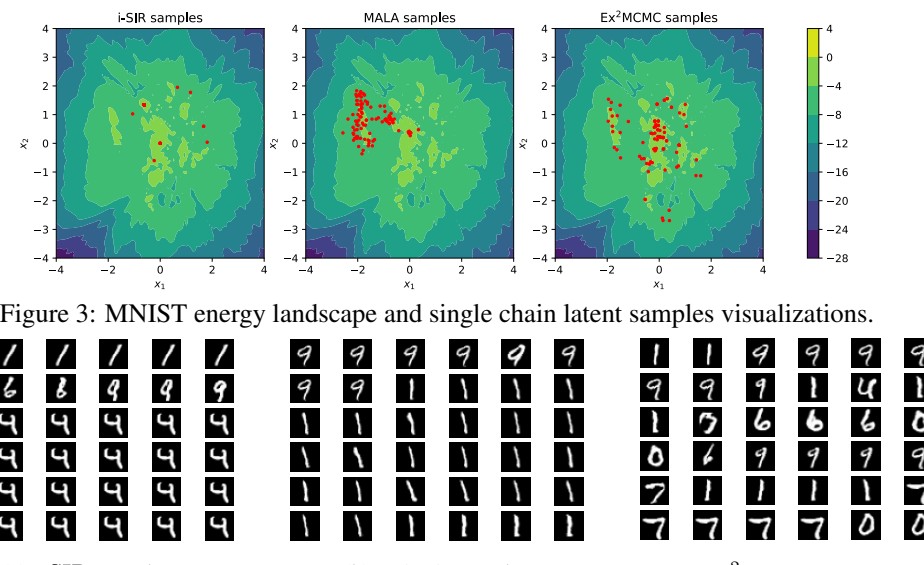

Figure 3: MNIST energy landscape and single chain latent samples visualizations.

(a) i-SIR samples        (b) MALA samples        (c) Ex$^2$MCMC samples

Figure 4: MNIST samples visualization. – Single chains run, sequential steps.

provided in Appendix E.7.2. To evaluate sampling quality, we report the values of the energy function $E(z)$, averaged over 500 independent runs of each sampler. We also visualize the inception score (IS) dynamics calculated over 10000 independent trajectories. We present the results in Figure 5 together with the images produced by each sampler. Note that Ex$^2$MCMC and FlEx$^2$MCMC reach low level of energies faster than other methods, and reach high IS samples in a limited number of iterations. Visualizations indicate that MALA is unlikely to escape the mode of the distribution $p(z)$ it started from, while i-SIR and Ex$^2$MCMC/FlEx$^2$MCMC better explores the target support. However, global move appear to become more rare after some number of iterations for Ex$^2$MCMC/FlEx$^2$MCMC, which then exploit a particular mode with MALA steps. We here hit the following limitation: i-SIR remains at relatively high-energies, failing to explore well modes basins but still accepting global moves, while Ex$^2$MCMC/FlEx$^2$MCMC explores well modes basins but eventually remains trapped. We predict that improving further the quality of the FlEx$^2$MCMC proposal by scaling the normalizing flow architecture would allow for more global moves. See Appendix E.7.2 for additional experiments (including ones with SN-GAN), FID dynamics, and visualizations.

## 6   Conclusions and further research directions

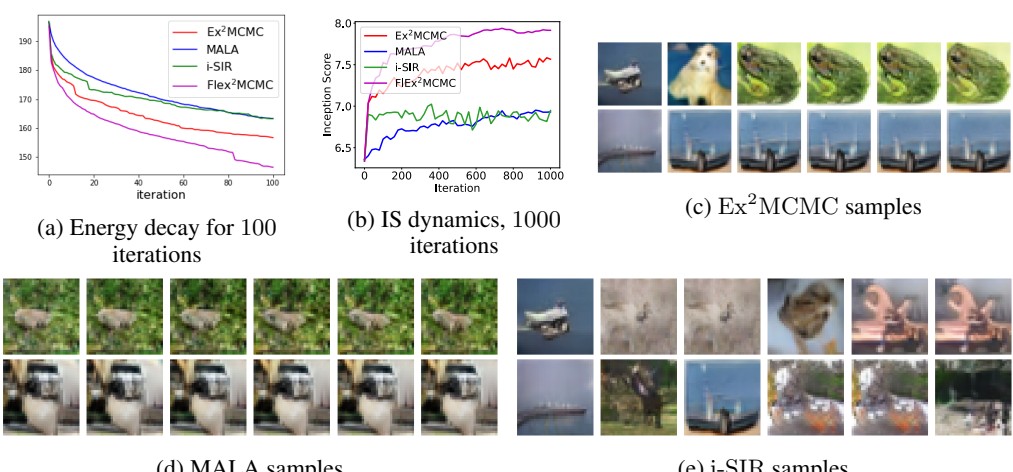

(a) Energy decay for 100 iterations

(b) IS dynamics, 1000 iterations

(c) Ex$^2$MCMC samples

(d) MALA samples            (e) i-SIR samples

Figure 5: Cifar-10 energy and sampling results with DC-GAN architecture. Along the horizonthal lines we visualize each 10th sample from a single trajectory.

The present paper examines the benefits of combining local and global samplers. From a theoretical point of view, we show that global samplers are more robust when coupled with local samplers. Namely, a $V$-geometric ergodicity is obtained for the $\mathrm{Ex}^2\mathrm{MCMC}$ kernel under minimal assumptions. Meanwhile, the global samplers drives exploration when properly adjusted. Therefore, we also describe the adaptive version $\mathrm{FlEx}^2\mathrm{MCMC}$ of the strategy involving the learning of a global proposal parametrized by a normalizing flow. We also check for the learning convergence along the adaptive MCMC run. Finally, a series of numerical experiments confirms the superiority of the strategy, including the high-dimensional examples. While the startegy was described and analyzed for the i-SIR global kernel, we note that it would be possible to extend the theory to other independent global samplers. We expect that the benefit of the combination would remain. Further studies of $\mathrm{FlEx}^2\mathrm{MCMC}$, in particular the derivation of its mixing rate, is an interesting direction for future work.

## Acknowledgement

The article was partly prepared within the framework of the HSE University Basic Research Program. This work was partly supported by SCAI Project-ANR-19-CHIA-0002. M.G. acknowledges funding from Hi!Paris. A.D. and E.M. acknowledges support from the Lagrange Mathematics and Computing Research Center.

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
