# A  i-SIR **Algorithm**

## A.1  i-SIR **and Multiple-try Metropolis (MTM) algorithm**

In the MTM algorithm, $N$ i.i.d. sample proposals $\{X_{k+1}^i\}_{i=1}^N$ are drawn from a kernel $\mathsf{T}(y, \cdot)$ in each iteration. In a second step, a sample $Y_{k+1}^*$ is selected with probability proportional to the weights (the exact expression of the weighting weights differs from ours, but this does not change the complexity of the algorithm). In a third step, $N - 1$ i.i.d. proposals are drawn from the kernel $\mathsf{T}(Y_{k+1}^*, \cdot)$ and it is assumed that the move is $Y_{k+1} = Y_{k+1}^*$ with an *generalised M-H* ratio, see [44, eq. 3]. This step is bypassed in i-SIR, reducing the computational complexity by a factor of 2.

## A.2  i-SIR **as a systematic scan two-stage Gibbs sampler**

We analyze a slightly modified version of the i-SIR algorithm, with an extra randomization of the state position. The $k$-th iteration is defined as follows. Given a state $Y_k \in \mathbb{X}$,

(i)  draw $I_{k+1} \in \{1, \ldots, N\}$ uniformly at random and set $X_{k+1}^{I_{k+1}} = Y_k$;

(ii)  draw $X_{k+1}^{1:N \setminus \{I_{k+1}\}}$ independently from the proposal distribution $\lambda$;

(iii)  compute, for $i \in \{1, \ldots, N\}$, the normalized importance weights

$$\omega_{N,k+1}^i = w(X_{k+1}^i) / \sum_{\ell=1}^N w(X_{k+1}^\ell);$$

(iv)  select $Y_{k+1}$ from the set $X_{k+1}^{1:N}$ by choosing $X_{k+1}^i$ with probability $\omega_{N,k+1}^i$.

Thus, compared to the simplified i-SIR algorithm given in the introduction, the state is inserted uniformly at random into the list of candidates instead of being inserted at the first position. Of course, this change has no impact as long as we are interested in integrating functions that are permutation invariant with respect to candidates, which is the case throughout our work. Still, this randomization makes the analysis much more transparent.

In what follows, we show that i-SIR can be interpreted as a systematic-scan two-stage Gibbs sampler sampling, which alternately samples from the full conditionals of the extended target $\varphi_N$, which is carefully defined below in terms of the state and candidate pool. Here we essentially follow the work of [73, 4, 5]. This is formalized by a dual representation of $\varphi_N$, presented below in Theorem 4, which provides the two complete conditionals in question. We introduce the Markov kernel

$$\mathbf{\Lambda}_N(y, \mathrm{d}x^{1:N}) = \frac{1}{N} \sum_{i=1}^N \delta_y(\mathrm{d}x^i) \prod_{j \neq i} \lambda(\mathrm{d}x^j)$$

on $\mathbb{X} \times \mathcal{X}^{\otimes N}$, which probabilistically describes the candidate selection operation in i-SIR. Note that by construction, for each $y \in \mathbb{X}$, $\ell \in \{1, \ldots, N\}$ and nonnegative measurable function $h : \mathbb{X} \to \mathbb{R}^+$,

$$\mathbf{\Lambda}_N h(y) = \int \mathbf{\Lambda}_N(y, \mathrm{d}x^{1:N}) h(x^\ell) = \left(1 - \frac{1}{N}\right) \lambda(h) + \frac{1}{N} h(y).$$

Using the kernel $\mathbf{\Lambda}_N$ we may now define properly the extended target $\varphi_N$ as the probability law

$$\varphi_N(\mathrm{d}(y, x^{1:N})) = \pi(\mathrm{d}y) \mathbf{\Lambda}_N(y, \mathrm{d}x^{1:N}) = \frac{1}{N} \sum_{i=1}^N \pi(\mathrm{d}y) \delta_y(\mathrm{d}x^i) \prod_{j \neq i} \lambda(\mathrm{d}x^j)$$

on $(\mathbb{X}^{N+1}, \mathcal{X}^{\otimes(N+1)})$. Note that since for every $A \in \mathcal{X}$, $\varphi_N(1_{A \times \mathbb{X}}) = \pi(A)$, the target $\pi$ coincides with the marginal of $\varphi_N$ with respect to the state. Moreover, it is easily seen that $\mathbf{\Lambda}_N$ provides the conditional distribution, under $\varphi_N$, of the candidate pool given the state.

On the other hand, using that $\pi(\mathrm{d}y) \delta_y(\mathrm{d}x^i) = w(x^i) \lambda(\mathrm{d}x^i) \delta_{x^i}(\mathrm{d}y) / \lambda(w)$, the marginal distribution $\boldsymbol{\pi}_N$ of $\varphi_N$ with respect to $x^{1:N}$ is given by

$$\boldsymbol{\pi}_N(\mathrm{d}x^{1:N}) = \frac{1}{\lambda(w)} \Gamma_N 1_{\mathbb{X}}(x^{1:N}) \prod_{j=1}^N \lambda(\mathrm{d}x^j), \tag{7}$$

where we have set

$$\Gamma_N(x^{1:N}, \mathrm{d}y) = \sum_{i=1}^{N} w(x^i)\delta_{x^i}(\mathrm{d}y)/N, \quad \Pi_N(x^{1:N}, \mathrm{d}y) = \Gamma_N(x^{1:N}, \mathrm{d}y)/\Gamma_N 1_{\mathbb{X}}(x^{1:N})$$

It is interesting to note that the marginal $\boldsymbol{\pi}_N$ has a probability density function, proportional to $\Gamma_N 1_{\mathbb{X}}(x^{1:N}) = \sum_{i=1}^{N} w(x^i)/N$, with respect to the product measure $\lambda^{\otimes N}$. Using (7), we immediately obtain the following result.

**Theorem 4** (duality of extended target)**.** *For every $N \in \mathbb{N}^*$,*

$$\boldsymbol{\varphi}_N(\mathrm{d}(y, x^{1:N})) = \pi(\mathrm{d}y)\boldsymbol{\Lambda}_N(y, \mathrm{d}x^{1:N}) = \boldsymbol{\pi}_N(\mathrm{d}x^{1:N})\Pi_N(x^{1:N}, \mathrm{d}y).$$

Using this dual representation of $\boldsymbol{\varphi}_N$, i-SIR can be interpreted as a two-stage Gibbs sampler. Given the state $Y_k$, $N$ candidates $X_{k+1}^{1:N}$ are sampled from $\boldsymbol{\Lambda}_N(Y_k, \cdot)$. In a second step, the next state $Y_{k+1}$ is sampled given the current candidates from $\Pi_N(X_{k+1}^{1:N}, \cdot)$. The two-stages Gibbs sampler generates a Markov chain $((Y_k, X_k^{1:N}))_{k \in \mathbb{N}}$ with Markov kernel

$$\mathbf{P}_N((y, x^{1:N}), C) = \int \boldsymbol{\Lambda}_N(y, \mathrm{d}\tilde{x}^{1:N})\Pi_N(\tilde{x}^{1:N}, \mathrm{d}\tilde{y})1_C(\mathrm{d}(y, \tilde{x}^{1:N})), \quad C \in \mathcal{X}^{\otimes(N+1)}.$$

Note that the Markov kernel $\mathbf{P}_N(y, x^{1:N}, \cdot)$ does not depend on $x^{1:N}$, which means that only the state $Y_k$ needs to be stored from one iteration to another. Given a distribution $\boldsymbol{\xi}$ on $(\mathbb{X}^{n+1}, \mathcal{X}^{\otimes(n+1)})$, we denote by $\mathbb{P}_{\boldsymbol{\xi}}$ the distribution of the canonical Markov chain $((Y_k, X_k^{1:N}))_{k \in \mathbb{N}}$ with kernel $\mathbf{P}_N$. With these notations, for any nonnegative measurable function $f : \mathbb{X}^{n+1} \to \mathbb{R}$, we get, for $k \in \mathbb{N}^*$,

$$\mathbb{E}_{\boldsymbol{\xi}}\left[ f(Y_k, X_k^{1:N}) \,\middle|\, \mathcal{F}_{k-1} \right] = \int \mathbf{P}_N((Y_{k-1}, X_{k-1}^{1:N}), \mathrm{d}(y, x^{1:N}))f(x^{1:N}) = \mathbf{P}_N f(Y_{k-1}, X_{k-1}^{1:N}).$$

The systematic scan two-stages Gibbs sampler is one of the MCMC algorithm structures that has given rise to many works. We summarize in the theorem below the important properties of this sampler; see [45], [61, Chapter 9], [3] and the references therein.

**Theorem 5.** *Assume that for any $y \in \mathbb{X}$, $w(y) > 0$. Then,*

- *The Markov kernel $\mathbf{P}_N$ is Harris recurrent and ergodic with unique invariant distribution $\boldsymbol{\varphi}_N$.*
- *The Markov kernel $\mathsf{P}_N$ is reversible w.r.t. $\pi$, Harris recurrent and ergodic.*

The proof follows from [61, Theorem 9.6, Lemma 9.11]. The following theorem establishes the unbiasedness of the estimator $\Pi_N f(X^{1:N})$ under $\boldsymbol{\varphi}_N$.

**Theorem 6.** *For every $N \in \mathbb{N}^*$ and $\pi$-integrable function $f$,*

$$\pi(f) := \int \Pi_N f(x^{1:N})\boldsymbol{\pi}_N(\mathrm{d}x^{1:N}) = \int \Pi_N f(x^{1:N})\pi(\mathrm{d}x^1)\prod_{j=2}^{N} \lambda(\mathrm{d}x^j).$$

*Proof.* Using (7) we get

$$\int \boldsymbol{\pi}_N(\mathrm{d}x^{1:N})\Pi_N f(x^{1:N}) = \int \frac{1}{N\lambda(w)} \sum_{\ell=1}^{N} w(x^\ell)\Pi_N f(x^{1:N}) \prod_{j=1}^{N} \lambda(\mathrm{d}x^j)$$

$$= \frac{1}{N\lambda(w)} \int \sum_{i=1}^{N} w(x^i)f(x^i) \prod_{j=1}^{N} \lambda(\mathrm{d}x^j) = \pi(f),$$

and the first identity follows. The second identity stems from the fact that the function $\Pi_N f(x^{1:N})$ is invariant under permutation. $\qquad\square$

# B  Proofs of Section 2

## B.1  Uniform geometric ergodicity of the i-SIR Markov kernel

Here we provide a simple direct proof of the bound (2). We preface the proof by a technical lemma.

**Lemma 7.** *Let $Y^{1:M}$ be $M$ independent random variables, satisfying $\mathbb{E}[Y_i] = 1$, and $\mathrm{Var}[Y_i] < \infty$ for $i \in \{1, \ldots, M\}$. Then, for $S_M = \sum_{i=1}^{M} Y_i$ and $a, b > 0$*

$$\mathbb{E}\left[(a + bS_M)^{-1}\right] \leq (a + bM/2)^{-1} + (4/a)\,\mathrm{Var}[S_M]/M^2 \,.$$

*Proof.* Let $K \geq 0$. Then we get

$$\frac{1}{a + bS_M} = \frac{1}{a + bS_M}\mathbf{1}\{S_M < K\} + \frac{1}{a + bS_M}\mathbf{1}\{S_M \geq K\} \leq \frac{1}{a + bK} + \frac{1}{a}\mathbf{1}\{S_M < K\}$$

and in particular, $\mathbb{E}[(a + bS_M)^{-1}] \leq (a + bK)^{-1} + a^{-1}\mathbb{P}(S_M < K)$. By Markov's inequality,

$$\mathbb{P}(S_M < K) = \mathbb{P}(S_M - M < -(M - K)) \leq \frac{\mathrm{Var}[S_M]}{(M - K)^2}$$

In particular, for $K = M/2$, we have $\mathbb{P}(S_M < K) \leq 4\,\mathrm{Var}[S_M]/M^2$. $\qquad\square$

*Proof of* (2). For $(x, \mathsf{A}) \in \mathbb{X} \times \mathcal{X}$, we get

$$\mathsf{P}_N(x, \mathsf{A}) = \int \delta_x(\mathrm{d}x^1) \sum_{i=1}^{N} \frac{w(x^i)}{\sum_{j=1}^{N} w(x^j)} \mathbf{1}_{\mathsf{A}}(x^i) \prod_{j=2}^{N} \lambda(\mathrm{d}x^j)$$

$$= \int \frac{w(x)}{w(x) + \sum_{j=2}^{N} w(x^j)} \mathbf{1}_{\mathsf{A}}(x) \prod_{j=2}^{N} \lambda(\mathrm{d}x^j) + \int \sum_{i=2}^{N} \frac{w(x^i)}{w(x) + \sum_{j=2}^{N} w(x^j)} \mathbf{1}_{\mathsf{A}}(x^i) \prod_{j=2}^{N} \lambda(\mathrm{d}x^j)$$

$$\geq \sum_{i=2}^{N} \int \frac{w(x^i)}{w(x) + w(x^i) + \sum_{j=2, j\neq i}^{N} w(x^j)} \mathbf{1}_{\mathsf{A}}(x^i) \prod_{j=2}^{N} \lambda(\mathrm{d}x^j)$$

$$\overset{(a)}{\geq} \sum_{i=2}^{N} \int \pi(\mathrm{d}x^i) \mathbf{1}_{\mathsf{A}}(x^i) \int \frac{\lambda(w)}{w(x) + w(x^i) + \sum_{j=2, j\neq i}^{N} w(x^j)} \prod_{j=2, j\neq i}^{N} \lambda(\mathrm{d}x^j) \,. \tag{8}$$

Here in (a) we used Fubini's theorem together with $w(x)\lambda(\mathrm{d}x) = \pi(\mathrm{d}x)\lambda(w)$. Finally, since the function $f \colon z \mapsto (z + a)^{-1}$ is convex on $\mathbb{R}_+$ and $a > 0$, we get for $i \in \{2, \ldots, N\}$,

$$\int \frac{\lambda(w)}{w(x) + w(x^i) + \sum_{j=2, j\neq i}^{N} w(x^j)} \prod_{j=2, j\neq i}^{N} \lambda(\mathrm{d}x^j)$$

$$\geq \frac{\lambda(w)}{\int w(x) + w(x^i) + \sum_{j=2, j\neq i}^{N} w(x^j) \prod_{j=2, j\neq i}^{N} \lambda(\mathrm{d}x^j)}$$

$$\geq \frac{\lambda(w)}{w(x) + w(x^i) + (N - 2)\lambda(w)} \geq \frac{1}{2\mathrm{L} + N - 2} \,.$$

With the bound above we obtain the inequality

$$\mathsf{P}_N(x, \mathsf{A}) \geq \pi(\mathsf{A}) \times \frac{N - 1}{2\mathrm{L} + N - 2} = \epsilon_N \pi(\mathsf{A}) \,. \tag{9}$$

This means that the whole space $\mathbb{X}$ is $(1, \epsilon_N\pi)$-small (see [21, Definition 9.3.5]). Since $\mathsf{P}_N(x, \cdot)$ and $\pi$ are probability measures, (9) implies

$$\|\mathsf{P}_N(x, \cdot) - \pi\|_{\mathrm{TV}} = \sup_{\mathsf{A} \in \mathcal{X}} |\mathsf{P}_N(x, \mathsf{A}) - \pi(\mathsf{A})| \leq 1 - \epsilon_N = \kappa_N \,.$$

The statement follows from [21, Theorem 18.2.4] applied with $m = 1$. $\qquad\square$

## B.2 Proof of Theorem 2

We preface the proof with some preparatory lemmas.

**Lemma 8.** *Let* $K \subset \mathbb{X}$*, such that* $w_{\infty,K} := \sup_{x \in K}\{w(x)/\lambda(w)\} < \infty$ *and* $\pi(K) > 0$*. Then, for all* $(x, A) \in K \times \mathcal{X}$*, we get that*
$$P_N(x, A) \geq \epsilon_{N,K}\pi_K(A) ,$$
*with* $\epsilon_{N,K} = (N-1)\pi(K)/[2w_{\infty,K} + N - 2]$ *and* $\pi_K(A) = \pi(A \cap K)/\pi(K)$*.*

Note that if the weight function $w$ is upper semi-continuous, then for any compact $K$, $w_{\infty,K} = \sup_{x \in K} w(x) < \infty$. Moreover, $\lim_{N \to \infty} \epsilon_{N,K} = \pi(K)$.

*Proof.* Let $(x, A) \in \mathbb{X} \times \mathcal{X}$. Then, using the lower bound (8), we obtain

$$P_N(x, A) \geq \sum_{i=2}^{N} \int \pi(\mathrm{d}x^i)1_A(x^i) \int \frac{\lambda(w)}{w(x) + w(x^i) + \sum_{j=2,j\neq i}^{N} w(x^j)} \prod_{j=2,j\neq i}^{N} \lambda(\mathrm{d}x^j)$$

$$\geq (N-1)\int \pi(\mathrm{d}y)1_A(y)\frac{1}{w(x)/\lambda(w) + w(y)/\lambda(w) + N - 2} ,$$

where the last inequality follows from Jensen's inequality and the convexity of the function $z \mapsto (z+a)^{-1}$ on $\mathbb{R}_+$. We conclude by noting that

$$P_N(x, A) \geq (N-1)\int \pi(\mathrm{d}y)1_{A\cap K}(y)\frac{1}{w(x)/\lambda(w) + w(y)/\lambda(w) + N - 2}$$

$$\geq \frac{N-1}{2w_{\infty,K} + N - 2}\int \pi(\mathrm{d}y)1_{A\cap K}(y) = \frac{(N-1)\pi(K)}{2w_{\infty,K} + N - 2}\pi_K(A) .$$

$\square$

**Lemma 9.** *Assume A1. Then for all* $x \in \mathbb{X}$*, any function* $V : \mathbb{X} \to [1, \infty)$ *with* $\pi(V) < \infty$*,* $\lambda(V) < \infty$*, and* $N \geq 3$*, it holds that*

$$P_N V(x) \leq V(x) + b_{P_N} , \tag{10}$$

*where* $b_{P_N}$ *is given in* (12)*.*

Note that

$$b_{P_\infty} := \lim_{N \to \infty} b_{P_N} = 2\pi(V) + 4\operatorname{Var}_\lambda[w]/\lambda(V) . \tag{11}$$

*Proof.* Note first that

$$P_N V(x) = V(x)\int \frac{w(x)}{w(x) + \sum_{j=2}^{N} w(x^j)}\prod_{j=2}^{N}\lambda(\mathrm{d}x^j) + \int \sum_{i=2}^{N} \frac{w(x^i)}{w(x) + \sum_{j=2}^{N} w(x^j)}V(x^i)\prod_{j=2}^{N}\lambda(\mathrm{d}x^j)$$

$$\leq V(x) + (N-1)U_N$$

where we have set

$$U_N = \int \frac{w(x^2)V(x^2)\lambda(\mathrm{d}x^2)}{w(x^2) + \sum_{j=3}^{N} w(x^j)}\prod_{j=3}^{N}\lambda(\mathrm{d}x^j) .$$

Since the function $z \mapsto z/(z+a)$ is concave on $\mathbb{R}_+$ for $a > 0$, we have

$$\int \frac{w(x^2)}{w(x^2) + \sum_{j=3}^{N} w(x^j)}V(x^2)\lambda(\mathrm{d}x^2) = \lambda(V)\int \frac{w(x^2)}{w(x^2) + \sum_{j=3}^{N} w(x^j)}\frac{V(x^2)\lambda(\mathrm{d}x^2)}{\lambda(V)}$$

$$\leq \lambda(V)\frac{\int w(x^2)V(x^2)\lambda(\mathrm{d}x^2)/\lambda(V)}{\int w(x^2)V(x^2)\lambda(\mathrm{d}x^2)/\lambda(V) + \sum_{j=3}^{N} w(x^j)} \leq \frac{\pi(V)\lambda(w)}{\pi(V)\lambda(w)/\lambda(V) + \sum_{j=3}^{N} w(x^j)} .$$

The bound above implies that, with renormalization,

$$U_N \leq \int \frac{\pi(V)}{\pi(V)/\lambda(V) + \sum_{j=3}^{N} w(x^j)/\lambda(w)}\prod_{j=3}^{N}\lambda(\mathrm{d}x^j)$$

Applying now Lemma 7 with $a = \pi(V)/\lambda(V)$, $b = 1$, $M = N - 2$, and $Y_j = w(x^j)/\lambda(w)$, we obtain that

$$U_N \leq \frac{\pi(V)}{\pi(V)/\lambda(V) + (N-2)/2} + \frac{4\,\mathrm{Var}_\lambda[w]}{(N-2)\lambda(V)} \,.$$

Combining the bounds above yields (10) with

$$\mathsf{b}_{\mathsf{P}_N} = \frac{(N-1)\pi(V)}{\pi(V)/\lambda(V) + (N-2)/2} + \frac{4(N-1)\,\mathrm{Var}_\lambda[w]}{(N-2)\lambda(V)} \,. \tag{12}$$

$\square$

**Lemma 10.** *Let* $\mathsf{P}$ *be a Markov kernel on* $(\mathbb{X}, \mathcal{X})$, $\gamma$ *be a probability measure on* $(\mathbb{X}, \mathcal{X})$, *and* $\epsilon > 0$. *Let* $\mathsf{C} \in \mathcal{X}$ *be an* $(1, \epsilon\gamma)$-*small set for* $\mathsf{P}$. *Then for arbitrary Markov kernel* $\mathsf{Q}$ *on* $(\mathbb{X}, \mathcal{X})$, *the set* $\mathsf{C}$ *is an* $(1, \epsilon\gamma_{\mathsf{Q}})$-*small set for* $\mathsf{PQ}$, *where* $\gamma_{\mathsf{Q}}(\mathsf{A}) = \int \gamma(\mathrm{d}y)\mathsf{Q}(y, \mathsf{A})$ *for* $\mathsf{A} \in \mathcal{X}$.

*Proof.* Let $(x, \mathsf{A}) \in \mathsf{C} \times \mathcal{X}$. Then it holds

$$\mathsf{PQ}(x, \mathsf{A}) = \int \mathsf{P}(x, \mathrm{d}y)\mathsf{Q}(y, \mathsf{A}) \geq \epsilon \int \gamma(\mathrm{d}y)\mathsf{Q}(y, \mathsf{A}) = \epsilon\gamma_{\mathsf{Q}}(\mathsf{A}) \,.$$

$\square$

**Lemma 11.** *Let* $\mathsf{P}$ *and* $\mathsf{Q}$ *be two irreducible Markov kernels with* $\pi$ *as their unique invariant distribution. Let* $V : \mathbb{X} \to [1, \infty)$ *be a measurable function. Suppose that there exist* $\lambda_{\mathsf{Q}} \in [0, 1)$ *and* $\mathsf{b}_{\mathsf{P}}, \mathsf{b}_{\mathsf{Q}} \in \mathbb{R}_+$ *such, that* $\mathsf{P}V(x) \leq V(x) + \mathsf{b}_{\mathsf{P}}$ *and* $\mathsf{Q}V(x) \leq \lambda_{\mathsf{Q}}V(x) + \mathsf{b}_{\mathsf{Q}}$. *Let* $r_0 \geq 1$. *Also assume that for all* $r \geq r_0$, *there exist* $\epsilon_r > 0$ *and a probability measure* $\gamma_r$ *such that for all* $(x, \mathsf{A}) \in \mathsf{V}_r \times \mathcal{X}$, $\mathsf{P}(x, \mathsf{A}) \geq \epsilon_r\gamma_r(\mathsf{A})$, *where* $\mathsf{V}_r = \{x \in \mathbb{X} : V(x) \leq r\}$. *Define* $\mathsf{K} = \mathsf{PQ}$ *and* $\lambda_{\mathsf{K}} = \lambda_{\mathsf{Q}}$, $\mathsf{b}_{\mathsf{K}} = \mathsf{b}_{\mathsf{P}} + \mathsf{b}_{\mathsf{Q}}$. *Then,*

$$\mathsf{K}V(x) \leq \lambda_{\mathsf{K}}V(x) + \mathsf{b}_{\mathsf{K}} \text{ and, for all } x \in \mathsf{V}_r, \mathsf{K}(x, \mathsf{A}) \geq \epsilon_r\gamma_{\mathsf{Q},r}(\mathsf{A}),$$

*where* $\gamma_{\mathsf{Q},r}(\mathsf{A}) = \int \gamma_r(\mathrm{d}y)\mathsf{Q}(y, \mathsf{A})$. *Moreover, let* $r \geq r_0$ *be such that* $\lambda_{\mathsf{K}} + 2\mathsf{b}_{\mathsf{K}}/(1 + r) < 1$. *Then, for any* $x \in \mathbb{X}$ *and* $k \in \mathbb{N}$,

$$\|\mathsf{K}^k(x, \cdot) - \pi\|_V \leq c_{\mathsf{K}}\{V(x) + \pi(V)\}\rho_{\mathsf{K}}^k \,,$$

*where*

$$\rho_{\mathsf{K}} = \frac{\log(1 - \epsilon_r)\log\bar{\lambda}_{\mathsf{K}}}{\log(1 - \epsilon_r) + \log\bar{\lambda}_{\mathsf{K}} - \log\bar{b}_{\mathsf{K}}} \,, \quad c_{\mathsf{K}} = (\lambda_{\mathsf{K}} + \mathsf{b}_{\mathsf{K}})(1 + \bar{b}_{\mathsf{K}}/[(1 - \epsilon_r)(1 - \bar{\lambda}_{\mathsf{K}})]),$$

$$\bar{\lambda}_{\mathsf{K}} = \lambda_{\mathsf{K}} + 2\mathsf{b}_{\mathsf{K}}/(1 + r) \,, \quad \bar{b}_{\mathsf{K}} = \lambda_{\mathsf{K}}r + \mathsf{b}_{\mathsf{K}} \,.$$

*Proof.* By Lemma 10, it holds that for any $(x, \mathsf{A}) \in \mathsf{V}_r \times \mathcal{X}$, $\mathsf{K}(x, \mathsf{A}) \geq \epsilon_r\gamma_{\mathsf{Q},r}(\mathsf{A})$. Moreover, for any $x \in \mathbb{X}$, $\mathsf{K}V(x) = \mathsf{PQ}V(x) \leq \lambda_{\mathsf{Q}}\mathsf{P}V(x) + \mathsf{b}_{\mathsf{Q}} \leq \lambda_{\mathsf{Q}}V(x) + \mathsf{b}_{\mathsf{Q}} + \mathsf{b}_{\mathsf{P}}$. The proof is completed with [21, Theorem 19.4.1]. $\square$

*Proof of Theorem 2.* The proof consists of the 3 main steps:

1. Lemma 8 implies that for all $r \geq r_{\mathsf{R}}$, the level sets $\mathsf{V}_r$ for the Markov kernel $\mathsf{P}_N$ are $(1, \epsilon_{r,N}\gamma_r)$-small for the Markov kernel $\mathsf{P}_N$, where

$$\epsilon_{r,N} = (N - 1)\pi(\mathsf{V}_r)/[2w_{\infty,r} + N - 2],$$

and $\gamma_r(\mathsf{A}) = \int \pi_{\mathsf{V}_r}(\mathrm{d}y)\mathsf{R}(y, \mathsf{A})$ with $\pi_{\mathsf{V}_r}(B) = \pi(B \cap \mathsf{V}_r)/\pi(\mathsf{V}_r)$, for any $B \in \mathcal{X}$.
2. Lemma 9 implies that for all $x \in \mathbb{X}$, $\mathsf{P}_N V(x) \leq V(x) + \mathsf{b}_{\mathsf{P}_N}$, where $\mathsf{b}_{\mathsf{P}_N}$ is given in (12).
3. We finally show (see Lemma 11) that the Markov kernel $\mathsf{K}_N$ also satisfies a Foster-Lyapunov condition with the same drift function $V$ as $\mathsf{R}$, that is, $\mathsf{K}_N V \leq \lambda_{\mathsf{R}} V + \mathsf{b}_{\mathsf{K}_N}$ with $\mathsf{b}_{\mathsf{K}_N} = \mathsf{b}_{\mathsf{R}} + \mathsf{b}_{\mathsf{P}_N}$.

We conclude by using Lemma 11. We choose $r_N = r_{\mathsf{R}} \vee \{4\mathsf{b}_{\mathsf{K}_N}/(1 - \lambda_{\mathsf{R}}) - 1\}$. Then $\lambda_{\mathsf{R}} + 2\mathsf{b}_{\mathsf{K}_N}/(1 + r_N) \leq (1 + \lambda_{\mathsf{R}})/2 < 1$, and Lemma 11 implies (3) with

$$\log\tilde{\kappa}_{\mathsf{K}_N} = \frac{\log(1 - \epsilon_{r,N})\log\bar{\lambda}_{\mathsf{K}_N}}{\log(1 - \epsilon_{r,N}) + \log\bar{\lambda}_{\mathsf{K}_N} - \log\bar{b}_{\mathsf{K}_N}} \,,$$

$$c_{\mathsf{K}_N} = (\lambda_{\mathsf{R}} + \bar{b}_{\mathsf{K}_N})(1 + \bar{b}_{\mathsf{K}_N}/[2(1 - \epsilon_{r_N,N})(1 - \bar{\lambda}_{\mathsf{K}_N})]) \,,$$

$$\bar{\lambda}_{\mathsf{K}_N} = (1 + \lambda_{\mathsf{R}})/2 \,, \quad \bar{b}_{\mathsf{K}_N} = \lambda_{\mathsf{R}}r_N + \mathsf{b}_{\mathsf{K}_N} \,.$$

Set $b_{\mathsf{K}_\infty} = \lim_{N\to\infty} b_{\mathsf{K}_N} = b_{\mathsf{R}} + b_{\mathsf{P}_\infty}$, where $b_{\mathsf{P}_\infty}$ is defined in (11), $r_\infty = r_{\mathsf{R}} \vee [4b_{\mathsf{K}_\infty}/(1-\lambda_{\mathsf{R}}) - 1]$ and $\epsilon_\infty = \pi(\mathsf{V}_{r_\infty})$. With these notations, we have

$$
\begin{aligned}
\log \tilde{\kappa}_{\mathsf{K}_\infty} &= \frac{\log(1-\epsilon_\infty)\log\bar{\lambda}_{\mathsf{K}_\infty}}{\log(1-\epsilon_\infty) + \log\bar{\lambda}_{\mathsf{K}_\infty} - \log\bar{b}_{\mathsf{K}_\infty}}\ , \\
c_{\mathsf{K}_\infty} &= (\lambda_{\mathsf{R}} + \bar{b}_{\mathsf{K}_\infty})(1 + \bar{b}_{\mathsf{K}_\infty}/[(1-\epsilon_\infty)(1-\bar{\lambda}_{\mathsf{K}_\infty})]) \\
\bar{\lambda}_{\mathsf{K}_\infty} &= (1+\lambda_{\mathsf{R}})/2\ ,\ \bar{b}_{\mathsf{K}_\infty} = \lambda_{\mathsf{R}} r_\infty + b_{\mathsf{K}_\infty}\ .
\end{aligned}
\tag{13}
$$

$\square$

## C  Metropolis-Adjusted Langevin rejunevation kernel

This section addresses the convergence of the Metropolis Adjusted Langevin algorithm (MALA) for sampling from a positive target probability density $\pi$ on $(\mathbb{R}^d, \mathcal{B}(\mathbb{R}^d))$, where $\mathcal{B}(\mathbb{R}^d)$ is the Borel $\sigma$ field of $\mathbb{R}^d$ endowed with the Euclidean topology. For simplicity, let $U = -\log\pi$ be the associated potential function. MALA is a Markov chain Monte Carlo (MCMC) method based on Langevin diffusion associated with $\pi$:

$$
\mathrm{d}\mathbf{X}_t = -\nabla U(\mathbf{X}_t)\mathrm{d}t + \sqrt{2}\mathrm{d}\mathbf{B}_t\ ,
\tag{14}
$$

where $(\mathbf{B}_t)_{t\geq 0}$ is a $d$-dimensional Brownian motion. It is known that under mild conditions this diffusion admits a strong solution $(\mathbf{X}_t^{(x)})_{t\geq 0}$ for any starting point $x \in \mathbb{R}^d$ and defines a Markov semigroup $(\mathbf{P}_t)_{t\geq 0}$ for any $t \geq 0$, $x \in \mathbb{R}^d$ and $\mathsf{A} \in \mathcal{B}(\mathbb{R}^d)$ by $\mathbf{P}_t(x, \mathsf{A}) = PP(\mathbf{X}_t^{(x)} \in \mathsf{A})$. Moreover, this Markov semigroup admits $\pi$ as its unique stationary measure, is ergodic and even $V$-uniformly geometrically ergodic with additional assumptions on $U$ (see [65, 47]). However, sampling a path solution of (14) is a real challenge in most cases, and discretizations are used instead to obtain a Markov chain with similar long-term behaviour. Here we consider the Euler-Maruyama discretization, which is given by (14), defined for all $k \geq 0$ by

$$
Y_{k+1} = Y_k - \gamma\nabla U(Y_k) + \sqrt{2\gamma}Z_{k+1}\ ,
\tag{15}
$$

where $\gamma$ is the step size of the discretization and. $\{Z_k,\ k \in \mathbb{N}^*\}$ is a i.i.d. sequence of $d$-dimensional standard Gaussian random variables. This algorithm was proposed by [24, 56] and later studied by [28, 29, 51, 65]. According to [65], this algorithm is called the Unadjusted Langevin algorithm (ULA). A drawback of this method is that even if the Markov chain $\{Y_k,\ k \in \mathbb{N}\}$ has a unique stationary distribution $\pi_\gamma$ and is ergodic (which is guaranteed under mild assumptions about $U$), $\pi_\gamma$ is different from $\pi$ most of the time. To solve this problem, in [67, 65] it is proposed to use the Markov kernel associated with the recursion defined by the Euler-Maruyama discretization (15) as a proposal kernel in a Metropolis-Hastings algorithm that defines a new Markov chain $\{X_k,\ k \in \mathbb{N}\}$ by:

$$
X_{k+1} = X_k + 1_{\mathbb{R}_+}(U_{k+1} - \alpha_\gamma(X_k, \tilde{Y}_{k+1}))\{\tilde{Y}_{k+1} - X_k\}\ ,
\tag{16}
$$

where $\tilde{Y}_{k+1} = X_k - \gamma\nabla U(X_k) + \sqrt{2\gamma}Z_{k+1}$, $\{U_k,\ k \in \mathbb{N}^*\}$ is a sequence of i.i.d. uniform random variables on $[0,1]$ and $\alpha_\gamma : \mathbb{R}^{2d} \to [0,1]$ is the usual Metropolis acceptance ratio. This algorithm is called Metropolis Adjusted Langevin Algorithm (MALA) and has since been used in many applications.

Denote by $r_\gamma$ the proposal transition density associated to the Euler-Maruyama discretization (15) with stepsize $\gamma > 0$, *i.e.*, for any $x, y \in \mathbb{R}^d$,

$$
r_\gamma(x, y) = (4\pi\gamma)^{-d/2}\exp\left(-(4\gamma)^{-1}\|y - x + \gamma\nabla U(x)\|^2\right)\ .
$$

Then, the Markov kernel $R_\gamma$ of the MALA algorithm (16) is given for $\gamma > 0$, $x \in \mathbb{R}^d$, and $\mathsf{A} \in \mathcal{B}(\mathbb{R}^d)$ by

$$
R_\gamma(x, \mathsf{A}) = \int_{\mathbb{R}^d} 1_{\mathsf{A}}(y)\alpha_\gamma(x, y)r_\gamma(x, y)\mathrm{d}y + \delta_x(\mathsf{A})\int_{\mathbb{R}^d}\{1 - \alpha_\gamma(x, y)\}r_\gamma(x, y)\mathrm{d}y\ ,
\tag{17}
$$

$$
\alpha_\gamma(x, y) = 1 \wedge \frac{\pi(y)r_\gamma(y, x)}{\pi(x)r_\gamma(x, y)}\ .
$$

It is well-known, see e.g. [65], that for any $\gamma > 0$, $R_\gamma$ is reversible with respect to $\pi$ and $\pi$-irreducible.

**H1.** *The function $U : \mathbb{R}^d \to \mathbb{R}$ is three times continuously differentiable. In addition, $\nabla U(0) = 0$ and there exists $\mathtt{L} \geq 0$ and $\mathtt{M} \geq 0$ such that $\sup_{x \in \mathbb{R}^d} \|\mathrm{D}^2 U(x)\| \leq \mathtt{L}$ such that $\sup_{x \in \mathbb{R}^d} \|\mathrm{D}^3 U(x)\| \leq \mathtt{M}$.*

The condition $\nabla U(0) = 0$ is satisfied (up to a translation) as soon as $U$ has a local minimum, which is the case when $\lim_{\|x\| \to +\infty} U(x) = +\infty$, since $U$ is continuous.

**H2.** *There exist $\mathtt{m} > 0$ and $\mathtt{K} \geq 0$ such that for any $x, y \in \mathbb{R}^d$, $\|x\| \geq \mathtt{K}$ and $\|y\| = 1$,*

$$\mathrm{D}^2 U(x)\{y\}^{\otimes 2} \geq \mathtt{m} .$$

Note that under **H**1 and **H**2, for any $x, y \in \mathbb{R}^d$, $\|y\| = 1$, it holds that

$$\mathrm{D}^2 U(x)\{y\}^{\otimes 2} \geq \mathtt{m} - (\mathtt{m} + \mathtt{L})1_{\mathrm{B}(0,\mathtt{K})}(x) .$$

In the case $\mathtt{K} = 0$, **H**2 amounts to $U$ being strongly convex and the convexity constant being equal to $\mathtt{m}$. However, if $\mathtt{K} > 0$, **H**2 is a slight strengthening of the condition of strong convexity at infinity considered in [18, 23]: there is $\mathtt{m}' > 0$ and $\mathtt{K}' \geq 0$ such that for each $x, y \in \mathbb{R}^d$, $\|x - y\| \geq \mathtt{K}'$

$$\langle \nabla U(x) - \nabla U(y), x - y \rangle \geq \mathtt{m}'\|x - y\|^2 . \tag{18}$$

Indeed, if (18) holds for any $x, y \in \mathbb{R}^d$ that $\|x\| \vee \|y\| \geq \mathtt{K}'$ instead of $\|x - y\| \geq \mathtt{K}'$, then a simple calculation implies that **H**2 holds with $\mathtt{m} \leftarrow \mathtt{m}'$ and $\mathtt{K} \leftarrow \mathtt{K}' + 1$. Finally, while the condition (18) holds for $x, y \in \mathbb{R}^d$, $\|x - y\| \geq \mathtt{K}'$, is weaker than **H**2, it may be more convenient in many situations to check whether the latter holds.

**Lemma 12.** *Assume **H**1 and **H**2 hold. The function $U$ satisfies for any $x \in \mathbb{R}^d$,*

$$\langle \nabla U(x), x \rangle \geq (\mathtt{m}/2)\|x\|^2 - \tilde{\mathtt{C}}1_{\mathrm{B}(0,\tilde{\mathtt{K}})}(x) ,$$

*with $\tilde{\mathtt{K}} = 2\mathtt{K}(1 + \mathtt{L}/\mathtt{m})$ and $\tilde{\mathtt{C}} = \mathtt{L}\tilde{\mathtt{K}}^2$.*

Note that under **H**1 and **H**2, $\mathtt{m} \leq \mathtt{L}$. Define for any $\eta > 0$, $V_\eta : \mathbb{R}^d \to [1, +\infty)$ for any $x \in \mathbb{R}^d$ by

$$V_\eta(x) = \exp(\eta\|x\|^2) . \tag{19}$$

The analysis of MALA is naturally related to the study of the ULA algorithm. More precisely, since for any $x \in \mathbb{R}^d$ and $\mathsf{A} \in \mathcal{B}(\mathbb{R}^d)$, the Markov kernel corresponding to ULA (15) is given by

$$Q_\gamma(x, \mathsf{A}) = \int_{\mathbb{R}^d} 1_\mathsf{A}(x - \gamma\nabla U(x) + \sqrt{2\gamma}z)\, \mathrm{g}(z)\mathrm{d}z.$$

To show that MALA satisfies a Lyapunov condition, we first state a drift condition for the ULA algorithm.

**Proposition 13.** *Assume **H**1 and **H**2 and let $\bar{\gamma} \in \left(0, \mathtt{m}/(4\mathtt{L}^2)\right]$. Then, for any $\gamma \in (0, \bar{\gamma}]$, $x \in \mathbb{R}^d$,*

$$Q_\gamma V_{\bar{\eta}}(x) \leq \exp(-\bar{\eta}\mathtt{m}\gamma\|x\|^2/4)V_{\bar{\eta}}(x) + b_{\bar{\gamma}}^{\mathrm{U}}\gamma 1_{\mathrm{B}(0,K^{\mathrm{U}})}(x) ,$$

*where $V_{\bar{\eta}}$ is defined in (19), $\bar{\eta} = \mathtt{m}/16$, $K^{\mathrm{U}} = \max(\tilde{\mathtt{K}}, 4\sqrt{d/\mathtt{m}})$, $\tilde{\mathtt{K}}$ is defined in Lemma 12 and*

$$b_{\bar{\gamma}}^{\mathrm{U}} = \left[\bar{\eta}\{\mathtt{m}/4 + (1 + 16\bar{\eta}\bar{\gamma})(4\bar{\eta} + 2\mathtt{L} + \bar{\gamma}\mathtt{L}^2)\}(K^{\mathrm{U}})^2 + 4\bar{\eta}d\right]$$
$$\times \exp(\bar{\gamma}\bar{\eta}\{\mathtt{m}/4 + (1 + 16\bar{\eta}\bar{\gamma})(4\bar{\eta} + 2\mathtt{L} + \bar{\gamma}\mathtt{L}^2)\}(K^{\mathrm{U}})^2 + 4\bar{\eta}\bar{\gamma}d) .$$

*Proof.* The proof follows from [22, Proposition 6]. $\qquad\square$

We now introduce for $\bar{\gamma} > 0$ the auxiliary constant

$$C_{1,\bar{\gamma}} = 2(2^{1/2}\mathtt{M} \vee \bar{\gamma}^{1/2}\mathtt{M}\mathtt{L} \vee 2\mathtt{L}^2[1 \vee \bar{\gamma}^{1/2} \vee \bar{\gamma}\mathtt{L} \vee (\bar{\gamma}\mathtt{L}^{4/3})^{3/2}]) . \tag{20}$$

For $\bar{\gamma} \in \left(0, \mathtt{m}^3/(4\mathtt{L}^4)\right]$, we also define $C_{2,\bar{\gamma}}$ as

$$C_{2,\bar{\gamma}} = 2\mathtt{L} + (\bar{\gamma}/2)\mathtt{L}^2 + 2^{-3/2}\bar{\gamma}^{3/2}\mathtt{L}^3 + \{2^{1/2}\mathtt{L}^2 + (2^{1/2}\mathtt{L}^2 + 2^{-3/2}\bar{\gamma}^{1/2})\mathtt{L}^3\}^2(2^4/\mathtt{m}^3) .$$

Using Proposition 13, we state a drift condition for the MALA kernel $R_\gamma$.

**Proposition 14.** *Assume **H** 1 and **H** 2. Then, there exist $\Gamma > 0$ (given in (21)) such that for any $\bar{\gamma} \in (0, \Gamma]$, $\gamma \in (0, \bar{\gamma}]$ and $x \in \mathbb{R}^d$,*

$$R_\gamma V_{\bar{\eta}}(x) \leq (1 - \varpi\gamma)V_{\bar{\eta}}(x) + b_{\bar{\gamma}}^{\mathrm{M}}\gamma 1_{\mathrm{B}(0, K^{\mathrm{M}})}(x) ,$$

*where $V_{\bar{\eta}}$ is defined by (19), $R_\gamma$ is the Markov kernel of MALA defined by (17), $\bar{\eta} = \mathtt{m}/16$, $\varpi = \bar{\eta}\mathtt{m}(K^{\mathrm{M}})^2/16$, and*

$$
\begin{aligned}
\Gamma_{1/2} &= \min\left(1, \mathtt{m}^3/(4\mathtt{L}^4), d^{-1}\right) , \quad \Gamma = \min\left(\Gamma_{1/2}, 4/\{\mathtt{m}\bar{\eta}(K^{\mathrm{M}})^2\}\right) , \quad (21)\\
K^{\mathrm{M}} &= \max(2^4, 2\mathtt{K}, K^{\mathrm{U}}, \tilde{\mathtt{K}}, 4b_{1/2}^{1/2}/(\mathtt{m}\bar{\eta})^{1/2}) , \quad b_{1/2} = C_{2,\Gamma_{1/2}}d + \sup_{u \geq 1}\{ue^{-u/2^7}\} ,\\
b_{\bar{\gamma}}^{\mathrm{M}} &= b_{\bar{\gamma}}^{\mathrm{U}} + \bar{\eta}\mathtt{m}(K^{\mathrm{M}})^2 e^{\bar{\eta}(K^{\mathrm{M}})^2}/16 + C_{1,\bar{\gamma}}\bar{\gamma}^{1/2}\left\{d + \sqrt{3}d^2 + (K^{\mathrm{M}})^2\right\} ,
\end{aligned}
$$

*where $K^{\mathrm{U}}, b_{\bar{\gamma}}^{\mathrm{U}}$ are defined in Proposition 13, and $\tilde{\mathtt{K}}$ is defined in Lemma 12.*

*Proof.* The proof follows from [22, Proposition 7]. $\qquad\square$

Quantitative bound on the mixing rate of the MALA sampler requires also the *minorization condition* for the MALA kernel. The result below is due to [22, Proposition 12].

**Proposition 15.** *Assume **H** 1 and **H** 2. Then for any $K \geq 0$ there exists $\tilde{\Gamma}_K > 0$ (given in (22) below), such that for any $x, y \in \mathbb{R}^d$, $\|x\| \vee \|y\| \leq K$, and $\gamma \in (0, \tilde{\Gamma}_K]$ we have*

$$\|\delta_x R_\gamma^{\lceil 1/\gamma\rceil} - \delta_y R_\gamma^{\lceil 1/\gamma\rceil}\|_{\mathrm{TV}} \leq 2(1 - \varepsilon(K)/2) ,$$

*where*

$$
\varepsilon(K) = 2\mathbf{\Phi}\left(-\sqrt{3}(\mathtt{L}+1)^{1/2}K\right) , \quad \tilde{\Gamma}_{1/2} = \mathtt{m}/(4\mathtt{L}^2) , \quad (22)
$$

$$
\tilde{\Gamma}_K = \tilde{\Gamma}_{1/2} \wedge \left[\frac{\varepsilon(K)}{2C_{1,\tilde{\Gamma}_{1/2}}(d + \sqrt{3}d^2 + K^2 + 2\tilde{b}_{\tilde{\Gamma}_{1/2}}^{\mathrm{U}}/\mathtt{m})}\right]^2 ,
$$

$$
\tilde{b}_{\tilde{\Gamma}_{1/2}}^{\mathrm{U}} = 2d + [\max(\tilde{\mathtt{K}}, 2\sqrt{(2d)/\mathtt{m}})]^2 \left(\tilde{\Gamma}_{1/2}\mathtt{L}^2 + 2\mathtt{L} + \mathtt{m}/2\right) ,
$$

*where $C_{1,\tilde{\Gamma}_{1/2}}$ is defined in (20), $\tilde{\mathtt{K}}$ is defined in Lemma 12, and $\mathbf{\Phi}(\cdot)$ is the cumulative distribution function of the Gaussian distribution with zero mean an unit variance on $\mathbb{R}$.*

It is interesting to note that $\gamma$ is the discretization step of the underlying Langevin diffusion. We have to iterate the kernel $1/\gamma$ times for the diffusion to progress by one time unit. Combining Proposition 14 and Proposition 15 yields the following ergodicity result in $V_{\bar{\eta}}$-norm.

**Theorem 16.** *Assume **H** 1 and **H** 2. Then, there exist $\bar{\Gamma} > 0$ (defined in (23) below), such that for any $\gamma \in (0, \bar{\Gamma}]$, there exist $C_{\bar{\Gamma}} \geq 0$ and $\rho_{\bar{\Gamma}} \in [0, 1)$ (given in (23)) satisfying for any $x \in \mathbb{R}^d$,*

$$\|\delta_x R_\gamma^k - \pi\|_{V_{\bar{\eta}}} \leq C_{\bar{\Gamma}}\rho_{\bar{\Gamma}}^{\gamma k}\{V_{\bar{\eta}}(x) + \pi(V_{\bar{\eta}})\} ,$$

*where $\bar{\eta} = \mathtt{m}/16$,*

$$
\begin{aligned}
\log\rho_{\bar{\Gamma}} &= \frac{\log(1 - 2^{-1}\varepsilon(K_{\bar{\Gamma}}))\log\bar{\lambda}}{\log(1 - 2^{-1}\varepsilon(K_{\bar{\Gamma}})) + \log\bar{\lambda} - \log\bar{b}_{\bar{\Gamma}}^{\mathrm{M}}} ,\\
\bar{\lambda} &= (1 + \lambda)/2 , \quad \lambda = e^{-\varpi} , \quad \bar{b}_{\bar{\Gamma}}^{\mathrm{M}} = \lambda b_{\bar{\Gamma}}^{\mathrm{M}} + M_{\bar{\Gamma}} , \quad \bar{\Gamma} = \Gamma \wedge \tilde{\Gamma}_{K_\Gamma} ,\\
M_{\bar{\gamma}} &= \left(\frac{4b_{\bar{\gamma}}^{\mathrm{M}}(1 + \bar{\gamma})}{1 - \lambda}\right) \vee 1 , \quad K_{\bar{\gamma}} = (\log(M_{\bar{\gamma}})/\bar{\eta})^{1/2} , \quad \bar{\gamma} \in \{\bar{\Gamma}, \Gamma\} ,\\
C_{\bar{\Gamma}} &= \rho_{\bar{\Gamma}}^{-1}\{\lambda + 1\}\{1 + \bar{b}_{\bar{\Gamma}}^{\mathrm{M}}/[1 - 2^{-1}\varepsilon(K_{\bar{\Gamma}})(1 - \bar{\lambda})]\} ,
\end{aligned}
\qquad (23)
$$

*and $\varpi$ is given in Proposition 14.*

*Proof.* The proof follows from [22, Theorem 2]. For completeness we repeat here the main steps of the proof. Proposition 14 shows that there exist $\Gamma > 0$ (given in (21)) such that for any $\bar{\gamma} \in (0, \Gamma]$, $\gamma \in (0, \bar{\gamma}]$ and $x \in \mathbb{R}^d$,

$$R_\gamma V_{\bar{\eta}}(x) \leq (1 - \varpi\gamma)V_{\bar{\eta}}(x) + b_{\bar{\gamma}}^{\mathrm{M}}\gamma ,$$

where the constants $\varpi$ and $b_{\bar{\gamma}}^{\mathrm{M}}$ are given in Proposition 14. Hence, setting $\lambda = \mathrm{e}^{-\varpi} < 1$, we obtain by induction that

$$R_\gamma^{\lceil 1/\gamma \rceil} V_{\bar{\eta}}(x) \leq \lambda V_{\bar{\eta}}(x) + b_{\bar{\gamma}}^{\mathrm{M}} .$$

Now we set $M_{\bar{\gamma}}$ and $K_{\bar{\gamma}}$ as in (23). Then Proposition 15 implies that for any $\bar{\gamma} \in \left(0, \tilde{\Gamma}_{K_\Gamma}\right]$, any $x, y \in \{V_{\bar{\eta}}(\cdot) \leq M_{\bar{\gamma}}\}$, and $\gamma \in (0, \bar{\gamma}]$,

$$\|\delta_x R_\gamma^{\lceil 1/\gamma \rceil} - \delta_y R_\gamma^{\lceil 1/\gamma \rceil}\|_{\mathrm{TV}} \leq 2(1 - \varepsilon(K_{\bar{\gamma}})) .$$

Now it remains to combine both statements with $\bar{\gamma} = \Gamma \wedge \tilde{\Gamma}_{K_\Gamma}$ and apply [21, Theorem 19.4.1] to the Markov kernel $R_\gamma^{\lceil 1/\gamma \rceil}$. $\qquad\square$

**Comparison with** $\mathrm{Ex}^2\mathrm{MCMC}$ **kernel.** Based on the results above, we first state the quantitative mixing rate bounds for $\mathrm{Ex}^2\mathrm{MCMC}$ algorithm with the MALA kernel $R_\gamma^{\lceil 1/\gamma \rceil}$ (iterated $\lceil 1/\gamma \rceil$ times) applied as rejuvenation kernel. The corresponding Markov kernel writes for $x \in \mathbb{R}^d$ and $\mathsf{A} \in \mathcal{B}(\mathbb{R}^d)$ as

$$\mathsf{K}_{N,\gamma}(x, \mathsf{A}) = \mathsf{P}_N R_\gamma^{\lceil 1/\gamma \rceil}(x, \mathsf{A}) = \int \mathsf{P}_N(x, \mathrm{d}y) R_\gamma^{\lceil 1/\gamma \rceil}(y, \mathsf{A}) ,$$

where $R_\gamma(x, \mathsf{A})$ is defined in (17). Note also that, for $r \geq 1$, and $V_{\bar{\eta}}$ defined in (19), the level sets

$$\mathsf{V}_{\bar{\eta},r} = \{x : V_{\bar{\eta}}(x) \leq r\} = \{x : \|x\| \leq \sqrt{\log r/\bar{\eta}}\} .$$

The result above allows to state the following ergodicity result for $\mathsf{K}_{N,\gamma}$ kernel.

**Theorem 17.** *Assume **H 1**, **H 2**, and **A1**,**A2** with $V_{\bar{\eta}}$ defined in (19). Then there exist $\bar{\Gamma}$ (defined in (23)), such that for any $\gamma \in \left(0, \bar{\Gamma}\right]$, $x \in \mathbb{R}^d$, and $k \in \mathbb{N}$,*

$$\|\mathsf{K}_{N,\gamma}^k(x, \cdot) - \pi\|_V \leq c_N \{\pi(V_{\bar{\eta}}) + V_{\bar{\eta}}(x)\}\tilde{\kappa}_N^k ,$$

*where $V_{\bar{\eta}}$ is defined in (19), and the constants $c_N$, $\tilde{\kappa}_N \in [0, 1)$ are given by*

$$\log \tilde{\kappa}_N = \frac{\log(1 - \epsilon_{r_N,N}) \log \bar{\lambda}}{\log(1 - \epsilon_{r_N,N}) + \log \bar{\lambda} - \log \bar{b}_N} , \quad r_N = 1 \vee \{4b_N/(1 - \lambda) - 1\} , \qquad (24)$$

$$\epsilon_{r_N,N} = (N - 1)\pi(\mathsf{V}_{\bar{\eta},r_N})/[2w_{\infty,r_N} + N - 2], \quad b_N = \mathsf{b}_{\mathsf{P}_N} + \bar{b}_{\bar{\Gamma}}^{\mathrm{M}} ,$$

$$c_N = (\lambda + \bar{b}_N)(1 + \bar{b}_N/[2(1 - \epsilon_{r_N,N})(1 - \bar{\lambda})])$$

$$\bar{\lambda} = (1 + \lambda)/2 , \quad \bar{b}_N = \lambda r_N + b_N ,$$

*and $\lambda$ is defined in (23).*

*Proof.* The proof follows from the combination of Theorem 2 and Proposition 14. $\qquad\square$

To derive the geometric ergodicity rates in Theorem 17, it is not required to identify the small sets of the MALA rejuvenation kernel $R_\gamma$. The only quantity of interest is the Foster-Lyapunov drift condition satisfied by $R_\gamma^{\lceil 1/\gamma \rceil}$. Theorem 16 implies that the rate of convergence of MALA is $\gamma \log \rho_{\bar{\Gamma}}$. The following statement allows to quantify the improvement in the convergence rate of $\mathsf{K}_{N,\gamma}$ compared to $R_\gamma^{\lceil 1/\gamma \rceil}$. Following [58], we consider the relative improvement of the *mixing time* of the considered Markov kernels. To introduce formally the mixing time, we need an auxiliary definition of the $V$-Dobrushin coefficient. We refer the reader to [21, Section 18.3] for more detailed exposition. Recall that $M_{1,V}(\mathbb{X})$ is a set of probability measures on $(\mathbb{X}, \mathcal{X})$, such that $\xi(V) < \infty$.

**Definition 18** (*V*-Dobrushin coefficient)*.* *Let* $V : \mathbb{X} \mapsto [1; +\infty)$ *be a measurable function, and* $\mathsf{Q}$ *be a Markov kernel on* $(\mathbb{X}, \mathcal{X})$*, such that* $\xi(V) < \infty$ *implies* $\xi\mathsf{Q}(V) < \infty$ *for any measure* $\xi \in M_{1,V}(\mathbb{X})$*. Then the* $V$*-Dobrushin coefficient of the Markov kernel* $\mathsf{Q}$*, is defined by*

$$\Delta_V(\mathsf{Q}) = \sup_{\xi \neq \xi' \in M_{1,V}(\mathbb{X})} \frac{\|\xi\mathsf{Q} - \xi'\mathsf{Q}\|_V}{\|\xi - \xi'\|_V} .$$

It is known (see e.g. [21, Theorem 18.4.1]), that $V$-geometric ergodicity of the Markov kernel $\mathsf{Q}$ (see Definition 1) is equivalent to the fact, that

$$\Delta_V(\mathsf{Q}^m) \leq 1 - \varepsilon$$

for some $m \in \mathbb{N}^*$ and $0 < \varepsilon < 1$.

**Definition 19.** *Let* $\mathsf{Q}$ *be* $V$*-geometrically ergodic Markov kernel. Then the corresponding mixing time* $t_{\mathrm{mix}} \in \mathbb{N}^*$ *is defined as*

$$t_{\mathrm{mix}} = \inf_{m \in \mathbb{N}^*} \{m : \Delta_V(\mathsf{Q}^m) \leq 1/4\} .$$

Note that if $\mathsf{Q}$ is $V$-geometrically ergodic with factor $0 < \rho < 1$ given in $Definition\ 1$, its mixing time $t_{\mathrm{mix}}$ is bounded as $t_{\mathrm{mix}} \leq (\log(1/\rho))^{-1} \log(4M)$.

Now we compare the mixing time of $\mathsf{K}_{N,\gamma}$, which is inversely proportional to $\log(1/\tilde{\kappa}_N)$, to the mixing time of $R_\gamma^{[1/\gamma]}$, which is inversely proportional to $\log(1/\rho_{\bar{\Gamma}})$.

**Theorem 20.** *Assume* **H***1-***H***2 and* **A***1-***A***2 with* $V_{\bar{\eta}}$*. Then there exist* $\bar{\Gamma}$ *(defined in* (23)*), such that for any* $\gamma \in \left(0, \bar{\Gamma}\right]$*, it holds that*

$$\lim_{N \to \infty} \frac{\log(\rho_{\bar{\Gamma}})}{\log(\tilde{\kappa}_N)} = \frac{\log(1 - 2^{-1}\varepsilon(K_{\bar{\Gamma}}))}{\log(1 - \epsilon_\infty)} \times \frac{\log(1 - 2^{-1}\varepsilon(K_{\bar{\Gamma}})) + \log\bar{\lambda} - \log\bar{b}_{\bar{\Gamma}}^{\mathrm{M}}}{\log(1 - \epsilon_\infty) + \log\bar{\lambda} - \log\bar{b}_\infty} , \quad (25)$$

*where* $\lambda, \bar{\lambda}$*, and* $\bar{b}_{\bar{\Gamma}}^{\mathrm{M}}$ *are defined in* (23)*,* $\varepsilon(\cdot)$ *is defined in* (22)*, and*

$$r_\infty = 1 \vee \{4b_\infty/(1 - \lambda) - 1\} , \quad \epsilon_\infty = \pi(V_{\bar{\eta}, r_\infty}) , \quad b_\infty = \mathsf{b}_{\mathsf{K}_\infty} + \bar{b}_{\bar{\Gamma}}^{\mathrm{M}} , \quad \bar{b}_\infty = \lambda r_\infty + b_\infty .$$

*Proof.* The proof follows by combining the expressions (23) and (24). $\qquad\square$

The ratio $\log(1 - 2^{-1}\varepsilon(K_{\bar{\Gamma}}))/\log(1 - \epsilon_\infty)$ is extremely small in most settings. This explains the observed behavior: the mixing time of the Ex2MCMC kernel is much smaller than the mixing time of the MALA algorithm, which we observe in practice in all the examples we discuss. The difference is even more spectacular when the dimension increases. To illustrate this phenomenon, we consider the following numerical scenario for (25). We assume that **H**1-**H**2 holds with $\mathtt{m} = 0.1, \mathtt{M} = 2.0, \mathtt{L} = 1.0$, and $\mathtt{K} = 5.0$. One can evaluate that even for $d = 2$ the respective value $K_{\bar{\Gamma}} \approx 10^3$. We now show, how the bound for $K_{\bar{\Gamma}}$ scales with the dimension $d$. The respective plot for $d \in [2; 100]$ is given in Figure 6. It implies that $K_{\bar{\Gamma}}$ grows as $\sqrt{d}$. At the same time, the standard bound $\Phi(-x) \leq \exp\{-x^2/2\}$, valid for $x \geq 0$, yields that $\varepsilon(K_{\bar{\Gamma}})/2 \leq \exp\{-(3/2)(L + 1)K_{\bar{\Gamma}}^2\}$. At the same time, $\epsilon_\infty$ typically does not decrease with the growth of $d$ due to the construction of $r_\infty$. Hence, the ratio (25) decreases exponentially with the growth of $d$ in our model scenario.

## D Proof of Theorem 3

The proof relies on results of stochastic approximation with Markovian dynamics, see e.g. [7, 8]. For reader's convenience, before going into the details, we give an outline of the proof. The motivation of such algorithms is to find the roots of the function $h : \Theta \to \mathbb{R}^q, \Theta \subset \mathbb{R}^q$

$$h(\theta) = \int_{\mathbb{U} \times \mathbb{E}} H(\theta, u, e)\mu(\mathrm{d}e)\rho_\theta(\mathrm{d}e) ,$$

for families of functions $\{H(\theta, u, e) : \Theta \times \mathbb{U} \times \mathbb{E} \to \Theta\}$, a family of probability distributions $\{\rho_\theta, \theta \in \Theta\}$ of $(\mathbb{E}, \mathcal{E})$ and a probability distribution $\mu$ on a space $(\mathbb{U}, \mathcal{U})$. These roots are not available analytically and a way of finding them numerically consists of considering the controlled Markov

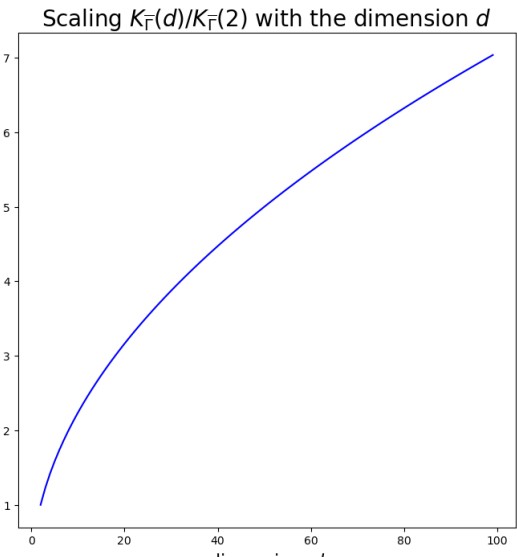

Figure 6: Scaling of $K_{\bar{\Gamma}}$ with dimension $d$, normalized by its value corresponding to $d = 2$.

chain on $\left\{ (\Theta \times \mathbb{U})^{\mathbb{N}}, (\mathcal{B}(\Theta) \otimes \mathcal{U})^{\otimes \mathbb{N}} \right\}$ initialized at some $(\theta_0, U_0) = (\vartheta, u) \in \Theta \times \mathbb{U}$ and defined recursively for a sequence of stepsize $\{\gamma_i, \ i \in \mathbb{N}\}$ by

$$U_{i+1} \sim P_{\theta_i}(U_i, \cdot), \quad E_{i+1} \sim \rho_{\theta_i}$$
$$\theta_{i+1} = \theta_i + \gamma_{i+1} H(\theta_i, U_{i+1}, E_{i+1}) .$$

Here $\{P_\theta, \theta \in \Theta\}$ is a family of Markov kernels such that for each $\theta \in \Theta$, $\mu P_\theta = \mu$. The rationale for this recursion goes as follows. Let us first rewrite the Robbins-Monro recursion

$$\theta_{i+1} = \theta_i + \gamma_{i+1}\{h(\theta_i) + \xi_{i+1}\},$$

where $\xi_{i+1} = H(\theta_i, U_{i+1}, E_{i+1})$ is referred to as the "noise". Therefore, $\{\theta_i\}$ is a noisy version of the sequence $\{\bar{\theta}_i\}$ defined as $\bar{\theta}_{i+1} = \bar{\theta}_i + \gamma_{i+1} h(\bar{\theta}_i)$. The convergence of such sequences has been studied by many authors, starting with [48] under various conditions. A crucial step of such convergence analysis consists of assuming that the sequence $\{\theta_i\}$ remains bounded with probability 1 in a compact set of $\Theta$. This problem has traditionally can be circumvented by means of modifications of the recursion. Indeed, one of the major difficulties specific to the Markovian dynamic scenario is that $\{\theta_i\}$ governs the ergodicity of the controlled Markov chain $\{U_i\}$ and that stability properties of $\{\theta_i\}$ require "good" ergodicity properties which might vanish whenever $\{\theta_i\}$ approaches $\partial \Theta$ often away from the roots of $h(\theta)$, resulting in instability. Most existing results rely on modifications of the updates designed to ensure a form of ergodicity of $\{\xi_i\}$ which in turn ensures that $\{\theta_i\}$ inherits the stability properties of $\{\bar{\theta}_i\}$; see e.g. [7, 10] and the discussion in [8, Section 3]. We follow here [10]. Let $\{\mathcal{R}_i\}$ be a sequence of compact subsets of $\Theta$ and consider the recursion:

$$U_{i+1} \sim P_{\theta_i}(U_i, \cdot) \quad E_{i+1} \sim \rho_{\theta_i}$$
$$\theta_{i+1}^* = \theta_i + \gamma_{i+1} H(\theta_i, U_{i+1}, E_{i+1})$$
$$\theta_{i+1} = \theta_{i+1}^* 1_{\mathcal{R}_{i+1}}(\theta_{i+1}^*) + \theta_{i+1}^{\mathrm{proj}} 1_{\mathcal{R}_{i+1}^c}(\theta_{i+1}^*)$$

where, denoting $\mathcal{F}_i = \sigma(U_0, \theta_j, j \leq i)$, $\theta_{i+1}^{\mathrm{proj}}$ is a random variable measurable w.r.t $\mathcal{F}_i \vee \sigma(\theta_{i+1}^*)$.

Using the results in [10], we aim to show that the SA-generated sequence $\{\theta_i\}$ remains in a feasible set $\Theta$ and do not approach $\partial \Theta$ with probability one for arbitrary initialization $(\theta_0, u) \in \mathcal{R}_0 \times \mathbb{U}$ under appropriate conditions on $\{H(\theta, u, e), (\theta, u, e) \in \Theta \times \mathbb{U} \times \mathbb{E}\}$, $\{P_\theta, \theta \in \Theta\}$ and $\{\mathcal{R}_i\}$. We denote throughout the probability distribution associated to the process $(\theta_i, U_i)_{i \geq 0}$ defined in Algorithm 1.1 and starting at $(\theta_0, U_0) \equiv (\theta, u) \in \Theta \times \mathbb{U}$ as $\mathbb{P}_{\theta,u}(\cdot)$ and the associated expectation as $\mathbb{E}_{\theta,u}[\cdot]$. The approach developed in [10] relies on the existence of a Lyapunov function $w : \Theta \to [0, \infty)$ for the recursion on $\theta$ and the subsequent proof that $\{w(\theta_i)\}$ is $\mathbb{P}_{\theta,u}$-a.s. under some adequate level. For any $M > 0$, we define the level sets $\mathcal{W}_M := \{\theta \in \Theta : w(\theta) \leq M\}$. Following [10], we consider the following assumptions:

**SA1.** *There exists a continuously differentiable function $w : \Theta \to [0, \infty)$ such that*

(i) *For all $\theta, \theta' \in \Theta$,*

$$\|\nabla w(\theta) - \nabla w(\theta')\| \leq C_w \|\theta - \theta'\| \,.$$

(ii) *the projection sets are increasing subsets of $\Theta$, that is, $\mathcal{R}_i \subset \mathcal{R}_{i+1}$ for all $i \geq 0$, and*

$$\hat{\Theta} := \bigcup_{i=0}^{\infty} \mathcal{R}_i \subset \Theta \,,$$

(iii) *there exists a constant $M_0 > 0$ such that for any $\theta \in \mathcal{W}_{M_0}^c \cap \hat{\Theta}$*

$$\langle \nabla w(\theta), h(\theta) \rangle \leq 0$$

(iv) *the family of random variables $\left\{ \theta_i^{\mathrm{proj}} \right\}_{i \geq 1}$ satisfies for all $i \geq 1$ whenever $\theta_i^* \notin \mathcal{R}_i$*

$$\theta_i^{\mathrm{proj}} \in \mathcal{R}_i \quad \textit{and} \quad w\left( \theta_i^{\mathrm{proj}} \right) \leq w\left( \theta_i^* \right) \quad \mathbb{P}_{\theta, u} - a.s..$$

(v) *there exists constants $c \in [0, \infty)$ and a non-decreasing sequence of constants $\zeta_i \in [1, \infty)$ satisfying $\sup_{\theta \in \mathcal{R}_i} |\nabla w(\theta)| \leq c\zeta_i$ for all $i \geq 0$.*

Following [10], we introduce $\bar{H}(\theta, u, e) := H(\theta, u, e) - h(\theta)$. We need to impose some additional constraints on the noise sequence:

**SA2.** *For any $(\theta, u) \in \mathcal{R}_0 \times \mathbb{U}$ it holds that*

(i) $\mathbb{P}_{\theta, u} \left( \lim_{i \to \infty} \gamma_{i+1} \|\nabla w(\theta_i)\| \cdot \|H(\theta_i, U_{i+1}, E_{i+1})\| = 0 \right) = 1$,

(ii) $\mathbb{E}_{\theta, u} \left[ \sum_{i=0}^{\infty} \gamma_{i+1}^2 \|H(\theta_i, U_{i+1}, E_{i+1})\|^2 \right] < \infty$,

(iii) $\mathbb{E}_{\theta, u} \left[ \sup_{k \geq 0} \left| \sum_{i=0}^{k} \gamma_{i+1} \left\langle \nabla w(\theta_i), \bar{H}(\theta_i, U_{i+1}, E_{i+1}) \right\rangle \right| \right] < \infty$.

(iv) $\lim_{\theta \to \partial \hat{\Theta}} w(\theta) = \infty$

**Theorem 21.** *Assume SA1-SA2. Then, for any $(\theta, u) \in \mathcal{R}_0 \times \mathrm{U}$*

$$\mathbb{P}_{\theta, u} \left( \limsup_{i \to \infty} w(\theta_i) < \infty \right) = 1.$$

*Proof.* The proof is a simple adaptation of [10, Theorem 2.5]. $\qquad\square$

The condition $\lim_{\theta \to \partial \hat{\Theta}} w(\theta) = \infty$ is weakened in [10, Section 2.2]. Verifiable conditions implying **SA2** are given in [10, Section 3, Condition 3.1]. They are summarized in the next assumption. In the assumptions below, it is implicitly assumed that **SA1** holds with constants $(\zeta_i)_{i \geq 0}$.

We denote $\tilde{H}(\theta, u) = \int \bar{H}(\theta, u, e) \rho(\mathrm{d}e)$ and we consider the following assumptions:

**SA3.** *For all $\theta \in \hat{\Theta}$, the solution $g_\theta : \mathbb{U} \to \Theta$ to the Poisson equation $g_\theta(u) - P_\theta g_\theta(u) \equiv \tilde{H}(\theta, u)$ exists and for all $i \geq 0$ the step size $\Gamma_{i+1}$ is independent of $\mathcal{F}_i$ and $U_{i+1}$. Moreover, there exist a measurable function $V : \mathbb{U} \to [1, \infty)$ and constants $c < \infty, \beta_H, \beta_g \in [0, 1/2]$ and $\alpha_g, \alpha_H, \alpha_V \in [0, \infty)$ such that for all $(\theta, u) \in \mathcal{R}_0 \times \mathbb{U}$*

(i) $\sup_{\theta \in \mathcal{R}_i} |\tilde{H}(\theta, u)| \leq c\zeta_i^{\alpha_H} V^{\beta_H}(u)$,

(ii) $\mathbb{E}_{\theta, u} [V(U_i)] \leq c\zeta_i^{\alpha_V} V(u)$,

(iii) $\sup_{\theta \in \mathcal{R}_i} [|g_\theta(u)| + |P_\theta g_\theta(u)|] \leq c\zeta_i^{\alpha_g} V^{\beta_g}(u)$,

(iv) $\sum_{i=1}^{\infty} \gamma_{i+1} \zeta_i \mathbb{E}_{\theta, u} \left[ \left| P_{\theta_i} g_{\theta_i}(U_i) - P_{\theta_{i-1}} g_{\theta_{i-1}}(U_i) \right| \right] < \infty$,

(v) $\sum_{i=1}^{\infty} \gamma_i^2 \zeta_i^{2+2((\alpha_H + \beta_H \alpha_V) \vee (\alpha_g + \beta_g \alpha_V))} < \infty$,

(vi) $\sum_{i=1}^{\infty} \gamma_{i+1} \gamma_i \zeta_i^{\alpha_H + \alpha_g + (\beta_H + \beta_g)\alpha_V} < \infty$,

(vii) $\sum^{\infty} |\gamma_{i+1} - \gamma_i| \zeta_i^{1+\alpha_g + \beta_g \alpha_V} < \infty$.

For geometrically ergodic Markov chain, these conditions may be shown to boil down to "uniform-in-$\theta$" geometric ergodicity conditions and "smoothness" of the mapping $\theta \mapsto P_\theta$.

**MC1.** *For any $r \in (0,1]$ and any $\theta \in \hat{\Theta}$, there exist constants $M_{\theta,r} \in [0,\infty)$ and $\rho_{\theta,r} \in (0,1)$, such that for any function $\|f\|_{V^r} < \infty$*

$$\left| P_\theta^k f(u) - \mu_\theta(f) \right| \leq V^r(u) \|f\|_{V^r} M_{\theta,r} \rho_{\theta,r}^k$$

*for all $k \geq 0$ and all $u \in \mathbb{U}$. Moreover, it holds that $\sup_{\theta \in \mathcal{R}_i} M_{\theta,r} \leq c_r \zeta_i^{\alpha_M}$ and $\sup_{\theta \in \mathcal{R}_i} (1 - \rho_{\theta,r})^{-1} \leq c_r \zeta_i^{\alpha_\rho}$.*

**MC2.** *For any $\theta, \theta' \in \hat{\Theta}$, there exist a constant $D_{\theta,\theta',r} \in [0,\infty)$ and a constant $\beta_D \in (0,\infty)$ independent of $\theta, \theta'$ and $r$ such that for any function $\|f\|_{V^r} < \infty$*

$$\|P_\theta f - P_{\theta'} f\|_{V^r} \leq \|f\|_{V^r} D_{\theta,\theta',r} |\theta - \theta'|^{\beta_D} .$$

*Moreover, $\sup_{(\theta,\theta') \in \mathcal{R}_i^2} D_{\theta,\theta',r} \leq c_r^D \zeta_i^{\alpha_D}$ for some constant $c_r^D \in [0,\infty)$ depending only on $r \in (0,1]$*

**MC3.** *SA3-(i) and (ii) hold with constants $\alpha_H, \beta_H$ and $\alpha_V$, and there exist constants $c < \infty, \alpha_\Delta \in [0,\infty)$ and $\beta_\Delta > 0$ such that*

$$\sup_{(\theta,\theta') \in \mathcal{R}_i^2} \left\| \tilde{H}(\theta, \cdot) - \tilde{H}(\theta', \cdot) \right\|_{V^{\beta_H}} \leq c \zeta_i^{\alpha_\Delta} |\theta - \theta'|^{\beta_\Delta} .$$

Up to this point, we have only considered the stability of the stochastic approximation process with expanding projections. Indeed, after showing the stability we know that the projections can occur only finitely often (almost surely), and the noise sequence can typically be controlled. Given this, the stochastic approximation literature provides several alternatives to show the convergence; see [41, 15]. We formulate below a convergence result following from [7].

**SA4.** *The set $\Theta \subset \mathbb{R}^d$ is open, the mean field $h : \Theta \to \mathbb{R}^d$ is continuous, and there exists a continuously differentiable function $\hat{w} : \Theta \to [0,\infty)$ such that*

(i) *there exists a constant $M_0 > 0$ such that*

$$\mathcal{L} := \{\theta \in \Theta : \langle \nabla \hat{w}(\theta), h(\theta) \rangle = 0\} \subset \{\theta \in \Theta : \hat{w}(\theta) < M_0\}$$

(ii) *there exists $M_1 \in (M_0, \infty]$ such that $\{\theta \in \Theta : \hat{w}(\theta) \leq M_1\}$ is compact.*
(iii) *for all $\theta \in \Theta \setminus \mathcal{L}$, the inner product $\langle \nabla \hat{w}(\theta), h(\theta) \rangle < 0$ and the closure of $\hat{w}(\mathcal{L})$ has an empty interior.*

**Theorem 22.** *Assume SA4 holds, and let $\mathcal{K} \subset \Theta$ be a compact set intersecting $\mathcal{L}$, that is, $\mathcal{K} \cap \mathcal{L} \neq \varnothing$. Suppose that $(\gamma_i)_{i \geq 1}$ is a sequence of non-negative real numbers satisfying $\lim_{i \to \infty} \gamma_i = 0$ and $\sum_{i=1}^\infty \gamma_i = \infty$. Consider the sequence $(\theta_i)_{i \geq 0}$ taking values in $\Theta$ and defined through the recursion $\theta_i = \theta_i - 1 + \gamma_i h(\theta_{i-1}) + \gamma_i \varepsilon_i$ for all $i \geq 1$, where $(\varepsilon_i)_{i \geq 1}$ take values in $\mathbb{R}^d$. If there exists an integer $i_0$ such that $\{\theta_i\}_{i \geq i_0} \subset \mathcal{K}$ and $\lim_{m \to \infty} \sup_{n \geq m} |\sum_{i=m}^n \gamma_i \varepsilon_i| = 0$, then $\lim_{n \to \infty} \inf_{x \in \mathcal{L} \cap \mathcal{K}} \|\theta_n - x\| = 0$.*

We have now all the necessary elements to prove Theorem 3. For simplicity, we set $\alpha_k = \alpha_\infty$ for any $k \in \mathbb{N}$ and $\gamma_k = 1/(1+k)^\iota$ where $\iota \in (1/2, 1]$. In this case, the state space is $\mathbb{U} = \mathbb{X}^M$ and $\mathbb{E} = \mathbb{Z}^{(N-1)\cdot M}$, $U_k = (Y_k[j])_{j=1}^M$, $E_k = (Z_k^{2:N}[j])_{j=2}^N$. With $u = (y[j])_{j=1}^M$ and $e = (z^{2:N}[j])_{j=1}^M$, $H(\theta, u, e)$ is given by

$$H(\theta, u) = M^{-1} \sum_{mj=1}^N \left\{ \alpha_\infty H^f(\theta, y[j], z^{2:N}[j]) + (1 - \alpha_\infty) H^b(\theta, z^{2:N}[j]) \right\} .$$

where $H^f$ and $H^b$ are defined respectively in (4) and (5). In this case, the Markov kernel $P_\theta$ is given for any nonnegative function $f$,

$$P_{\theta,N} f(y[1], \ldots, y[M]) = \int \prod_{j=1}^N \mathsf{K}_{\theta,N}(y[j], \mathrm{d}\tilde{y}[j]) f(\tilde{y}[1], \ldots, \tilde{y}[M]) ,$$

and $\mathsf{K}_{\theta,N}$ is defined in (2.2) with $\lambda \leftarrow \lambda_\theta$ and $w \leftarrow w_\theta$. By construction, for any $\theta \in \Theta$, $P_\theta$ has a unique stationary distribution which is given by $\mu = \pi^{\otimes M}$. Using Theorem 6, and, for all $\theta \in \Theta$,

$$H^f(\theta, x^{1:N}) = \Pi_{\theta,N}[\nabla_\theta \log \lambda_\theta](x^{1:N})$$

we get that

$$h(\theta) = -\alpha_\infty \nabla_\theta \mathrm{KL}(\pi||\lambda_\theta) - (1-\alpha_\infty)\nabla_\theta \mathrm{KL}(\lambda_\theta||\pi) .$$

Recall that $\Theta = \mathbb{R}^q$. To check **SA**1, we set

$$w(\theta) = \alpha_\infty \mathrm{KL}(\pi||\lambda_\theta) - (1-\alpha_\infty)\mathrm{KL}(\lambda_\theta||\pi) , \text{ for } \theta \in \Theta.$$

and for $i \in \mathbb{N}$, $\zeta_i = \log(i+1)$. The subset $\mathcal{R}_i$ is a ball centered at 0 and of radius $r_i$ where $r_i$ is chosen so that $\sup_{\|\theta\| \le r_i} \nabla w(\theta)\| \le c\zeta_i$ (such $r_i$ exists using **A**3). It is easily checked that **SA**1 is satisfied thanks to **A**3 (note in particular that $\nabla w$ is globally Lipshitz under the stated conditions). Conditions **SA**3-(v)-(vi)-(vii) are automatically satisfied.

We choose the drift function for the Markov kernel $P_{\theta,N}$ as

$$V(y[1], \ldots, y[M]) = \sum_{i=1}^{M} V(y[i]) ,$$

where $V$ is the drift function in **A**1. **MC**1 follows from Theorem 2 under **A**4. It is important to note that it is essential to have explicit controls on the drift and reduction conditions here. Conditions **MC** 2 and **MC**2 follow from **A**3. The precise tuning of constants is done along the same lines as [10, Section 5.3].

# E  Numerical experiments

## E.1  Metrics

**ESTV**  To compute Empirical sliced total variation distance (ESTV), we perform 25 random one-dimensional projections and then perform Kernel Density Estimation there for reference and produced samples. We then take the TV-distance between two distributions over $1D$ grids of 1000 points. We consider the value averaged over the projections to show the divergence between the MCMC distribution and the reference distribution.

**EMD**  We compute the EMD as the transport cost between sample and reference points in $L_2$ using the algorithm proposed in [14]. Then we report the EMD rescaled by the target dimension $d$.

**ESS**  ESS (effective sample size) measures how many independent samples from target yield (approximately) the same variance for estimating the mean of some function. The closer ESS is to 1, the better is the sampler. Following [26], we compute ESS component-wise for multivariate distributions. Namely, given a sample $\{Y_t\}_{t=1}^{M}, Y_t \in \mathbb{R}^d$ of size $M$, for $i = 1, \ldots, d$, we compute

$$\mathrm{ESS}_i = \frac{1}{1 + \sum_{k=1}^{M} \rho_k^{(i)}} .$$

Here $\rho_k^{(i)} = \frac{\mathrm{Cov}(Y_{t,i}, Y_{t+k,i})}{\mathrm{Var}(Y_{t,i})}$ is the autocorrelation at lag $k$ for $i-$th component. We replace $\rho_k^{(i)}$ by its sample counterpart $\widehat{\rho}_k^{(i)}$, an report $\mathrm{ESS} = d^{-1} \sum_{i=1}^{d} \widehat{\mathrm{ESS}}_i$, where

$$\widehat{\mathrm{ESS}}_i = \frac{1}{1 + \sum_{k=1}^{M} \widehat{\rho}_k^{(i)}} .$$

## E.2  Unimodal Gaussian target and impact of dimension

With the simple experiment presented on Figure 7, we illustrate the sensitivity of the purely global i-SIR to the match between the proposal and target, which typically worsens with dimension. Namely, the rate $\kappa_N$ can be close to 1 when the dimension $d$ is large, even when the restrictive condition that weights are uniformly bounded $|w|_\infty < \infty$ is satisfied.

To illustrate this phenomenon, we consider a simple problem of sampling from the standard normal distribution $\mathcal{N}(0, \mathrm{I}_d)$ with the proposal $\mathcal{N}(0, 2\,\mathrm{I}_d)$ in increasing dimensions $d$ up to 300. Results visualized in Figure 7 show that the performance of vanilla i-SIR quickly deteriorates as most proposals get rejected. This problem can be tackled by using the Explore-Exploit strategy coupling i-SIR with local MCMC steps to define a new sampler. This simple experiment previously considers $\mathrm{Ex}^2\mathrm{MCMC}$ with MALA applied as R.

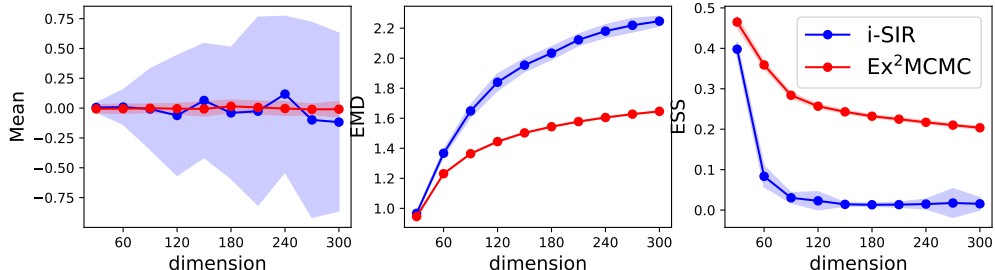

Figure 7: Sampling from $\mathcal{N}(0, \mathrm{I}_d)$ with the proposal $\mathcal{N}(0, 2\,\mathrm{I}_d)$. – See Appendix E.1 for the definitions of EMD and ESS metrics. We display confidence intervals for i-SIR and Ex²MCMC obtained from 100 independent runs as blue and red regions, respectively. Ex²MCMC helps to achieve efficient sampling even in high dimensions.

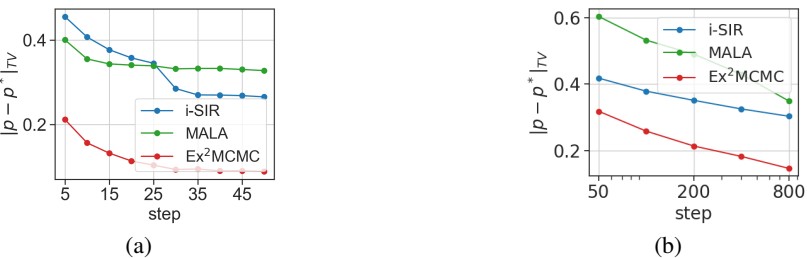

Figure 8: Inhomogeneous 2d Gaussian mixture. – Quantitative analysis during burn-in of parallel chains (a, $M = 500$ chains KDE) and for after burn-in for single chains statistics (b, $M = 100$ average).

### E.3 Mixtures of Gaussians

**Equally weighted Gaussians in two dimension**   The target density is

$$p_\beta(x) \propto \sum_{i=1}^{3} \beta_i \exp\{-\|x - \mu_i\|^2/(2\sigma^2)\} \,. \tag{26}$$

Here we choose $\sigma = 1$, $\beta_i = 1/3$, and $\mu_i$, $i \in \{1, 2, 3, \}$ as vertices of an equilateral triangle with side length $4\sqrt{3}$ and center $(0,0)$. The contour representation of (26) can be found in Figure 1a. We compare 3 sampling strategies:

- i-SIR algorithm with $N = 3$ particles and $\mathcal{N}(0, 4\,\mathrm{I})$ proposal distribution;
- MALA with step size $\gamma = 0.5$, tuned to obtain acceptance rate  0.67;
- Ex²MCMC algorithm with the same parameters as i-SIR and 3 consecutive MALA steps with $\gamma = 0.5$ as rejuvenations.

We generate 100 observations within each sampler and represent them in Figure 1a. For the MALA sampler, we generate 300 samples and select every 3th to maintain compatibility with the Ex²MCMC setup. Note that in this example, the variance of the global proposals in i-SIR should be relatively large to cover well all modes of the (26) mixture. However, since the modes are narrow, the step size of MALA cannot be very large to obtain a sensible acceptance rate. Therefore, Figure 1a shows the drawbacks of the two approaches: i-SIR covers all modes of the target, but the chain often gets stuck at a certain point, which affects the variability of the samples. MALA allows a better local exploration of each mode, but does not cover the whole support of the target. The Ex²MCMC algorithm combines the advantages of both methods by combining i-SIR-based global exploration with MALA -based local exploration.

Now, the mixture model of (26) is modified with the weights parameters $\beta = (\beta_1, \beta_2, \beta_3) = (2/3, 1/6, 1/6)$ and same values of $\mu_i$ and $\sigma$. To compare the quality of the methods, we perform the following procedure

- starting with the initial distribution $\mathcal{N}(0, 4\,\mathrm{I})$, we generate the trajectory $(X_1, \ldots, X_n)$ for different values of $n \in [25, 800]$ for each of the compared methods (i-SIR, MALA, Ex$^2$MCMC ). Sampler hyperparameters are the same as above, and the burn-in period equals 50;
- We perform the kernel density estimate (KDE) $\widehat{p}_n$ based on the observations $(X_1, \ldots, X_n)$, and compute the total variation distance between $\widehat{p}_n$ and the target density $p_\beta$, and the forward $\mathrm{KL}(\widehat{p}_n \| p_\beta)$. Then we average the results over 100 independent runs of each sampler.

Now we use the same values for the means and covariances but set the mixing weights to $\beta = (\beta_1, \beta_2, \beta_3) = (2/3, 1/6, 1/6)$. To compare the different sampling methods, we perform the following procedure.

- starting from the initial distribution $\mathcal{N}(0, 4\,\mathrm{I})$, we generate the trajectory $(X_1, \ldots, X_n)$ for different values of $n \in [25, 800]$ for each of the compared methods (i-SIR, MALA, Ex$^2$MCMC ). The hyperparameters of the sampler are the same as above, and the burn-in period is 50;
- We perform kernel density estimation (KDE) $\widehat{p}_n$ based on the observations $(X_1, \ldots, X_n)$ and calculate the total variation distance between $\widehat{p}_n$ and the target density $p_\beta$, as well as the forward value $\mathrm{KL}(\widehat{p}_n \| p_\beta)$. We then average the results over 100 independent runs of each sampler.

The results for each sampler are given in Figure 1c, Figure 8b. We also provide a simple illustration to the statements of (2) and Theorem 2. Starting from the initial distribution $\xi \sim \mathcal{N}(0, 4\,\mathrm{I})$, we draw 500 independent chains of length 50 for each of the compared methods. Using these 500 observations, we create a KDE $\widehat{p}_n$ for the density corresponding to the distribution of $\xi Q^n$ for different $n \in \{5, \ldots, 50\}$ and Q corresponding to i-SIR, MALA or Ex$^2$MCMC. Then we calculate the total variation distance between $\widehat{p}_n$ and the target density $p_\beta$. Corresponding plots can be found in Figure 1b, Figure 8a. Note that Ex$^2$MCMC significantly outperforms the results of both MALA and i-SIR. Indeed, the inhomogeneous mixture model is a complicated target for the Langevin-based methods. The trajectories generated by MALA tend to remain in a single mode of mixture (26), which reduces the reliability of the estimates and requires the generation of long trajectories even for $d = 2$. At the same time, it is difficult for i-SIR type methods without local exploration trajectories to quickly cover all the modes.

## E.4 Normalizing flow RealNVP

We use the RealNVP architecture ([20]) for our experiments with adaptive MCMC. The key element of RealNVP is a coupling layer, defined as a transformation $f : \mathbb{R}^D \to \mathbb{R}^D$:

$$y_{1:d} = x_{1:d}$$
$$y_{d+1:D} = x_{d_1:D} \odot \exp(s(x_{1:d})) + t(x_{1:d})$$

where $s$ and $t$ are some functions from $\mathbb{R}^D$ to $\mathbb{R}^D$. Thus, it is clear that the Jacobian of such a transformation is a triangular matrix with nonzero diagonal terms. We use fully connected neural networks to parameterize the functions $s$ and $t$.

In all experiments with normalizing flows, we use the optimizer Adam ([38]) with $\beta_1 = 0.9$, $\beta_2 = 0.999$ and weight decay 0.01 to avoid overfitting.

## E.5 High-dimensional multi-modal distribution

In an additional experiment we consider a high-dimensional toy target distribution: a Gaussian mixture similar as Appendix E.3 above in $50d$. Modes are equally weighted, isotropic and well-separated.

A purely local sampler would not mix between modes, as in the $2d$ case. A unimodal Gaussian proposal also fails in large dimension because of the concentration of the target measure in a small fraction of the proposal's bulk. Hence we only examine the performance of FlEx$^2$MCMC. We set the number of proposals per iterations to $N = 20$.

Using a RealNVP flow, we compare in Figure 9 the different outcomes depending on the choices of initialization of the MCMC walkers and training loss. Training the proposal offline through uniquely the backward KL (i.e. $\alpha = 0$ in the combinaison of KL losses) is typically unstable in this multimodal case and the network collapse on the first detected mode. Successful backward-KL training is probably possible, yet at the cost of designing a proper annealing schedule of the target distribution as in [76]. Resorting instead to a loss involving the forward KL ($\alpha = 0.9$ in this experiment), mixing

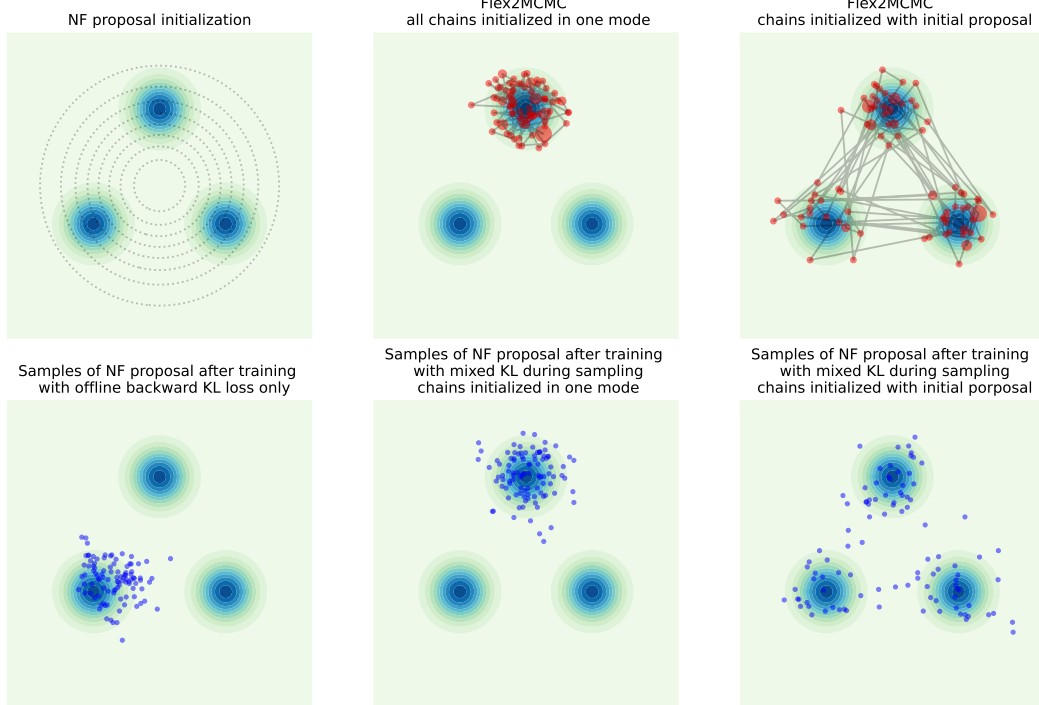

Figure 9: Importance of initialization and forward KL loss for multi-modal high-dimensional targets - All panels are $2d$ projections of a $50d$ Euclidian space, with a target mixture of 3 isotropic Gaussian. Using a normalizing flow proposal distribution initialized as an isotropic Gaussian covering the 3 modes (top left), training with backward KL loss only still typically leads to mode collapse on one of the modes (bottom left). Running instead the simulataneous training and sampling of FlEx$^2$MCMCwith the mixture of backward and forward KL loss can lead to successful mixing between distant modes (top and bottom right), yet at the condition that chains are initialized such that all modes can be reached by the local-rejuvenation kernel- which is here enforced by an initialization as random draws of the initial proposal. Conversely, if all the chains are initialized in a single mode, the forward-KL estimated with states visited by the chains will not prevent a mode collapse (top and bottom center panels).

between the well separated modes in high-dimension is possible, provided that chain initialization ensures that all modes can be reached by the local kernel.

To summarize, the choice of loss composition depends on the information a priori available on the considered target distribution. If rough location of modes is available - as it might be the case in chemistry applications where isomers of interest are known but sampling is necessary for relative free energy calculations - relying on the forward KL to draw the proposal to the modes is a simple and efficient strategy. Conversely, if little is known, there is no free lunch with the local-global kernels and an annealing might be necessary to train the global proposal, possibly using only the backward KL loss.

### E.6 Distributions with complex geometry

In this section, we study the sampling quality from high-dimensional distributions, whose density levels have high curvature (Banana shaped and Funnel distributions, details below). With such distributions, standard MCMC algorithms like MALA or i-SIR, fail to explore fully the density support.

The corresponding densities are given for $x \in \mathbb{R}^d$ by

$$p_f(x) = \mathrm{Z}^{-1} \exp\left(-x_1^2/2a^2 - (1/2)\mathrm{e}^{-2bx_1}\sum\nolimits_{i=2}^d\{x_i^2 + 2bx_1\}\right), \quad d \geq 2,$$

$$p_b(x) = \mathrm{Z}^{-1} \exp\left(-\sum\nolimits_{i=1}^{d/2}\{x_{2i}^2/2a^2 - (x_{2i-1} - bx_{2i}^2 + a^2b)^2/2\}\right), \quad d = 2k, k \in \mathbb{N}.$$

(27)

where Z is a normalizing constant. We set $a = 2$, $b = 0.5$ for funnel and $a = 5$, $b = 0.02$ for banana-shape distributions, respectively. For MALA we use an adaptive step size tuning strategy to maintain acceptance rate approximately 0.5. For i-SIR and $\mathrm{Ex}^2\mathrm{MCMC}$ algorithms we use wide Gaussian global proposal $\mathcal{N}(0, \sigma_p^2\,\mathrm{I})$ with $\sigma_p^2 = 4$ for Funnel and $\sigma_p^2 = 9$ for Banana-shape distribution.

For $\mathrm{FlEx}^2\mathrm{MCMC}$ use a simple RealNVP-based normalizing flow [20] with 4 hidden layers. Note that for $p_f(x)$ the energy landscape in the region with $x_1 < 0$ is steep, so the distributions (27) are hard to capture, especially when the dimension $d$ is large. Moreover, due to the complex geometry of the distribution support, we cannot hope that local samplers (MALA) or global samplers (i-SIR ) alone will give good results. In this example, we want to compare $\mathrm{FlEx}^2\mathrm{MCMC}$ with i-SIR  MALA and the HMC-based NUTS sampler [35]. We also add a vanilla version of the $\mathrm{Ex}^2\mathrm{MCMC}$ algorithm to the comparison. To generate the ground-truth samples, we use the explicit reparametrisation of (27). Indeed, given a random vector $(Z_1, \ldots, Z_d) \sim \mathcal{N}(0, \mathrm{I})$, we consider its transformation $(X_1, \ldots, X_d)$ under the formulas

$$\begin{cases} X_1 = aZ_1 \\ X_i = \mathrm{e}^{bX_1}Z_i, \quad i \in \{2, \ldots, d\}. \end{cases}$$

It is easy to check that $(X_1, \ldots, X_d)$ follows the density $p_f(x), x \in \mathbb{R}^d$. Similarly, for $d = 2k$ consider the transformation

$$\begin{cases} Y_{2i} = aZ_{2i} \\ Y_{2i-1} = Y_{2i} + bY_{2i}^2 - ba^2, \quad i \in \{1, \ldots, k\}. \end{cases}$$

Then $(Y_1, \ldots, Y_d)$ follows the density $p_b(x), x \in \mathbb{R}^d$. We provide the average computation time for NUTS, adaptive i-SIR and $\mathrm{FlEx}^2\mathrm{MCMC}$ algorithms in Table 1 and Table 2 for the Funnel and Banana-shape distributions, respectively, averaged over 50 runs. Note that different runs of NUTS algorithm yields high variance of the running time, especially for the Funnel distribution and dimensions $d \geq 50$.

We give the computation time for the above algorithms and additional implementation details in Appendix E.6. The implementation of $\mathrm{FlEx}^2\mathrm{MCMC}$ is based on the use of 5 MALA steps as rejuvenation steps.

| Method | $d = 10$ | $d = 20$ | $d = 50$ | $d = 100$ | $d = 200$ |
|---|---|---|---|---|---|
| NUTS | $33.4 \pm 8.2$ | $41.1 \pm 12.3$ | $61.6 \pm 30.2$ | $82.3 \pm 73.2$ | $88.4 \pm 59.5$ |
| Adaptive i-SIR | $38.1 \pm 3.2$ | $39.4 \pm 2.8$ | $45.3 \pm 2.5$ | $59.8 \pm 0.7$ | $80.4 \pm 0.4$ |
| $\mathrm{FlEx}^2\mathrm{MCMC}$ | $46.8 \pm 3.2$ | $48.2 \pm 2.8$ | $54.2 \pm 2.5$ | $68.8 \pm 0.8$ | $89.5 \pm 0.5$ |

Table 1: Computational time for the Funnel distribution.

| Method | $d = 20$ | $d = 40$ | $d = 60$ | $d = 80$ | $d = 100$ |
|---|---|---|---|---|---|
| NUTS | $27.6 \pm 1.8$ | $32.1 \pm 1$ | $34.2 \pm 0.5$ | $35.2 \pm 0.5$ | $35.9 \pm 0.4$ |
| Adaptive i-SIR | $24.5 \pm 0.2$ | $26.8 \pm 0.3$ | $28.5 \pm 0.2$ | $30.1 \pm 0.2$ | $32.8 \pm 0.2$ |
| $\mathrm{FlEx}^2\mathrm{MCMC}$ | $39.3 \pm 0.5$ | $41.8 \pm 0.3$ | $43.5 \pm 0.3$ | $45.1 \pm 0.3$ | $47.8 \pm 0.4$ |

Table 2: Computational time for the Banana-shape distribution.

## E.7  GANs as energy-based models

### E.7.1  MNIST results

For this example, we consider both the Wasserstein GAN (WGAN) setup with energy function $E_W(z)$ and the classical Jensen-Shannon GAN with energy function $E_{JS}(z)$. In both cases, we use

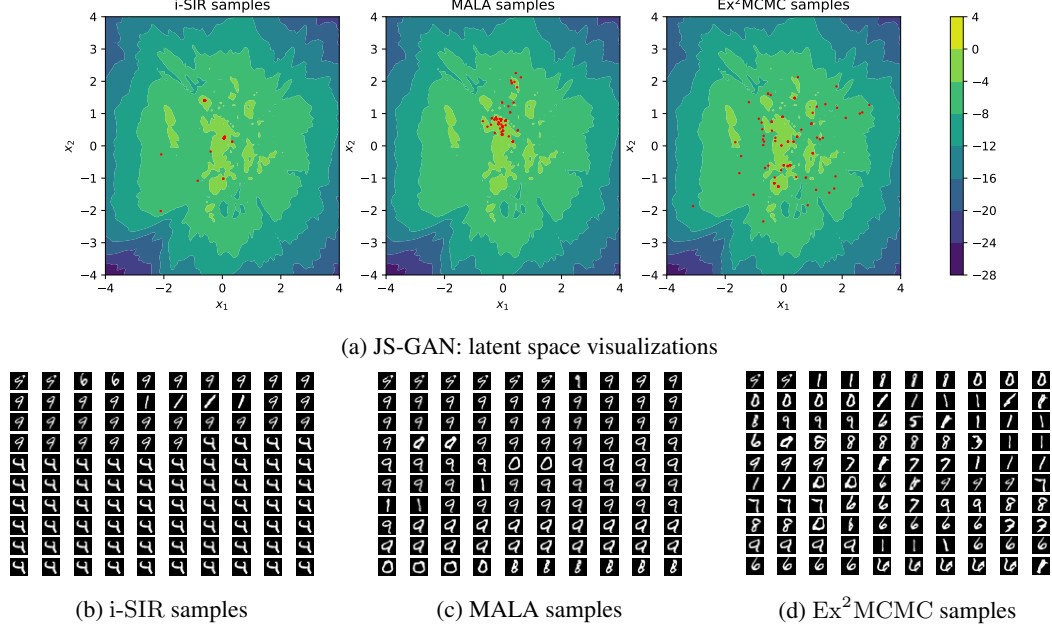

(a) JS-GAN: latent space visualizations

(b) i-SIR samples  (c) MALA samples  (d) Ex$^2$MCMC samples

fully connected networks with 3 convolutional layers for discriminator and 3 linear + 3 convolutional layers for generator. For WGAN training, we use gradient penalty regularisation, following [31]. We provide additional visualisations of the latent space and samples along a given trajectory for Jensen-Shannon GAN in Appendix E.7.1 and for Wasserstein GAN in Appendix E.7.1. Sampling hyperparameters are summarized in Table 3. For fair comparison, we take each 3-rd sample produced by the MALA, when running this algorithm separately. Both for WGAN-GP and vanilla GAN experiments we apply i-SIR and Ex$^2$MCMC with wide Gaussian global proposal $\mathcal{N}(0, \sigma_p^2)$. The particular values of $\sigma_p^2$ are specified in Table 3.

### E.7.2  Cifar-10 results

We consider two popular GAN architectures, DC-GAN [60] and SN-GAN [49]. Below we provide the details on experimental setup and evaluation for both of the models.

### E.8  Training and sampling details.

For DC-GAN and SN-GAN experiments, we took the implementation and training script of the models from Mimicry repository `https://github.com/kwotsin/mimicry`. Both models were trained on a single GPU GeForce GTX 1060 for approximately 20 hours.

Both for DC-GAN and SN-GAN, the latent dimension is equal to $d = 128$. Following [17], for both models we consider sampling from the latent spatial distribution

$$p(z) = \mathrm{e}^{-E_{JS}(z)}/Z \,, \quad z \in \mathbb{R}^d \,, \quad E_{JS}(z) = -\log p_0(z) - \mathrm{logit}\big(D(G(z))\big) \,,$$

where $\mathrm{logit}(y) = \log\left(y/(1-y)\right) y \in (0,1)$ is the inverse of the sigmoid function and $p_0(z) = \mathcal{N}(0, \mathrm{I})$.

**Evaluation protocol**  We perform $n = 100$ iterations of the algorithms MALA, i-SIR Ex$^2$MCMC and FlEx$^2$MCMCFor both the vanilla Ex$^2$MCMC algorithm (Algorithm 2) and FlEx$^2$MCMC we

| Method | # iterations | MALA step size $\gamma$ | # particles, $N$ | $\sigma_p^2$ | # MALA steps |
|--------|-------------|------------------------|------------------|-------------|--------------|
| JS-GAN | 100 | 0.02 | 10 | 9 | 3 |
| WGAN-GP | 100 | 0.02 | 10 | 9 | 3 |

Table 3: MNIST hyperparameters.

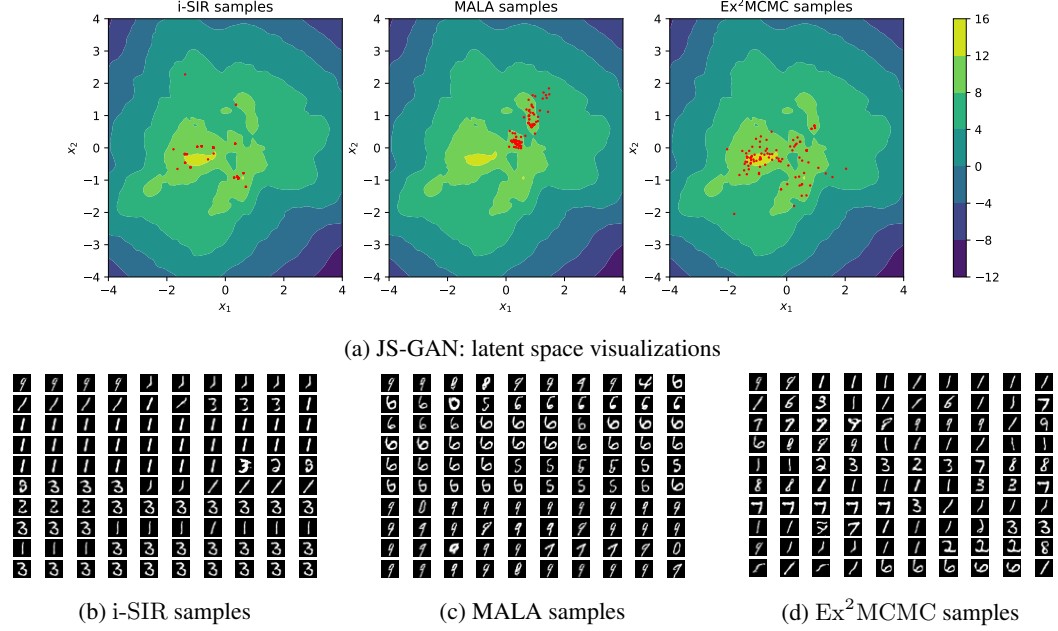

(a) JS-GAN: latent space visualizations

(b) i-SIR samples          (c) MALA samples          (d) Ex$^2$MCMC samples

use the Markov kernel (17), which corresponds to 3 MALA steps, as the rejuvenation kernel. The step size $\gamma$ given for the algorithm Ex$^2$MCMC corresponds to its rejuvenation kernel MALA. For more experimental details, see Table 4. For i-SIR and Ex$^2$MCMC algorithms we use $\mathcal{N}(0, \sigma_p^2 I)$ with $\sigma_p^2 = 1$ as a global proposal distribution.

We run $M = 500$ independent chains for each of the above MCMC algorithms. Then, for the $j-$th iteration, we compute the average value of the energy function $E(z)$ averaged over $M$ chains. Hyperparameters are specified in Table 4. Energy profiles for different algorithms for DC-GAN and SN-GAN are provided in Figure 17 and Figure 14, respectively. Note that in both cases Ex$^2$MCMC or FlEx$^2$MCMC algorithms yields lower energy samples. We visualize 10 randomly chosen trajectories obtained with each sampling methods in Figure 15-Figure 16 for SN-GAN and Figure 18-Figure 19 for DC-GAN, respectively. For each trajectory we visualize every 10-th sample. Both architectures indicate the same findings: MALA typically is not available to escape the mode of the corresponding target density $p(z)$ during one particular run. i-SIR travels well across the support of $p(z)$, yet the corresponding energy values are higher then the ones of Ex$^2$MCMC or FlEx$^2$MCMC. Some i-SIR trajectories can get trapped in one particular image due to the absence of local exploration moves. At the same time, Ex$^2$MCMC as illustrated in Figure 16-16a and Figure 19-19a, can both exploit the particular mode of the distribution and perform global moves over the support of $p(z)$. Of course, these global moves are more likely to occur during the first sampling iterations. For the DC-GAN architecture, we provide also the dynamics of FID (Frechet Inception Distance, [34]), and IS (Inception Score, [69]) values computed over 10000 independent trajectories. We plot the metrics in Figure 13a and Figure 13b. Metrics illustrate the image quality improvement achieved by FlEx$^2$MCMC and Ex$^2$MCMC algorithms.

| GAN type | # iterations | MALA step size $\gamma$ | # particles, $N$ | $\sigma_p^2$ | # MALA steps |
|---|---|---|---|---|---|
| SNGAN | 100 | $5 \times 10^{-3}$ | 10 | 1 | 3 |
| DCGAN | 100 | $10^{-3}$ | 10 | 1 | 3 |

Table 4: CIFAR-10 hyperparameters.

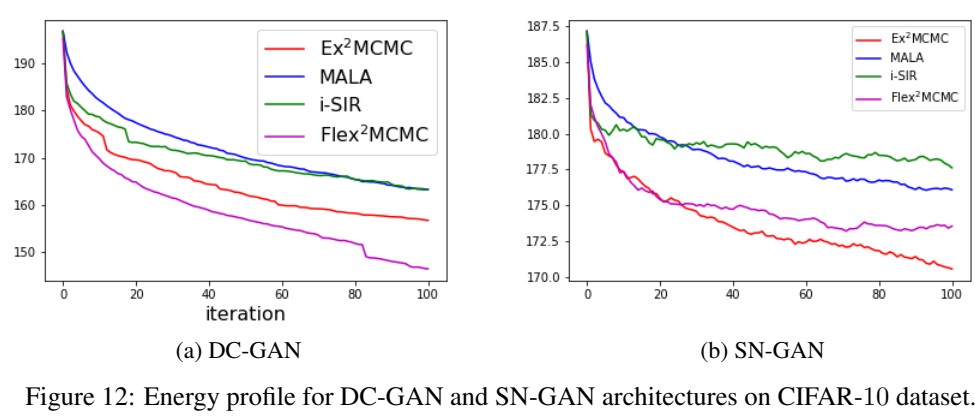

(a) DC-GAN

(b) SN-GAN

Figure 12: Energy profile for DC-GAN and SN-GAN architectures on CIFAR-10 dataset.

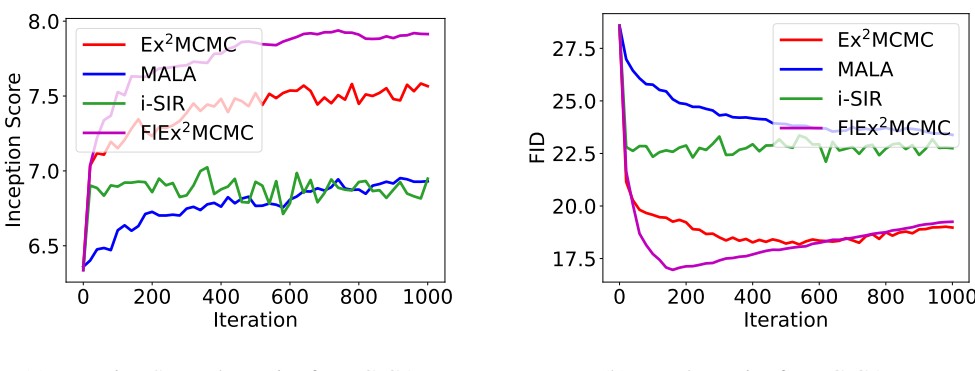

(a) Inception Score dynamics for DC-GAN

(b) FID dynamics for DC-GAN

Figure 13: Dynamics of Inception Score (a) and FID (b) computed over 10000 independent trajectories for DC-GAN trained on CIFAR-10 dataset.

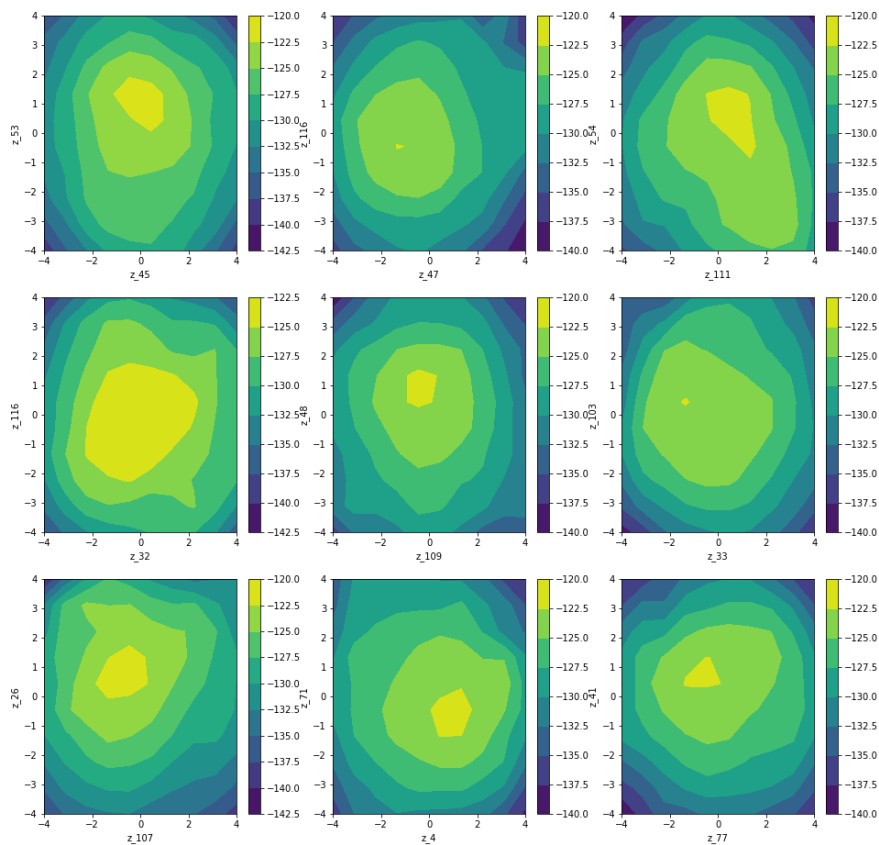

Figure 14: Energy profile for random axis pairs, SN-GAN

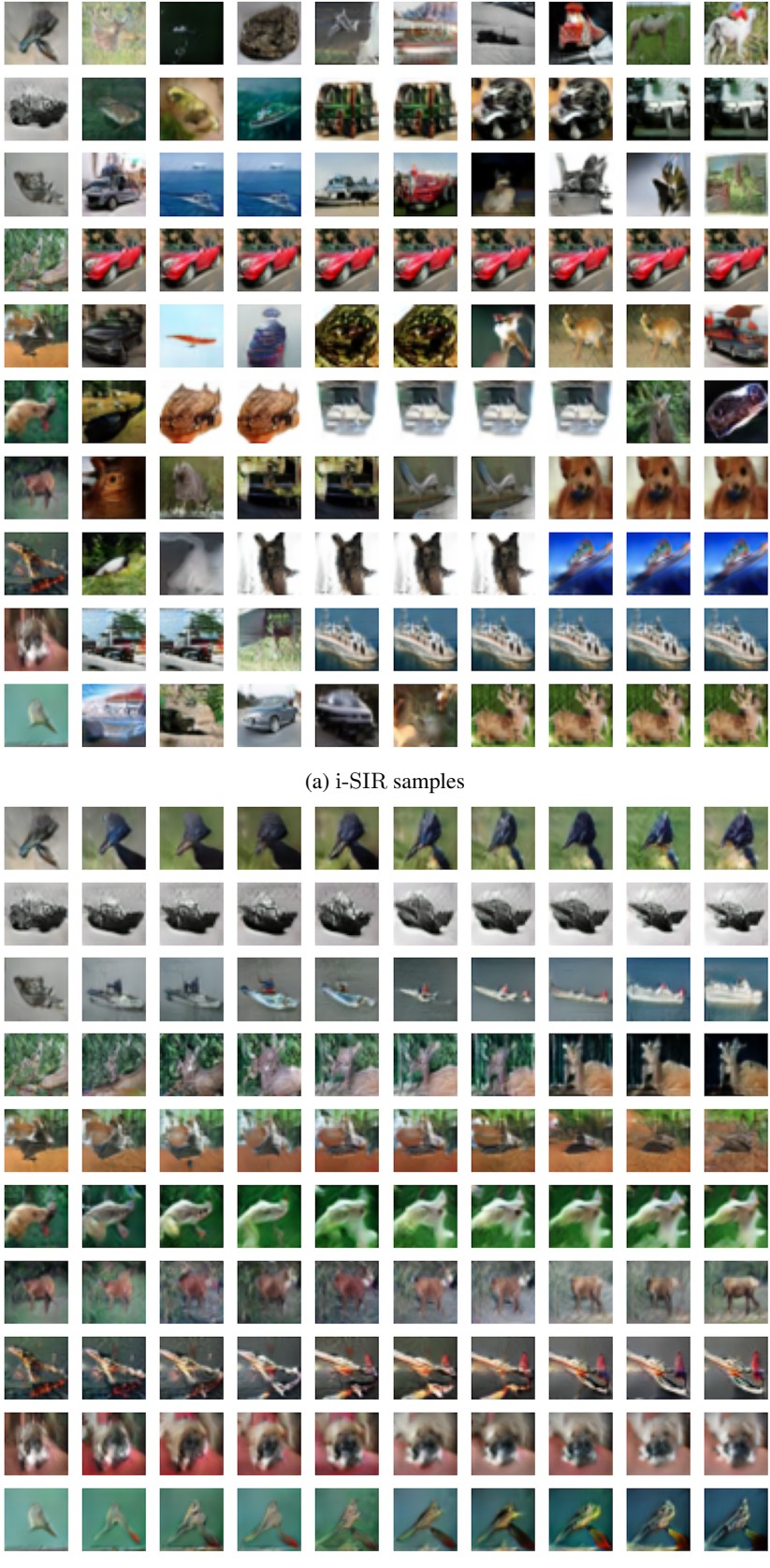

(a) i-SIR samples

(b) MALA samples

Figure 15: i-SIR and MALA samples, SN-GAN.

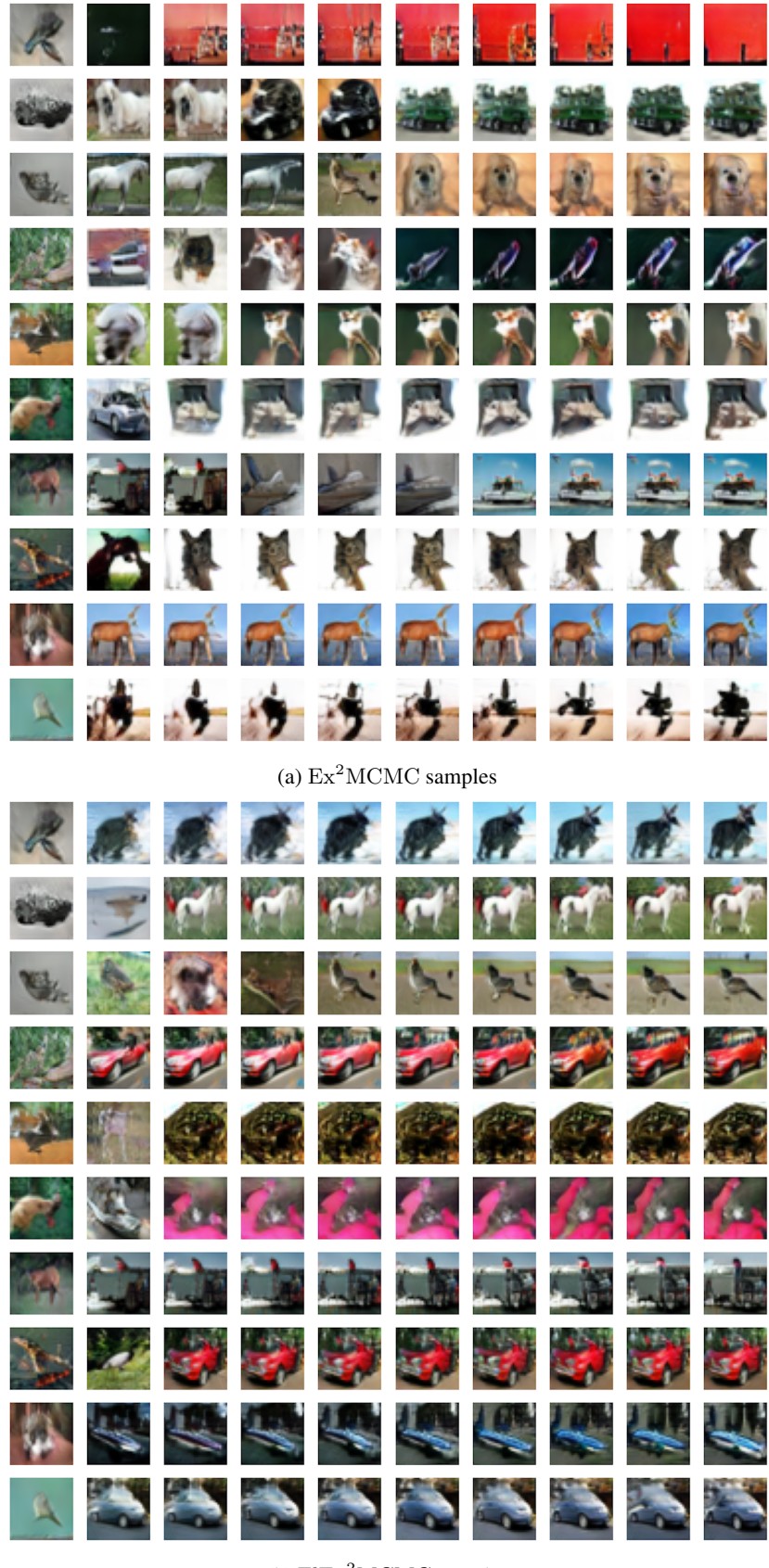

(a) Ex$^2$MCMC samples

(b) FlEx$^2$MCMC samples

Figure 16: Ex$^2$MCMC and FlEx$^2$MCMC samples, SN-GAN.

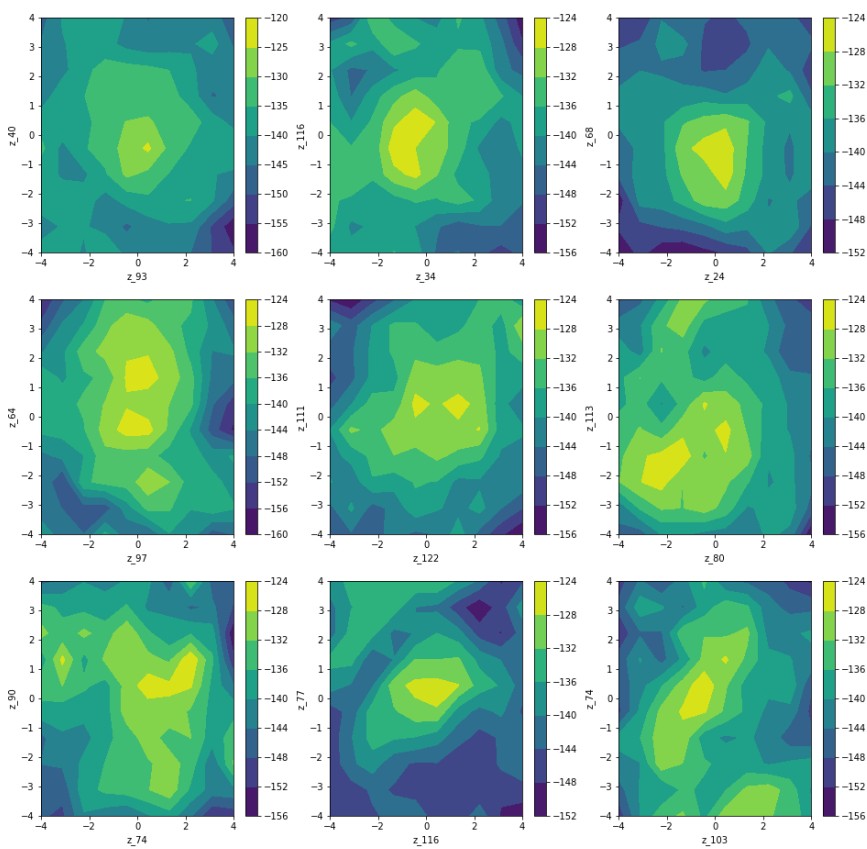

Figure 17: Energy profile for random axis pairs, DC-GAN

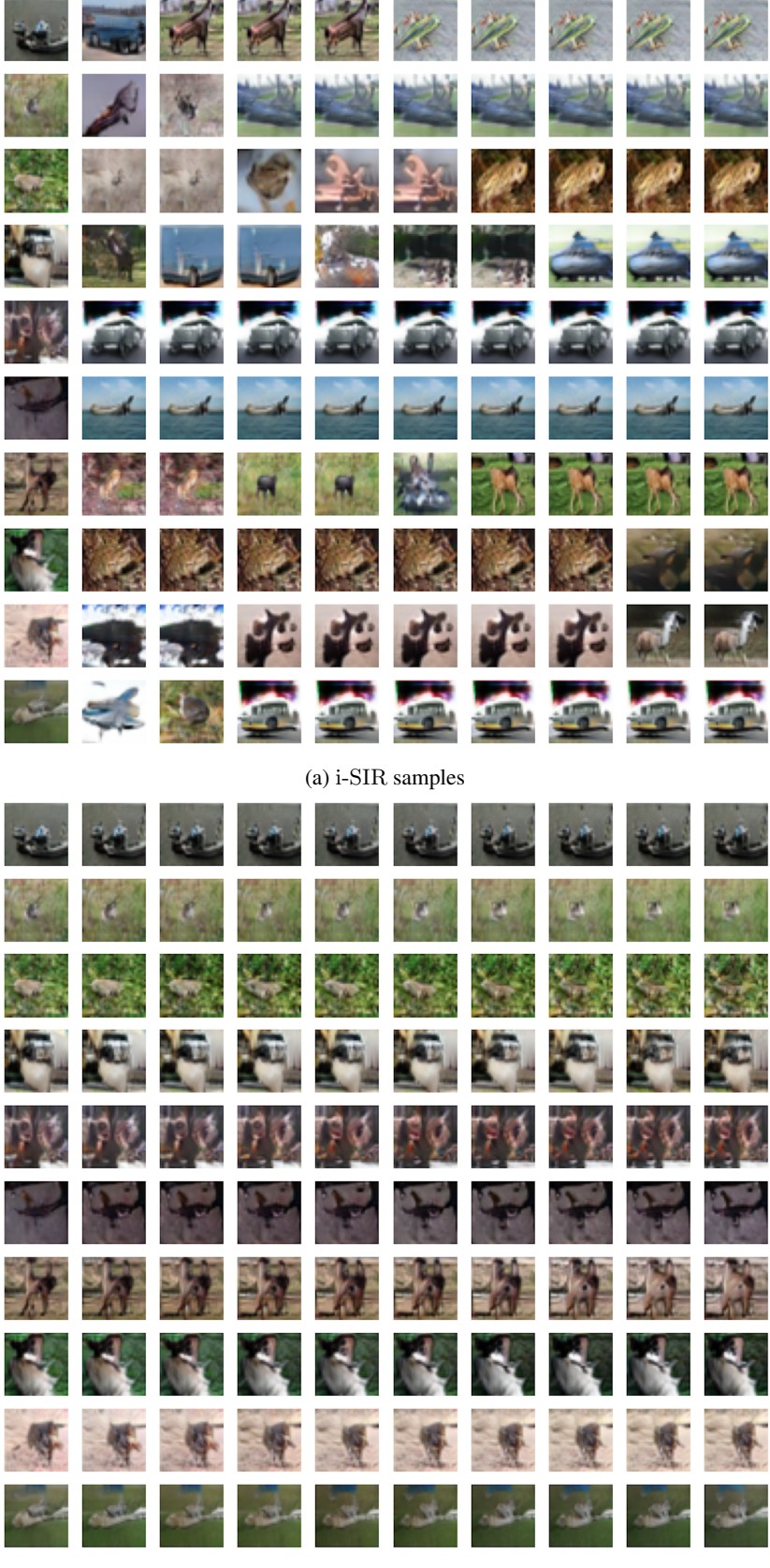

(a) i-SIR samples

(b) MALA samples

Figure 18: i-SIR and MALA samples, DC-GAN.

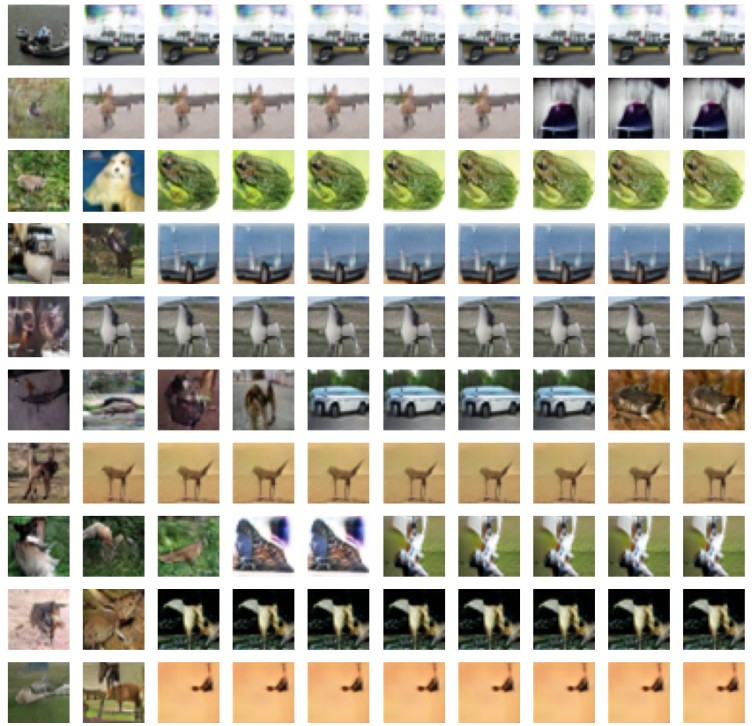

(a) Ex$^2$MCMC samples

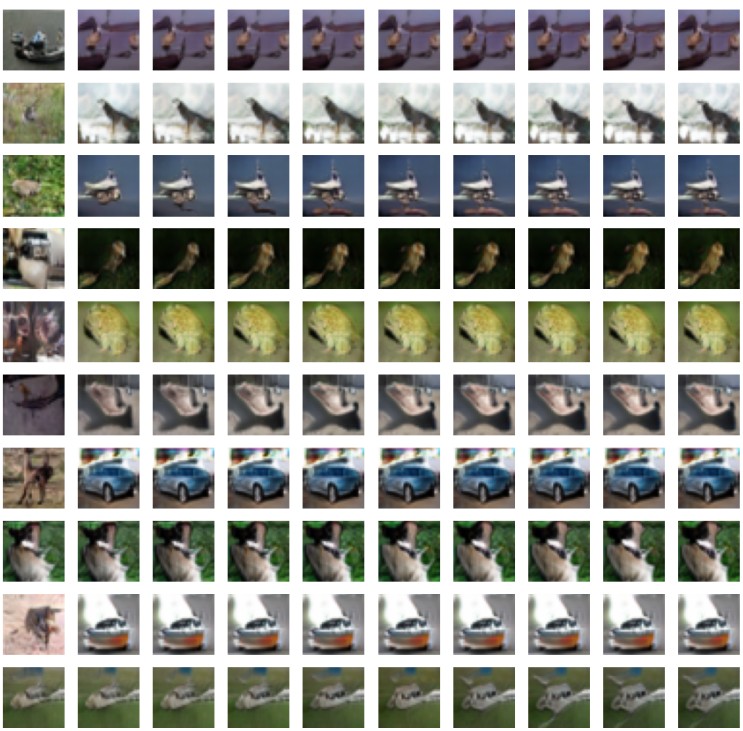

(b) FlEx$^2$MCMC samples

Figure 19: Ex$^2$MCMC and FlEx$^2$MCMC samples, DC-GAN.