# OpenReview forum: "Local-Global MCMC kernels: the best of both worlds"
_NeurIPS.cc/2022/Conference — NeurIPS 2022 Accept_

### Official Review · Reviewer_PqJ7 · 2022-07-07

**Rating:** 8
**Confidence:** 4
**Soundness:** 4 excellent
**Presentation:** 4 excellent
**Contribution:** 3 good

**Summary:**

This paper addresses MCMC sampling from unnormalized probability distributions. The main idea here is to combine sampling importance resampling using a global proposal with an interleaved local update -- a local MCMC transition kernel (such as MALA or HMC) with a metropolis correction. The authors argue that this simple approach effectively combines the best of both worlds -- faster mixing between modes (from the global proposal) and better local exploitation of modes (from the local kernel). The authors provide theory and empirical results which back up this claim. A key theoretical result is that the proposed method is geometrically ergodic even if the importance weights are unbounded (assuming properties of the local kernel which hold for most sensible choices). This is in contrast to i-SIR which requires bounded importance weights. Thus the approach retains the nice mixing properties of i-SIR while inheriting some convergences guarantees from the local methods.

The authors also propose an adaptive variant of the method where the global proposal is a normalizing flow which is tuned online using the outputs of the sampler. Modifying the proposal online can break ergodicity but the authors present sufficient conditions to guarantee it (basically just ensure that the flow converges).

The authors present some nice toy results to help visualize the performance of the approach compared to some obvious baselines (pure-global and pure-local MCMC). They then present results on some standard benchmark distributions of increasing dimension and show that their method outperforms most baselines except for NUTs (only outperformed in high dimensions). They then sample from high-dimensional EBM-GANs and show similar improvements.


**Questions:**

Most of my questions relate to the adaptive variant of the method. First, I am curious how important it is to train using the mixture of forward and reverse kl. Do we need both? I know you note (correctly) that reverse kl can suffer from mode-dropping, but it can also be made to work in many scenarios if done carefully. It would be great to have some sort of ablation on this? Maybe compare reverse only, forward only, and the mixture. I would not be surprised if the mixture performs better but it would be useful to back up the choice used here.

Next, I am curious how important the initialization of the proposal distribution is to the performance of the method. In my experience, reverse kl can be made to work if the proposal is initialized properly. Does the mixture save us from that? It might also be nice to see how the approach performs when we use a good init (distribution with the right scale) to a bad init (distribution with scale too small or too large). It would also be nice to discuss a practical initialization procedure such as: start with a wide proposal, run K steps of HMC and take the mean/std of the resulting samples and use that to parameterize the initial proposal. It would be very cool to see if the method is relatively robust to these choices.

**Limitations:**

I feel the limitations of the method are clear from reading the work. There is no specific limitations section (might be nice to add), but I do not think it is absolutely necessary.

**Strengths And Weaknesses:**

Strengths:

I enjoyed reading this work. This method is very simple (a good thing!) and once you hear it, it makes perfect sense. Its always great to see simple, intuitive ideas that appear to perform well. In my experience, it is methods like these that end up having the most impact (which is what we should all be trying to achieve). While the method is relatively simple, it is not obvious where and when it should outperform related methods. The theory shown here definitely makes that clear and does so in a way that is easily understandable and not over-cluttered with unnecessary math that would obfuscate the presentation of the method and the results.

The results are presented nicely. I like the progression from toy visualizations, to challenging standard benchmarks, to real-world deep learning models. I am convinced that the approach scales to high-dimensional problems and still performs well.

Weaknesses:

I do not have too many large critiques for the work but I do have a few questions (which will be expanded on in the next section).

I do feel there is some related work that has not been addressed (not my work I swear!). There has been a fair bit of work in the EBM space using a mix of local MCMC and global proposals ([3] is a good starting point but this family of methods has been explored quite thoroughly). Most of this has been focused on improving MCMC used to train energy-based models. To my knowledge, these methods have not been given the same theoretical treatment presented here, so it might be nice to compare these methods to your work and discuss where your theory could (or could not) be applied.

I feel the experimental results in the image generation section could be improved. I know it can be difficult to come up with quantitative measures of sampler performance for image data but some decent proxies do exist. I am not super-convinced by looking at energy values alone. Some measure which could compare likelihood and entropy would be nice. While not a perfect metric, FID could be used on the CIFAR experiments to give a solid proxy for this. Another idea would be to use a tractable model as a target. For example, you could train a normalizing flow on MNIST or CIFAR and sample from the flow, treating it as unnormalized. Then you could compare likelihood histograms between the MCMC samples and actual samples from the model. This has been done in other work dealing with unnormalized models [1, 2].

Like I said, I enjoyed this work, but some further large-scale experiments like this would greatly improve its experimental section.

[1] https://arxiv.org/pdf/1905.07088.pdf
[2] https://arxiv.org/abs/2010.04230.pdf
[3] http://www.stat.ucla.edu/~ywu/CoopNets/doc/CoopNets_AAAI.pdf

---

> ### Author Response · Authors · 2022-08-02
> **Answer to the Reviewer PqJ7**
>
> We thank the reviewer for their careful reading of our work and interesting suggestions of connected lines of works and further developments.
>
> Regarding the line of work by J. Xie and collaborators ([3] in the reviewers comment, but also [4]), it is an interesting point of comparison. The major difference in this line of work, as intuited by the reviewer, is the purely empirical motivation and validation of the proposed methods. Using typically successive restarts and short chains in "non-convergent MCMC", it seems extremely difficult to derive any theoretical guarantee for the proposed methods.
> A second important difference is the focus on EBM training and evaluation in this sense. It remains unclear how to evaluate even heuristically the performance of these algorithms as samplers per say, was again these are by definition not converged. One direction could be along the lines of [5], but we leave this investigation for future work.
>
> Following the reviewer's suggestion we computed the FID and IS (Inception score) of the images generated along the MCMC chains of the different algorithms in our MCMC-GAN experiment. We agree with the reviewer that these measures are imperfect, but still interesting to report. We report these metrics computed over $10000$ independent MCMC trajectories for each sampler. Please see Figure $5$ in the main text for DC-GAN IS results, and Figure $13$ in the Appendix E for the associated FID dynamics. The results demonstrate that the studied Ex2 and Flex2 MCMC typically reach low FID (and high IS) samples in a reasonable number of iterations. We could also provide similar dynamics for SN-GAN architecture, yet we did not have enough computational resources to evaluate both architectures in the limited time available.
>
> Finally, we also added an experiment on a 50-dimensional Gaussian mixture to quickly illustrate the impact of the choice of loss function and robustness to chain initializations, see Appendix E.5. Even with a relatively covering proposal initialization it is difficult to avoid mode collapse when training only with the backward KL. The reason is that in practice the bulk of the target distribution only occupies a very small fraction of the bulk of the proposal in such large dimension, and the training ends up focusing on the first mode for which fluctuations in the estimation of the backward KL provides signal in the loss. Following ref [6] using an annealing scheme for the target would likely solve this problem, yet a known difficulty of annealing is the sensitivity of success to careful choices of hyperparameters (high temperature, speed of annealing). Conversely, training online with the MCMC chain to evaluate the forward KL is robust, as demonstrated in the experiment, provided that rough locations of modes is known a priori. This last assumption is realistic in series of applications, e.g. trained EBM measure sampling where training data can serve as initializations, relative free energy computations and phase transition detections between collections of few known states in statistical mechanics.
>
> [4] A TALE OF TWO FLOWS: COOPERATIVE LEARNING OF LANGEVIN FLOW AND NORMALIZING FLOW TOWARD ENERGY-BASED MODEL - Jianwen Xie, Yaxuan Zhu, Jun Li, Ping Li - ICLR 2022
> [5]  https://arxiv.org/pdf/2110.13017.pdf
> [6] https://journals.aps.org/prl/abstract/10.1103/PhysRevLett.122.080602

---

> > ### Comment · Reviewer_PqJ7 · 2022-08-06
> > **Thanks**
> >
> > I thank the authors for their responses to my comments and questions. I appreciate the FID and IS evaluation on their GAN experiments and the results indicate the benefit from the proposed samplers. Initially, I enjoyed this work and advocated for its acceptance. My thoughts have not changed and my score advocating for the work's acceptance will remain the same.

---

### Official Review · Reviewer_9sXT · 2022-07-12

**Rating:** 6
**Confidence:** 3
**Soundness:** 4 excellent
**Presentation:** 3 good
**Contribution:** 2 fair

**Summary:**

The authors propose to combine a global MCMC kernel with a local MCMC kernel to achieve exploration and exploitation. Specifically, they use iterate sampling importance resampling proposed in [H. Tjelmeland, 2004] as the global kernel and MALA as the local kernel. To further improve the proposal distribution in the global kernel, the authors apply normalizing flow to learn a proposal that is close to the target distribution. The authors prove the V -uniform geometric ergodicity of the proposed method. Empirical results on synthetic distributions, and the latent space of Generative adversarial networks demonstrate the effectiveness.



**Questions:**

In Figure 1, how does local MCMC (which is MALA on this task) perform? On a standard Gaussian, I guess MALA should perform well. It might be more convincing to show a case, such as the multimodal distribution in Section 5.1, where Ex2MCMC is better than purely i-SIR and purely MALA.

Theorem 2 provides the mixing rate of the proposed Ex2MCMC. Could the authors compare it with the mixing rate of i-SIR and MALA? Under which conditions, Ex2MCMC is faster? I do not find a discussion on it in the current form.

It might be clearer to summarize the adaptive Ex2MCMC in an algorithm box, then we can see what are the differences between Ex2MCMC and adaptive Ex2MCMC by comparing it with Algorithm 2.

The related work section is a bit vague. It is unclear to me how the proposed method differs from previous work and what are the pros and cons. For example, [1] also combines local and global MCMC kernels and the global kernel is also learned by a normalizing flow. Is the difference only in the global sampler where [1] uses independent MH and the proposed method uses i-SIR? In the second paragraph, the authors simply claimed, “another line of work exploits both normalizing flows and common local MCMC kernels for sampling, yet in a different way” without explaining in what different way.

In the experiments, I think it is necessary to include previous local-global methods, such as [1], as baselines. The empirical results look reasonable to me. It is expected that combining local and global moves improves the sampling results on multimodal distributions.



**Limitations:**

There is no related discussion on limitations in the paper.

**Strengths And Weaknesses:**

Strengths

- The motivation is clear and the paper is easy to follow.

- As far as I understand, the proposed method is technically sound.

- The authors provide both theoretical and empirical analysis to the proposed method.

Weaknesses

- It is unclear whether the methodology significantly differs from previous local-global MCMC work. Thus, the originality is unclear.

- The empirical results do not include previous local-global MCMC methods as baselines, so it is unclear whether the empirical improvement is significant.

My main concern is the originality and significance. To me, the proposed method is very similar to [1] and the authors did not compare it with [1] in the experiments. If the authors can show significant differences and improvements of the proposed method over the previous work in the rebuttal, I'm happy to increase the score.

[1] Adaptive Monte Carlo augmented with normalizing flows

---

> ### Author Response · Authors · 2022-08-02
> **Originality and novelty compared to the previous local-global MCMC works**
>
> Concerning the reviewer's main concern on originality, we clarify in the revised version through proper vocabulary and extension of the related-works section that we do not claim that combining local and global MCMC kernels is unheard of. In particular, as noted by the reviewer the main algorithmic difference between the Ex2MCMC we analyze and the method of [1] is the type of global sampler. However, the present submission provides for the first time clear mathematical foundations and thorough empirical demonstration of the advantages of a sampler of this local-global type.
>
> In particular, we would like to emphasize an important motivation of our paper. If we compose two Markov kernels $P$ and $Q$ which have the same invariant distribution $\pi$, the composed kernel $PQ$ admits $\pi$ as an invariant distribution. If the kernel $P$ is positive recurrent (but not geometrically ergodic) and $Q$ is  $V$-geometrically ergodic,  the composed kernel $PQ$ will by no means be $V$-geometrically ergodic in general. Therefore, one cannot  compose two kernels and expect the resulting kernel to have the “nice” ergodicity properties. The fact that one can combine the i- SIR kernel with any local $V$-uniform geometric ergodic kernel to obtain a composite $V$-uniform geometric ergodic kernel follows from the fact that the i- SIR kernel verifies a "weak" Foster-Lyapunov condition for any drift function (integrable under the invariant law)- Lemma 9-10-11-. The two key properties of the i-SIR algorithm in the analysis are therefore:
>
> - All the sets on which the importance functions are bounded are "small-sets" (satisfy a “local” Doeblin condition)
> - All the functions integrable under the proposal law and the target law satisfy a weak Foster-Lyapunov condition (Lemma 9).
>
> To the best of our knowledge these precise analyses are entirely novel.

---

> ### Author Response · Authors · 2022-08-02
> **Comparing the mixing rate of Theorem 2 with the mixing rates of i-SIR and MALA**
>
> Our review of the literature on iSIR in Section 2.1 recalls that geometric ergodicity can only be demonstrated for iSIR in the case where the ratio of the density of proposal and target density is uniformly bounded. Theorem 2 shows that these restrictive assumption can be relaxed for Ex2MCMC thanks to the rejuvenation kernel. Thus our theoretical guarantees for Ex2MCMC are in fact stronger that the ones available for iSIR.
>
> Regarding the reviewer’s question on the comparison of the theoretical guarantees obtained for Ex2MCMC and MALA, Theorem $20$ in Appendix C shows the relative improvement of the Ex2MCMC mixing time compared to the one of the MALA. The improvement comes from the fact that available bounds for the mixing rate of the MALA kernel depends upon the minorization constant of the MALA kernel (see Proposition $15$). The quantity $\varepsilon(K)$ in the latter bound decreases exponentially fast with the growth of the problem dimension $d$. So the convergence rate $\rho_{\overline{\Gamma}}$ for the MALA provided in Theorem $16$ becomes extremely close to $1$. At the same time, the minorization constant of the Ex2MCMC kernel is estimated in Lemma 8 (see also Lemma 10), and is typically much larger, allowing for better convergence rate if the number of proposals $N$ is large enough. This behaviour is also illustrated in the discussion after Theorem $20$.

---

> ### Author Response · Authors · 2022-08-02
> **Clarifying the related work section and Figure $1$**
>
> We re-wrote part of the related work section to clarify what might have been too vague in the first version. Indeed the difference of the algorithm described here and in [1] is the global sampler as already stated above. Also, the different way of combining the flows and local sampling algorithms to which the quote  "another line of work exploits both normalizing flows and common local MCMC kernels for sampling, yet in a different way" refers to is  "neural transport". This method is described in the sentences following the quote in l-230-236. We hope the current formulation makes it clearer that the description follows the quote.
>
> As suggested by the reviewer, we added an algorithmic box for the Flex2MCMC algorithm to improve readability of this adaptive variant. We also moved the former Figure 1 to the Appendix: while the illustration on the single Gaussian of the deterioration of the performance of iSIR alone with increasing dimension (when the proposal is mismatched) is informative (former Figure 1), we agree with the reviewer that a multi-modal example for which the advantage of Ex2MCMC over both i-SIR and MALA is clear is a better first picture (former Figure 2 which became Figure 1).

---

> ### Author Response · Authors · 2022-08-02
> **Comparison with the paper [1]**
>
> As our main goal is to demonstrate the benefit from combining local and global kernels we focus our experiments on ablation comparisons with only global or only local algorithms rather than with the local-global sampler of [1] which fits in the framework advocated here. With more time, we could add this algorithm in our experiments, expecting performances similar to Ex2MCMC for comparable computational budgets: this intuition stems from the reference [2] discussing performance of IMH and Multiple-try-Metropolis (MTM) - the reference was added to the related works section. Still we note that a computational advantage of the i-SIR algorithm is an easy parallel implementation (as most of the computational cost comes from the computation of propositions and importance functions) whereas the IMH algorithm is inherently sequential in treating a similar number of proposals and thus is more challenging to speed-up. It is clear that for high dimensional problems, a large pool of candidates ($N=10^3$ to $N=10^4$) is highly beneficial when available resources allow.
>
> Finally, let us emphasize the contributions of the present submission compared to [1]:
>
> - Proofs for the V-geometric ergodicity of the local-global kernel, in particular showing that the local part of the kernel allows for the proof to go through even if the ratio of the density of proposal and target density are unbounded (see Section 3, Theorem 2). We also quantify the mixing time improvement of Ex2MCMC as compared to the local sampler only in Appendix C, Theorem 20.
> - Proof of the convergence of the adaptive scheme (Section 4).
> - Empirical qualitative clarification of the importance of the local kernel in sampling the tails of the distribution with complex geometries (Section 5.1).
> - Quantitative ablation studies of the local/global kernels demonstrating the benefit arising from the combinaison. (Section 5)
>
> We believe that these non-trivial clarifications are very valuable to practitioners.
>
> [2] M. Bédard, R. Douc, and E. Moulines. Scaling analysis of multiple-try MCMC methods. Stochastic Processes and their Applications, 122(3):758–786, 2012.

---

> > ### Comment · Reviewer_9sXT · 2022-08-08
> > **Thanks**
> >
> > I thank the authors for their responses.
> >
> > I'm a bit confused by the reasons that the authors did not compare with [1]. Doesn't [1] already demonstrate the benefit from combining local and global kernels? The difference of the proposed method seems to be only the global kernel and the mentioned advantage also comes from this kernel (i-SIR is easy to parallel while IMH is not). I'm not fully convinced that this is a significant algorithmic contribution.
> >
> > In my opinion, the main contribution of the paper is the theoretical results, which show that the "i-SIR kernel with any local-uniform geometric ergodic kernel to obtain a composite-uniform geometric ergodic kernel" (from the response on "originality and novelty" below). I agree that these results are new and interesting, which might inspire new local-global algorithms.
> >
> > I encourage the authors to clarify the contributions in the revision, situating this paper clearly in the literature.

---

> ### Author Response · Authors · 2022-08-02
> **Answer to the Reviewer 9sXT**
>
> We thank the reviewer for their assessment of our work and pointing us towards possible improvements. Please check below our answers to your concerns.

---

### Official Review · Reviewer_UfPV · 2022-07-12

**Rating:** 6
**Confidence:** 3
**Soundness:** 4 excellent
**Presentation:** 4 excellent
**Contribution:** 3 good

**Summary:**

This work proposes a novel MCMC kernel which consists of a "local" move $\mathsf{R}$ (e.g. a Metropolis-adjusted Langevin update) followed by a "global" i-SIR update. The latter is an MCMC kernel which proposes multiple particles from a distribution $\lambda$ which does not depend on the current state of the Markov chain and then potentially accepts one of these as the new state. The authors prove that the algorithm is geometrically ergodic.

The authors also consider adapting the proposal distribution $\lambda$ to the target distribution by specifying it through a normalising flow and taking gradient steps to minimise a suitable convex combination of forward and backward KL divergence at each iteration. Under additional assumptions, they show that the amount of adaptation converges to zero, i.e. adapting the proposal does not "break" the algorithm.

**Questions:**

I'm curious why this has been submitted here rather than to a computational statistics journal where the theoretical guarantees may find more appreciation (and where reviewers have more time to review these).


**Limitations:**

Yes

**Strengths And Weaknesses:**

Strengths:

1. Extensive theoretical guarantees including for the adaptive case (although I didn't not check the proofs in the appendix). To my knowledge, these are novel.

2. The presentation is very clear.

3. The proposed algorithm is demonstrated to be competitive in low to moderate dimensions.

Weaknesses:

1. The proposed algorithm itself is not the most original since it is simply a composition of two well-known MCMC kernels.

2. I appreciate that the page limit is very tight but it would be useful to have more explanation of the constants that appear in Theorem 2 in the main manuscript. For instance, in Theorem 2, does the analysis demonstrate that the convergence rate is provably better for the proposed algorithm than for just the rejuvenation kernel alone?

3. In the "multimodal distributions" example in Section 5.1, I think it needs to be stressed that the proposed method would likely not work well for larger dimensions $d$ (the example uses $d = 2$). The way the example is presented might give some non-experts the idea that the proposed algorithm can overcome the problem targetting a distribution with well-separated modes in higher dimensions which is not the case. Of course, in fairness, I don't know of any other method that would work in such a scenario.


Some typos:

- l.117: missing space after "strategy"
- l.213: work -> works
- l.213: alternates -> alternate
- l.218: author -> authors
- l.248: stalls -> stall
- l.257: "remind"
- l.257: define the "$[x;y]$" notation
- l.272: outperforms -> outperform
- Bibliography: there are missing details/inconsistent use of journal name abbreviations/missing capitalisation of proper nouns in a number of entries
- Theorem 2: Equation 13 is referenced here but can only be found in the appendix

---

> ### Author Response · Authors · 2022-08-02
> **Answer to the Reviewer UfPV**
>
> We thank the reviewer for their appreciation of our work and pointing us to improvements.
>
> As far as originality is concerned, as pointed by two of the reviewers, the main idea behind the investigated algorithm is simple, yet little has been done in the litterature to clarify very concretely why this local-global combinaison is remarkably efficient. In this paper we gather theoretical guarantees, pedagogical visualizations and assessments in challenging settings to explain the advantage of the approach.
>
> We would like to stress that the main theoretical result, which the reviewer acknowledges as novel, is non-trivial. If we take any two kernels, $P$ and $Q$, where $P$ has $\pi$ as invariant law and $Q$ is $V$-geometrically ergodic and has $\pi$ as invariant law, their composition $PQ$ admits $\pi$ as its invariant measure but is not necessarily $V$-geometrically ergodic. Thus, one cannot "compose" two different kernels targeting the same invariant distributions and expect good results. We show that it is justified to associate a kernel of type i- SIR and a local $V$-geometric kernel like MALA or HMC [or iterations of MALA].
>
> Following the reviewer's suggestion we extended our comments in the main text around Theorem 2, in order to provide better intuition on the improvement of the combined kernel over rejuvenation kernel $R$ alone. Theorem $20$ in Appendix C shows the relative improvement of the Ex2MCMC mixing time compared to the one of the MALA. The improvement comes from the fact that available bounds for the mixing rate of the MALA kernel depends upon the minorization constant of the MALA kernel (see Proposition $15$). The quantity $\varepsilon(K)$ in the latter bound decreases exponentially fast with the growth of the problem dimension $d$. So the convergence rate $\rho_{\overline{\Gamma}}$ for the MALA provided in Theorem $16$ becomes extremely close to $1$. This behaviour is also illustrated in the discussion after Theorem $20$. At the same time, the minorization constant of the Ex2MCMC kernel is estimated in Lemma 8 (see also Lemma 10), and is typically much larger, allowing for better convergence rate if the number of proposals $N$ is large enough.
>
> Concerning the presentation of Section 5.1 which the reviewer feared could be misleading, we added a paragraph clarifying the high-dimensional case and pointing to a new experiment in the Appendix E.5 (see Figure 9). Namely, the reviewer is right to expect that there would be no mixing in high dimension with the unimodal Gaussian proposal to target the Gaussian mixture because of the poor overlap between the latters. However, our new experiment in 50-dimension illustrates that the adaptive proposal of the discussed Flex2MCMC can lead to good mixing, provided that the rough location of modes is known a priori. Our experiment also displays the failure of the algorithm when this condition is not fulfilled so as to avoid any confusion for non-expert readers. Finally, this experiment also adds a discussion on loss functions upon the suggestion of another reviewer.
>
> There are multiple reasons why the present paper was submitted at NeurIPS. We note that today it would be impossible to propose an optimization algorithm without a sound theoretical proof (while optimization is at the core of operations research mathematics), but that in sampling problems (which are also important) the concern for justifying the theoretical performance of algorithms takes a back seat (with some exceptions). On the contrary, we believe that today it is important not only to propose new algorithms, but also to develop theoretical guarantees, even if they are less "straightforward" than for optimization algorithms, because obtaining long-term convergence of Markov chains remains a difficult problem! This is the first reason that led us to present this theory. The second reason is that the structure of the Ex2MCMC algorithm is particularly well suited to sampling problems such as those encountered in generative models. The best example of this is the GAN MCMCs, for example, where we "already" have a "good" generator (which we can of course improve by the adaptive procedure). So, there is something in the structure of the problems we face that is "directly" inspired by the context of generative models. Finally, we believe that it is important to share with the machine learning community the merits but also the limitations of applications of ML methods across computational fields. The present work shows that local traditional kernel should not be replaced by trained proposals but that the combination of both is synergetic.

---

> > ### Comment · Reviewer_UfPV · 2022-08-08
> > **Thank you for the detailed response**
> >
> > The authors have satisfactorily addressed my concerns about the paper (which were rather minor, to begin with). I continue to support its acceptance.

---

### Author Response · Authors · 2022-08-02
**New revision**

We thank the three reviewers for their feedback. Within the allowed time, we proceeded to a revision along their suggestions, which we believe improves further our submission. Let us summarize here the changes:

- We extended comments around our main theoretical result, both in the main text (after Theorem 2) and in the Appendix C. In particular, we reviewed the disussion on the comparison of mixing rates of the MALA and Ex2MCMC after Theorem 20.
- We added an algorithmic box for the adaptive version of the algorithm (Algorithm 3).
- We rewrote the related works section to clarify how the present submission relates to previous works.
- We added an experiment on a high-dimensional mixture to illustrate the robustness of local-global kernels, provided that proper training losses and chain initializations are used (Appendix E.5).
- We computed image quality metrics (Frechet Inception Distance and Inception Score) to complement the results of our EBM-GAN experiments, showing that the local-global kernels are typically sampling "better" images than purely local or purely global samplers (Figure 5, Figure 13).
- We moved former Figure 1 comparing the scaling for increasing dimension of the performance of iSIR and Ex2MCMC in sampling a unimodal Gaussian to the Appendix (now Figure 7).
- We modified the former Figures 2 and 3 into now Figure 1 (main text) and Figure 8 (Appendix).

Changes are highlighted in color in the revised pdf.

---

### Meta-Review · Area_Chair_4tQ6 · 2022-08-27

**Recommendation:** Accept
**Confidence:** Certain

**Metareview:**

Three domain experts recommended acceptance for this paper. One reviewer in particular made a confident and strong recommendation. I find all three convincing.

Reviewers agree that the paper is clear (both in presenting the motivation and the argument), and that the contribution to theory is substantial (proving desirable mixing properties and convergence guarantees, and developing an adaptive variant of the algorithm). Reviewers also agree on the whole that the empirical section adequately supports the paper's main point, even though (as Reviewer PqJ7 mentioned) larger-scale simulations would strengthen it further.

There were some questions raised early on about related work. Reviewer PqJ7 pointed out specific related lines of work worth discussing in the paper, and a thread with Reviewer 9sXT drew some comparisons to another paper that combines local and global MCMC kernels. In both cases, concerns seemed adequately addressed during the discussion that ensued, and were covered in the draft update. Reviewer PqJ7 initially suggested computing additional metrics (namely FID) as part of the experiments and the reviewers did this too (adding an IS calculation as well).

Discussion was productive – a case where the course of review strengthened the paper slightly further. Thanks to the reviewers for being thorough and responsive and to the authors for incorporating their feedback well. Altogether this is a nice and well-presented result.

**Award:**

No

---

### Decision · Program_Chairs · 2022-09-14

Accept